# ARTICLES

# Integrated multi-omics reveal polycomb repressive complex 2 restricts human trophoblast induction

Dick W. Zijlmans[1,13], Irene Talon[2,13], Sigrid Verhelst[3,13], Adam Bendall[4,13], Karlien Van Nerum[2], Alok Javali[5], Andrew A. Malcolm[4,6], Sam S. F. A. van Knippenberg[2], Laura Biggins[7], San Kit To[2], Adrian Janiszewski[2], Danielle Admiraal[8], Ruth Knops[9], Nikky Corthout[10], Bradley P. Balaton[2], Grigorios Georgolopoulos[2], Amitesh Panda[2], Natarajan V. Bhanu[11], Amanda J. Collier[4], Charlene Fabian[4], Ryan N. Allsop[2], Joel Chappell[2], Thi Xuan Ai Pham[2], Michael Oberhuemer[2], Cankat Ertekin[2], Lotte Vanheer[2], Paraskevi Athanasouli[2], Frederic Lluis[2], Dieter Deforce[3], Joop H. Jansen[9], Benjamin A. Garcia[11], Michiel Vermeulen[1,8], Nicolas Rivron[5], Maarten Dhaenens[3,14 ✉], Hendrik Marks[8,14 ✉], Peter J. Rugg-Gunn[4,6,12,14 ✉] and Vincent Pasque[2,14 ✉]

**Human naive pluripotent stem cells have unrestricted lineage potential. Underpinning this property, naive cells are thought to lack chromatin-based lineage barriers. However, this assumption has not been tested. Here we define the chromatin-associated proteome, histone post-translational modifications and transcriptome of human naive and primed pluripotent stem cells. Our integrated analysis reveals differences in the relative abundance and activities of distinct chromatin modules. We identify a strong enrichment of polycomb repressive complex 2 (PRC2)-associated H3K27me3 in the chromatin of naive pluripotent stem cells and H3K27me3 enrichment at promoters of lineage-determining genes, including trophoblast regulators. PRC2 activity acts as a chromatin barrier restricting the differentiation of naive cells towards the trophoblast lineage, whereas inhibition of PRC2 promotes trophoblast-fate induction and cavity formation in human blastoids. Together, our results establish that human naive pluripotent stem cells are not epigenetically unrestricted, but instead possess chromatin mechanisms that oppose the induction of alternative cell fates.**

Epiblast and trophectoderm cells of the human embryo display a prolonged period of developmental plasticity. Contrary to the mouse blastocyst, where the epiblast and trophoblast lineages are restricted, these lineages are not yet committed in human blastocyst[1–7]. This unrestricted lineage potential of cells of early human blastocysts is retained in naive human pluripotent stem cells (hPSCs), derived from pre-implantation blastocysts, which have the potential to differentiate into both embryonic and extra-embryonic cell types including the trophoblast lineage[6,8–14]. The developmental plasticity of naive hPSCs also endows them with the capacity to form blastoids, which are generated from naive hPSCs that self-organize into structures resembling blastocysts[15–19]. In contrast, primed hPSCs share properties with postimplantation epiblast cells and differentiate into trophoblast cells less efficiently. Hence, they are not suitable to generate blastoids[13,17–21].

Trophoblast cells rarely arise spontaneously in robust naive hPSC cultures but they can be converted from this state using trophoblast stem cell culture conditions[6,11–14]. This suggests that the trophoblast fate is actively suppressed in naive hPSCs and is activated in response to appropriate cues and in a regulated manner. Considering the important role of chromatin-based processes in regulating cell identity, this raises the possibility that epigenetic barriers could exist to regulate the transition from naive pluripotency towards the trophoblast lineage. Defining these barriers would shed light on developmental mechanisms regulating developmental transitions and lead to better control of trophoblast specification and differentiation.

¹Department of Molecular Biology, Faculty of Science, Radboud Institute for Molecular Life Sciences (RIMLS), Oncode Institute, Radboud University Nijmegen, Nijmegen, The Netherlands. ²Department of Development and Regeneration, Leuven Stem Cell Institute, Leuven Institute for Single Cell Omics (LISCO), KU Leuven, Leuven, Belgium. ³ProGenTomics, Laboratory of Pharmaceutical Biotechnology, Ghent University, Ghent, Belgium. ⁴Epigenetics Programme, Babraham Institute, Cambridge, UK. ⁵Institute of Molecular Biotechnology of the Austrian Academy of Sciences, Vienna, Austria. ⁶The Wellcome–MRC Cambridge Stem Cell Institute, University of Cambridge, Cambridge, UK. ⁷Bioinformatics Group, Babraham Institute, Cambridge, UK. ⁸Department of Molecular Biology, Faculty of Science, Radboud Institute for Molecular Life Sciences (RIMLS), Radboud University Nijmegen, Nijmegen, The Netherlands. ⁹Laboratory of Hematology, Department of Laboratory Medicine, Radboud University Nijmegen Medical Centre (RadboudUMC), Nijmegen, the Netherlands. ¹⁰VIB Center for Brain and Disease Research, KU Leuven, VIB Bioimaging Core, Leuven, Belgium. ¹¹Epigenetics Institute, Department of Biochemistry and Biophysics, Perelman School of Medicine, University of Pennsylvania, Philadelphia, PA, USA. ¹²The Centre for Trophoblast Research, University of Cambridge, Cambridge, UK. ¹³These authors contributed equally: Dick W. Zijlmans, Irene Talon, Sigrid Verhelst, Adam Bendall. ¹⁴These authors jointly supervised this work: Maarten Dhaenens, Hendrik Marks, Peter J. Rugg-Gunn, Vincent Pasque. ✉e-mail: Maarten.Dhaenens@UGent.be; Hendrik.Marks@ru.nl; peter.rugg-gunn@babraham.ac.uk; vincent.pasque@kuleuven.be

Chromatin and epigenetic-based processes are key regulators of cell identity, fate specification and developmental gene expression programmes[22–24]. Striking differences in the transcriptome, DNA methylome and genome organization have been uncovered between naive and primed hPSC states, which correspond to their distinct developmental identities[8,25–29]. A limited number of other chromatin-based epigenetic properties have also been examined in naive and primed hPSCs, including histone H3 lysine 27 trimethylation (H3K27me3), which is a histone modification catalysed by polycomb repressive complex 2 (PRC2) and is associated with transcriptional repression[30]. H3K27me3 levels differ between human pluripotent states, although it remains unclear whether global levels of H3K27me3 are higher in primed hPSCs compared with naive hPSCs[31], or the opposite[32]. Genome mapping by chromatin immunoprecipitation with sequencing showed that a greater number of gene promoters are marked by H3K27me3 in primed hPSCs compared with naive hPSCs[28,33]. It therefore remains enigmatic which chromatin-associated proteins and histone post-translational modifications (hPTMs) characterize and regulate the unrestricted lineage potential of naive hPSCs.

Naive hPSCs can be maintained in the absence of epigenetic repressors, including PRC2, DNMT1 and METTL3, whereas these factors are required for stable self-renewal and maintaining the pluripotent status of primed hPSCs[21,34–36]. Based on these observations, naive hPSCs are considered 'epigenetically unrestricted'. However, because the role of chromatin-based mechanisms in controlling the transcriptome, epigenome and differentiation potential of naive hPSCs has not been examined, whether these mechanisms establish a lineage barrier in human cell pluripotency and control fate specification remains an important unresolved question.

Here we apply an integrated multi-omics approach to comprehensively map the chromatin-associated proteome, hPTMs and transcriptome of naive and primed hPSCs. We unexpectedly discovered that PRC2 activity opposes the induction of trophoblast in naive hPSCs and blastoids, thereby establishing that naive pluripotent cells are not epigenetically unrestricted but that instead, chromatin barriers limit their ability to differentiate into trophoblast.

## Results

**Comprehensive chromatin profiling in hPSCs.** To define the chromatin landscapes of hPSCs, we performed an integrated multi-omics analysis of naive (cultured in PXGL medium) and primed (cultured in E8 medium) H9 human embryonic stem cells (hESCs; Fig. 1a). This analysis incorporated chromatin proteomes, DNA methylation levels, hPTMs and transcriptomes, thus including chromatin regulatory factors as well as their modifications and transcriptional outcomes. For convenient access to these data, we created a searchable tool to explore the data (https://www.bioinformatics.babraham.ac.uk/shiny/shiny_omics/Shiny_omics). Transcriptional analyses validated the anticipated expression of pluripotent-state markers (Extended Data Fig. 1a and Supplementary Tables 1,2).

To identify chromatin regulators associated with both pluripotent states, we analysed chromatin-bound proteins using chromatin enrichment for proteomics (ChEP), followed by mass spectrometry[37,38]. We identified 4,576 proteins, of which 1,819 changed significantly between the naive and primed states ($P < 0.05$, fold change (FC) > 2; Fig. 1b and Supplementary Table 3). Gene ontology analysis of the chromatin-bound proteins that were more abundant in primed hPSCs showed an association with development and neuronal differentiation (Fig. 1c), in agreement with the more advanced developmental stage of primed hPSCs[39]. Gene ontology terms associated with proteins that were more abundant in naive hPSCs included transcriptional regulation and RNA processing (Fig. 1c).

We next analysed prominent proteins involved in pluripotency, DNA methylation and chromatin remodelling (Fig. 1d and Extended Data Fig. 1b). In naive hPSCs, we identified an increase in the chromatin-associated levels of known naive factors (KLF4, KLF5 KLF17, TFCP2L1, PRDM14 and TFAP2C) in addition to unanticipated factors (UTF1, DPPA2, LIN28B and MYCN)[8,28,40]. In primed hPSCs, transcription factors including ZIC2, ZIC5, LIN28A and L1TD1 were more abundant compared with naive hPSCs[41]. Shared proteins included core pluripotency factors (POU5F1, SALL4 and SOX2) and chromatin remodellers (BRD3, BRD4 and SMARCC2; Fig. 1d and Extended Data Fig. 1b,c).

We confirmed that naive hPSCs were globally DNA hypomethylated compared with primed hPSCs (Extended Data Fig. 1d), corroborating previous findings[8,26,42]. Despite this difference in DNA methylation, our ChEP analysis showed that there was little to no change in the chromatin-bound levels of the DNA methyltransferases DNMT3A and DNMT3B when naive and primed hPSCs were compared (Fig. 1d). However, we detected a decrease in DNMT1 and its known interactor UHRF1 (ref. [43]) as well as an increase in TET1 in naive hPSCs, which are differences that could potentially reinforce the hypomethylated state of naive hPSCs.

We detected changes between members of several chromatin regulatory complexes between naive and primed hPSCs, including PRC1, PRC2, NuRD and histone deacetylase complexes (Fig. 1e and Extended Data Fig. 1c). For PRC2, we noticed a modest increase in core components in naive cells as well as increased MTF2 and decreased JARID2, which suggests a shift in PRC2 subcomplexes from PRC2.2 to PRC2.1. Finally, we found changes in ATP-dependent chromatin remodelling complexes (Fig. 1d and Extended Data Fig. 1c). Notably, we detected higher levels of SMARCA2 (also known as BRM) in the naive state and SMARCA1 (also known as SNF2L) in the primed state, in line with the OCT4-specific association of these factors to regulate chromatin accessibility[43,44].

Together, this analysis identified a compendium of chromatin-associated proteins in naive and primed pluripotent states, including widespread differences in DNA-binding factors as well as in the writers, readers and erasers of hPTMs, highlighting the distinct chromatin landscapes of human pluripotent states.

**hPTMs of pluripotent cells.** Although hPTMs are pivotal mediators of chromatin structure and function, they have mainly been studied in hPSCs using targeted sequencing-based approaches[25,28,33,40,45]. We performed a bottom-up mass spectrometry analysis following acid extraction to assay the global abundance of hPTMs and histone variants. This approach quantified 43 individual hPTMs on histones H3 and H4 (Fig. 2a,b and Supplementary Table 4), of which 23 were significantly different between the two cell states ($P < 0.05$). There was a strong increase in PRC2-mediated H3K27me2 and H3K27me3, and DOT1L-mediated H3K79me2 in naive cells compared with primed cells. Modifications that were lower in naive hPSCs included H3K27 acetylation and H3K36me2, which is consistent with the antagonism of these modifications with H3K27me2 and H3K27me3 (refs. [46–48]), as well as a global decrease in H4-tail acetylation. Histone variants also affect chromatin states (Extended Data Fig. 2a). Particularly in naive hPSCs, the histone H1 and H2A repertoire shifts significantly, with a prominent increase in H1.1 and H2A1B/1H, and a decrease in the H2A variants H2AW and H2AY (macroH2A). Extending these findings, we performed hPTM profiling in the same H9 hPSC line but using alternative naive cell culture conditions (t2iLGö medium[8]) and mass spectrometry protocols, and furthermore compared the results with previously profiled H9 hPSCs cultured in ENHSM medium[32] (Fig. 2c). Overall, the hPTM patterns were similar for the three culture conditions, further validating the differences between naive and primed states.

Histone profiling additionally identifies alkaline proteins that are co-purified, referred to as the acid extractome, which contains many nucleic-acid binders (Supplementary Table 5)[32]. There was a good correlation between the abundance of proteins detected in both the chromatin proteomes and acid extractomes (Extended

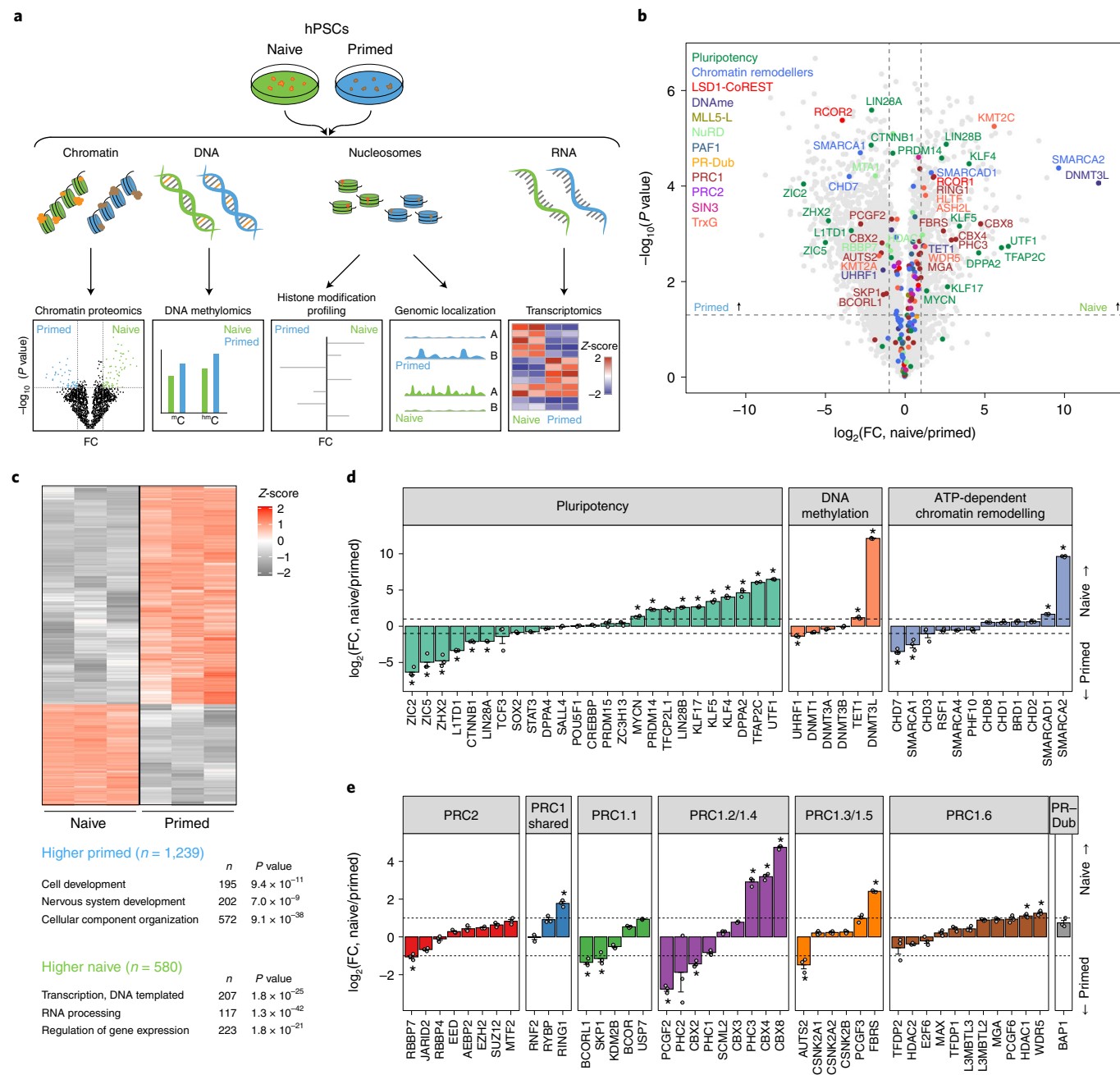

**Fig. 1 | Naive and primed hPSCs contain distinct chromatin proteomes. a**, Schematic of the workflow used for multi-omics profiling of naive and primed hPSCs. **b**, Volcano plot of quantified chromatin-associated proteins ($n = 4{,}576$ proteins) in naive and primed hPSCs. Major classes of chromatin regulators and protein complexes, indicated in the top-left corner, are highlighted. Complex members and regulators with significantly changed levels of expression (two-sided Student's $t$-test, $P < 0.05$, FC > 2) are labelled; $n = 3$ biologically independent samples for each cell type. Horizontal dashed lines represent $P = 0.05$; vertical dashed lines represent FC = 2. **c**, Heatmap of the normalized $Z$-score of the differentially expressed proteins identified in **b** ($n = 1{,}819$ proteins; top). Representative gene ontology terms for proteins that were significantly enriched (two-sided Student's $t$-test, $P < 0.05$; FC > 2) in naive or primed hPSCs are listed (bottom). **d,e**, Comparison of the chromatin occupancy for major regulators of pluripotency, DNA methylation and chromatin remodelling (**d**), and polycomb repressive complexes (**e**) between naive and primed hPSCs ($n = 3$ biologically independent samples). Data are presented as the mean ± s.e.m. The dashed lines represent FC = 2; *$P < 0.05$ and FC > 2 (two-sided Student's $t$-test). **e**, Only high-change proteins ($\log_2(\text{FC}) > 0.5$) involved in ATP-dependent chromatin remodelling are shown. Low-change proteins are shown in Extended Data Fig. 1d. Source data are provided.

Data Fig. 2b). The acid extractome adds insights by identifying proteins that were not detected in the chromatin proteome; for example, the WNT signalling regulator APC2 is increased in naive hPSCs (Extended Data Fig. 2c). In addition, the acid extractome showed higher levels of ribosomal and nucleolar proteins in naive compared with primed hPSCs (Extended Data Fig. 2d), in line with the

enrichment of the gene ontology term 'RNA processing' observed in the transcriptomics data (Fig. 1c). Related to this, MMP-2 activates ribosome biogenesis by enzymatic clipping of the histone H3 amino (N)-terminal tail following binding of the ribosomal-RNA gene promoter[49], which can initiate ribosome synthesis in preparation of large cellular transitions such as between naive and primed

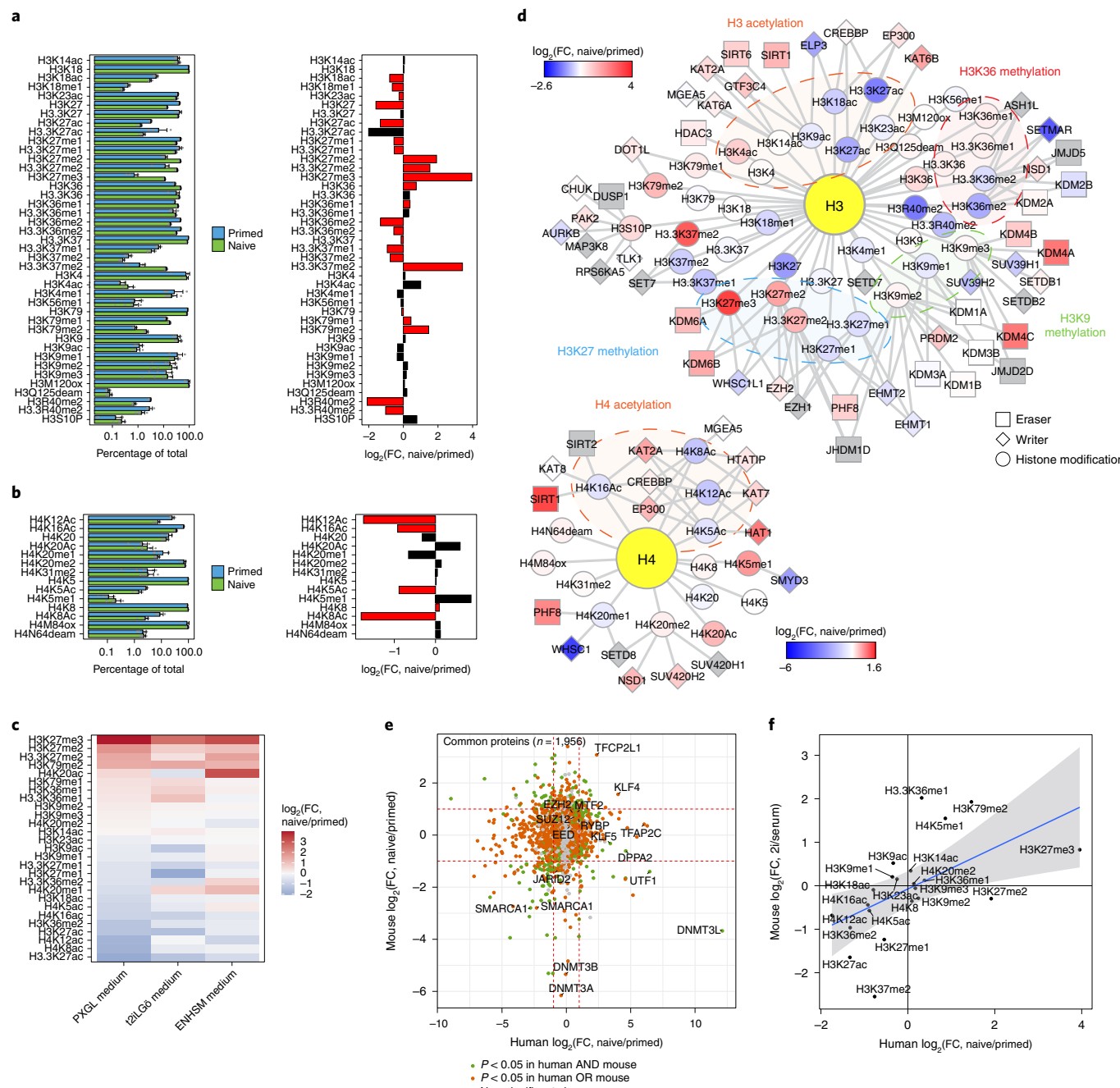

**Fig. 2 | Profiling of hPTMs reveals decoupling of chromatin-modifier activity and abundance when comparing naive and primed pluripotency. a,b**, Levels of H3 (**a**) and H4 (**b**) hPTMs in naive and primed hPSCs. The relative abundance of each hPTM as a percentage of the total for the histone residue (for example, the relative abundances of H3K79me1, H3K79me2 and H3K79 unmodified all add up to 100%) is provided (left). Unmodified histones are only shown for residues with >1 modification. Data are presented as the mean ± s.d. Change in hPTMs between naive and primed hPSCs as log$_2$-transformed FC values (right). The red bars indicate significantly changed hPTMs (two-sided Student's *t*-test with Benjamini–Hochberg correction, *P* < 0.05); *n* = 7 (naive hPSCs) and 5 (primed hPSCs) biologically independent samples. **c**, Comparison of hPTMs in naive and primed H9 hPSCs; the naive hPSCs were cultured in different media conditions. Only hPTMs identified in all datasets were retained. Data on naive hPSCs cultured in ENHSM medium were taken from[32]. **d**, Integration of the chromatin proteome and hPTM measurements for naive and primed hPSCs, separated by histone H3 and H4 modules. Nodes represent chromatin modifiers and hPTMs, and are coloured according to the log$_2$-transformed abundance FC. Edges indicate known functional connections (write or erase) between the nodes. Highlighted hubs indicate major hPTM groups. Chromatin modifiers in grey nodes were not detected. **e**, Comparison of the human and mouse chromatin proteomes of naive relative to primed pluripotent states. Only proteins identified in both human and mouse datasets were retained. Proteins with *P* < 0.05 were deemed as significantly changed (two-sided Student's *t*-test). Red dashed lines indicate FC = 2. Proteins referred to in the text as well as polycomb proteins are labeled. Mouse chromatin proteome data were obtained from[38]. **f**, Comparison of the human and mouse hPTMs in naive relative to primed pluripotent states. Only hPTMs identified in both human and mouse datasets were retained. The blue line indicates the best-fit linear regression; the shaded grey area indicates the 95% confidence interval. Mouse hPTM data were obtained from[38]. Source data are provided.

hPSCs. In line with this, we observed increased clipping at H3K27 in naive cells compared with primed cells (Extended Data Fig. 2e).

We next integrated the chromatin proteome with hPTM data by connecting histone marks with their respective writers and erasers (Fig. 2d). We found several surprising differences between hPTM and chromatin-mediator abundance. For instance, H3K27me3 was much higher in naive hPSCs but its major writer EZH2 was only slightly increased on chromatin compared with primed hPSCs, and the levels of the H3K27me3 erasers KDM6A and KDM6B were higher in naive hPSCs. These results suggest that the activity of PRC2 is also increased in naive cells relative to primed cells. In addition, we observed increased levels of the H3K9me2 and H3K9me3 erasers KDM4A, KDM4B and KDM4C but no change in those hPTMs (Fig. 2d). Conversely, increased DOT1L expression correlated well with the increase in H3K79me2, as does the reduced H3K36me2 level mediated by SETMAR. These results suggest that both the composition and activities of chromatin regulatory complexes change between naive and primed hPSC states.

In conclusion, naive and primed hPSCs have distinct chromatin landscapes with specific transcription factors as well as their own and shared chromatin complexes. Each state has its own unique hPTM signature, with naive hPSCs containing more H3K27me3 overall compared with primed cells. Surprisingly, the hPTM signature of each pluripotent state does not always directly correlate with the protein abundances of their writers and erasers on chromatin.

**Conserved and species-specific chromatin features.** To compare the chromatin-based properties of mouse and human pluripotent states, we integrated our dataset with a previous study of mouse PSCs[38]. Global analysis of chromatin-bound proteomes revealed that many naive and primed factors are similar between human and mouse. This includes transcription factors, such as KLF4 and TFCP2L1, that occur at higher levels in the naive state (Fig. 2e) and PRC2 core and sub-complex members, such as MTF2 and JARDI2. However, despite these similarities, several prominent proteins showed an opposite trend between mouse and human. Notable examples include KLF5, TFAP2C and DPPA2, which were strongly enriched on the chromatin of naive human cells compared with primed cells but not in mouse cells (Fig. 2e), which for TFAP2C is consistent with its human-specific role in early development[40]. LIN28B is mainly present in the chromatin of naive PSCs in humans, whereas in mice it is associated with primed pluripotency[50]. Other striking differences included UTF1 and DNMT3L, which were strongly enriched in human naive cells but showed the opposite trend in mouse pluripotent cells (Fig. 2e). In addition, DNMT3A and DNMT3B are strongly enriched on chromatin in primed mouse cells but were detected on chromatin at similar levels in naive and primed human cells (Fig. 2e). We also identified proteins that were detected uniquely in the ChEP proteomes of hPSCs

but not in mouse ChEP proteomes, which might therefore have human-specific roles in pluripotent cells (Extended Data Fig. 2f). Finally, hPTM patterns are largely conserved between human and mouse pluripotent states in naive and primed cells (Fig. 2f), as is H3K27 clipping (Extended Data Fig. 2e)[38,50].

**H3K27me3 marks lineage-determining genes in the naive state.** As H3K27me3 is associated with the control of gene regulation and cell identity and showed the largest difference between human pluripotent states, we investigated the genome-wide distribution of this chromatin mark in naive and primed hPSCs. We adapted the cleavage under targets and release using nuclease (CUT&RUN) method[51,52] by incorporating calibrated spike-in normalization (cCUT&RUN) to enable quantitative comparisons (Extended Data Fig. 3a,b and Supplementary Table 6). Consistent with our mass spectrometry results, cCUT&RUN confirmed there was a higher global level of H3K27me3 in naive compared with primed hPSCs (Fig. 3a). Several repetitive element classes also had higher levels of H3K27me3 in naive hPSCs (Fig. 3b), potentially also contributing to the pluripotent state-specific differences in H3K27me3 levels.

Contrary to the global trend, however, peak-based analysis revealed stronger and more focused regions of H3K27me3 enrichment in primed hPSCs compared with naive hPSCs (Fig. 3c,d and Extended Data Fig. 3c,d). Furthermore, a threefold-greater proportion of cCUT&RUN reads were within peaks in primed cells (Fig. 3e). We detected elevated levels of H3K27me3 in the regions surrounding peaks, providing further evidence that H3K27me3 coats the genome of naive hPSCs (Fig. 3d). These results show that although primed hPSCs have lower global H3K27me3 signal, the cCUT&RUN reads are more concentrated within defined and narrower peak regions.

As expected, a large proportion of the peaks in either cell type were near promoters and this proportion was higher in primed hPSCs (36%) compared with naive hPSCs (21%; Extended Data Fig. 3e). The reduced number of promoters marked by H3K27me3 in naive hPSCs is consistent with observations in naive hPSCs cultured in other media conditions[25,28,33] and similar to observations in mice[38]. However, based on our quantitative profiling, the number of H3K27me3-marked promoters in naive hPSCs is substantially higher than previously reported[28,33]. Differential analysis categorized peaks into regions enriched for H3K27me3 in either naive or primed hPSCs. We found that a subset of primed-enriched peaks marked naive-specific genes, including *KLF4* and *TFCP2L1* (Fig. 3f and Extended Data Fig. 3f). In addition, many primed-specific peaks were marked in both cell types but accumulated more H3K27me3 in the primed state (Fig. 3c), suggesting that regions marked by H3K27me3 in primed hPSCs are often already established in the naive state. Conversely, the naive-enriched regions were largely

**Fig. 3 | H3K27me3 localization, as determined by cCUT&RUN in naive and primed hPSCs. a,** Kernel density estimate of H3K27me3 cCUT&RUN reads in naive and primed hPSCs after normalization to the *Drosophila* spike-in. The genome was divided into 1-kb bins, the number of H3K27me3 reads in each bin was quantitated and the log$_2$-transformed value of the counts was calculated; $n=2$ biologically independent experiments for all samples (primed and naive H3K27me3 and IgG cCUT&RUN) excepting naive IgG, which is from $n=1$ experiment. **b,** Normalized H3K27me3 reads mapped at repetitive element classes in the human genome as a percentage of the total sequenced reads for naive and primed hPSCs. SINE and LINE, short and long interspersed nuclear elements, respectively; LTR, long terminal repeat. **c,** Heatmap of normalized H3K27me3 (left) and IgG (right) cCUT&RUN read counts within a 10-kb peak-centred window in naive and primed hPSCs. Regions were subsetted into primed-enriched ($n=5,086$ regions; top), common ($n=7,851$ regions; middle) and naive-enriched ($n=6,308$ regions; bottom) sites. **d,** Metaplots showing average profiles of normalized H3K27me3 counts across peaks, with relative abundance and distribution within 25 kb either side of the peak centre for primed-specific (middle), shared (right) and naive-specific (left) peaks. **e,** Percentage of normalized H3K27me3 reads within defined peaks for naive and primed hPSCs. **f,** Normalized H3K27me3 (top) and IgG (bottom) cCUT&RUN genome browser tracks over naive-specific (*DUSP6* and *SFRP2*; left) and primed-specific (*KLF4* and *TFCP2L1*; right) H3K27me3-marked genes. **g,** Normalized H3K27me3 (top) and IgG (bottom) cCUT&RUN genome browser tracks for exemplar trophoblast (*CDX2*, *GATA3*, *GATA2*, *KRT8* and *KRT18*; top), primitive endoderm (*GATA4*, *GATA6*, *PDGFRA* and *FOXA2*; middle) and additional alternative lineage marker genes (*HAND1*, *PAX6* and *SOX17*; bottom) in naive and primed hPSCs. Regions with $P<0.05$ after Benjamini–Hochberg multiple-testing correction were identified as differentially enriched. Source data are provided.

devoid of H3K27me3 in primed cells (Fig. 3c,f and Extended Data Fig. 3f). The gain and loss of H3K27me3 correspond to transcriptional differences between pluripotent states (Extended Data Fig. 3g)

and, overall, the presence of H3K27me3 at naive-specific genes was associated with reduced expression levels compared with primed cells (Extended Data Fig. 3h).

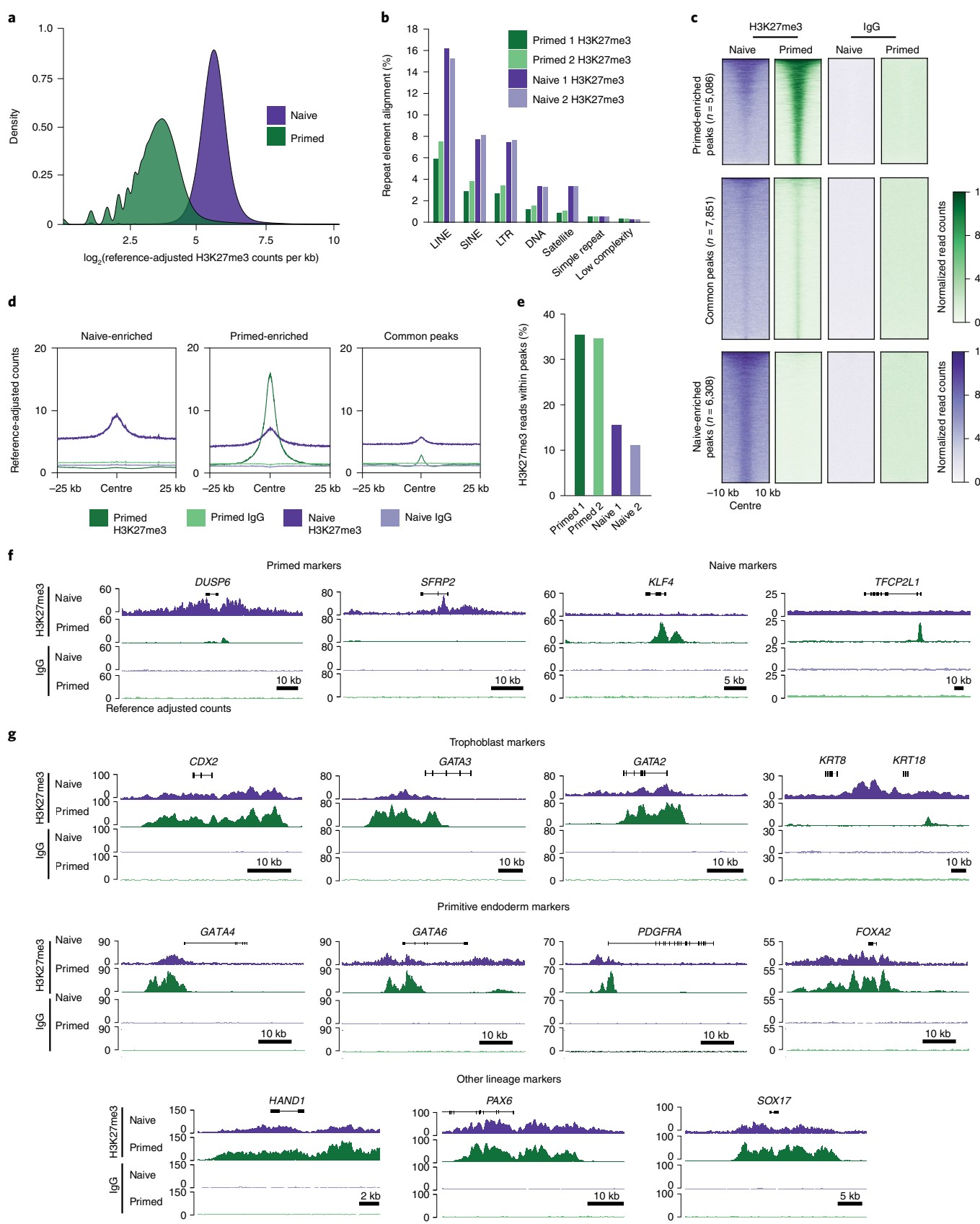

Many of the genes marked by H3K27me3 were shared between naive and primed cells ($n = 2,384$ genes; Extended Data Fig. 3f and Supplementary Table 7). Importantly, this category contained genes associated with embryonic- and extra-embryonic-lineage specification, which were unexpectedly marked by H3K27me3 in naive hPSCs as well as primed hPSCs (Fig. 3g). This gene set included germ-layer determinants including *PAX6*; primitive endoderm factors, such as *PDGFRA*, *GATA6* and *GATA4*; and trophoblast regulators such as *CDX2*, *GATA3*, *GATA2*, *KRT8* and *KRT18* (Fig. 3g). The unexpected presence of H3K27me3 at the promoters of key lineage regulators in naive hPSCs raises the possibility that PRC2-mediated H3K27me3 might oppose cell-fate specification in naive hPSCs. Several of the trophoblast factors marked by H3K27me3 in naive hPSCs are expressed at high levels in trophectoderm cells of human blastocysts and their enforced expression induces the trophoblast cell fate[4,53–55]. Consequently, because naive hPSCs have the capacity to produce trophoblasts in vitro[6,11–14], we sought to use trophoblast differentiation as a cell model to investigate a potential role for H3K27me3 in controlling lineage induction in human naive pluripotency.

**PRC2 activity opposes the induction of trophoblast fate.** To test the hypothesis that PRC2 activity in naive hPSCs restricts the induction of the trophoblast lineage, we established conditions that could rapidly deplete PRC2-mediated H3K27me3 in naive hPSCs, thereby limiting secondary effects or cell culture adaptations. Application of the PRC2 inhibitor UNC1999 (ref. [56]) in PXGL culture medium for 4 d robustly and reversibly depleted global H3K27me3 levels (Fig. 4a and Extended Data Fig. 4a–d) and removed H3K27me3 from gene promoters (Fig. 4b,c and Extended Data Fig. 4e). We observed minimal disruption to the chromatin-bound proteome (Fig. 4d and Extended Data Fig. 4c,d,f,g) or DNA methylation levels (Extended Data Fig. 4h). The hPTM landscape was somewhat more affected, as expected, due to H3K27me3 crosstalk with other hPTMs, such as H3K27ac and H3K36me2 (Fig. 4a,d and Extended Data Fig. 4b–d). There was no impact on cell viability or the expression of pluripotency-associated genes and cell-surface proteins (Extended Data Fig. 4i-k). Furthermore, very few transcriptional changes were detected (Fig. 4d–g and Extended Data Fig. 4j), as expected for this short-term inhibitor treatment in naive cell media. However, we did detect upregulation of the trophoblast-associated genes *IGF2*, *SOCS3*, *SATB1*, *SATB2* and *SOX21* (Fig. 4e,f and Supplementary Table 8)[53,57–61] as well as established polycomb targets, such as *HOXD13* and *CCND2* (Fig. 4f and Extended Data Fig. 4k).

Given our ability to acutely deplete H3K27me3, we used defined culture conditions to convert naive hPSCs into trophoblast cells[11,13] to test whether PRC2 inhibition affected trophoblast-fate induction. In this assay we applied the PRC2 inhibitor for 4 d before trophoblast conversion (days −4 to 0 in PXGL medium), during trophoblast conversion (days 0 to 4 in trophoblast stem cell culture conditions, ASECRiAV[62]) or throughout the whole experiment (a total of 8 d) starting with naive hPSCs (Fig. 5a). As before, in the absence of trophoblast-induction cues most trophoblast genes remained largely transcriptionally repressed following four or more days of PRC2 inhibition (Fig. 5b, Extended Data Fig. 5a and Supplementary Table 8). In addition, very few GATA3+ trophoblast cells were detected in naive hPSC cultures maintained in PXGL medium and there was no difference between the control and PRC2 inhibitor-treated samples (Extended Data Fig. 5b–d). As expected, when exposed to conditions that promote trophoblast conversion, the number of undifferentiated NANOG+ nuclei decreased after 4 d (Extended Data Fig. 5e,f). No difference in cell viability between the PRC2 inhibitor-treated and control cells was observed during trophoblast conversion (Extended Data Fig. 5g). Cells acquired a morphology resembling trophoblast cells (Extended Data Fig. 5f) and activated the expression of the trophoblast genes *GATA2*, *GATA3* and *KRT7* as well as *VGLL1*, which marks the cytotrophoblast and mature extravillous trophoblast[63], collectively indicating robust trophoblast-fate induction (Fig. 5b and Extended Data Fig. 5h). Importantly, PRC2 inhibition increased the number of GATA3+ nuclei during trophoblast conversion compared with controls (Fig. 5c and Extended Data Fig. 5a,i). The same result was obtained when using an alternative hPSC line (Extended Data Fig. 5j). In addition, PRC2 inhibition accelerated the exit from naive pluripotency, as shown by the stronger decrease of naive markers following PRC2 inhibition (Extended Data Fig. 5k). Finally, previous studies have proposed that trophoblast induction is much less efficient when starting from primed hPSCs compared with naive hPSCs[6,11–13]. This reduced efficiency could not be overcome by PRC2 inhibition on day 4 of conversion in trophoblast conditions (Extended Data Fig. 5l). Collectively, these results suggest that PRC2 stabilizes the chromatin and transcriptional states of naive hPSCs and limits the induction of trophoblast differentiation.

**Single-cell transcriptomes of trophoblast-fate induction.** To determine the effects of PRC2 inhibition on trophoblast induction with increased resolution, we carried out single-cell RNA sequencing (scRNA-seq) on days 0 and 4 of trophoblast induction from naive hPSCs under both control and PRC2-inhibition conditions. Uniform manifold approximation and projection (UMAP) analysis separated the day 0 and day 4 samples (Fig. 5d,e). Graph-based clustering defined three main clusters—that is, one large cluster that corresponds to day 0 naive samples (cluster 0, C0) and two separated clusters (clusters 1 and 2, C1 and C2) overlapping the day 4 samples (Fig. 5f). As expected, naive and core pluripotency markers were expressed in C0 cells (Fig. 5g,h and Extended Data

**Fig. 4 | Histone, chromatin and transcriptional responses following short-term acute PRC2 inhibition. a**, Levels of H3 hPTMs in naive and primed hPSCs with and without PRC2 activity inhibition for 4 d (PRC2i). Data are presented as the log₂-transformed FC between the two conditions indicated above each panel. Data are ordered according to the left panel. The red bars indicate significantly changed hPTMs (two-sided Student's *t*-test with Benjamini–Hochberg correction, $P < 0.05$; $n = 7$ (naive hPSCs), 5 (primed hPSCs), 6 (naive hPSCs + PRC2i) and 8 (primed hPSCs + PRC2i) biologically independent samples). **b**, Heatmaps of normalized H3K27me3 cCUT&RUN read counts within a 10-kb peak-centred window in naive and primed hPSCs with and without PRC2i. Regions were subsetted into primed-enriched ($n = 5,086$ regions), common ($n = 7,851$ regions) and naive-enriched ($n = 6,308$ regions) sites; $n = 2$ biologically independent experiments for all samples (primed and naive H3K27me3 and IgG cCUT&RUNs, both with and without PRC2i) excepting naive IgG, which is from $n = 1$ experiment. Samples without PRC2i treatment are reproduced from Fig. 3. **c**, Genome browser tracks show normalized H3K27me3 and IgG cCUT&RUN reads for trophoblast genes (*CDX2*, *GATA2*, *GATA3*, and *KRT8* and *KRT18*) in naive and primed hPSCs with and without PRC2i. **d**, Principal component (PC) analysis of the chromatin proteome (left), hPTM landscape (middle) and transcriptome (right) of naive and primed hPSCs with and without PRC2i ($n = 3$ biologically independent samples for chromatin proteome and transcriptomes). **e**, Gene expression levels, determined through RNA-seq analysis, of trophoblast-associated genes (*IGF2*, *SOCS3*, *SATB1*, *SATB2* and *SOX21*) in naive hPSCs with and without PRC2i ($n = 3$ biologically independent samples). Data are presented as the mean ± s.d. **f,g**, Differential gene expression in naive (**f**) and primed (**g**) hPSCs with and without PRC2i ($n = 3$ biologically independent samples). Genes enriched in the untreated condition are highlighted in red and those enriched after PRC2i are highlighted in blue; the number of differentially expressed genes in both conditions are indicated. Dashed lines indicate $P < 0.05$ and log₂(FC) > 1 (two-sided Student's *t*-test). Source data are provided.

Fig. 6a–c). Cells in the C1 'intermediate' cell cluster had reduced expression levels of pluripotency genes and increased expression of *KRT18*, *TFAP2C* (Fig. 5g), *ARID3A* and *EPCAM* (Extended Data

Fig. 6b,c). Most of the day 4 conversion samples, both with and without PRC2-inhibitor treatment (74 and 88% of cells, respectively), contributed to the intermediate cluster (C1; Extended

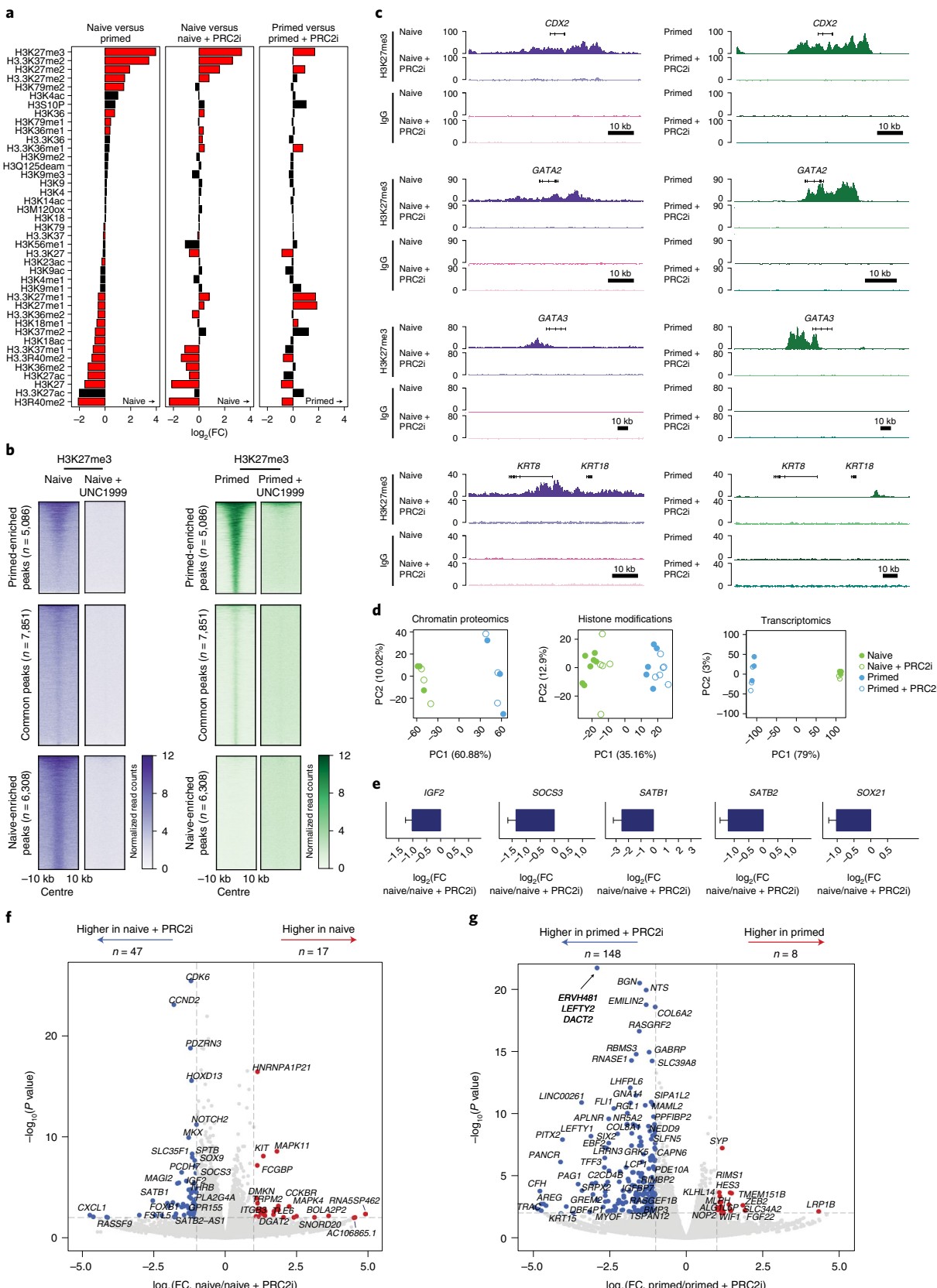

Data Fig. 6d and Supplementary Table 9). The C2 cells showed a strong decrease in expression of pluripotency genes and the activation of multiple trophoblast markers[4,5] (Fig. 5g,h and Extended Data Fig. 6b,c). We termed cells in the C2 cluster as trophoblast cells because they aligned to human embryo trophectoderm and early trophoblast (Fig. 6a–d and Extended Data Fig. 6e,f). Importantly, PRC2 inhibition promoted the acquisition of trophoblast fate, as the proportion of cells in the C2 trophoblast cluster more than doubled following PRC2 inhibition (26% for the PRC2-inhibited cells versus 11% for the control cells; Fig. 6e and Supplementary Table 9). Together, these results show that PRC2 inhibition promotes naive pluripotency exit and increases trophoblast-fate induction.

**PRC2 inhibition enhances trophoblast formation in blastoids.** To further investigate a role for PRC2 in human trophoblast specification and morphogenesis, we used human blastoids as a three-dimensional blastocyst-like model[19] (Fig. 7a). We tested whether PRC2 inhibition affects trophoblast specification and epithelial morphogenesis, processes that are necessary to form a blastocoel-like cavity. We inhibited PRC2 in naive hPSCs for 4 d before blastoid formation and assessed the effect of PRC2 inhibition on trophoblast-fate induction by measuring the proportion of trophoblast cells in human blastoids using the pan-trophoblast markers TROP2 and GATA3. PRC2 inhibition increased the proportion of TROP2[+] and GATA3[+] cells in blastoids at 36 h and 60 h (Fig. 7b,c and Extended Data Fig. 7a–d). The increase in trophoblast induction was accompanied by a decrease in the ratio of NANOG[+] epiblast-like cells (Fig. 7c and Extended Data Fig. 7a,b,d). Although there were very few primitive endoderm-like cells (FOXA2[+]) at 60 h, there was a trend towards increased primitive endoderm induction following PRC2 inhibition (Extended Data Fig. 7c).

During blastocyst development the trophoblast forms an epithelium that pumps water directionally to generate the blastocoel cavity. PRC2 inhibition accelerated cavity formation by about 24 h (Fig. 7d–f and Supplementary Table 10). Multiple cavities seemed to progressively coalesce, possibly through the action of aquaporin 3, the water transporter most highly expressed in human blastocysts[5]. This effect of the PRC2 inhibitor was no longer observed at 60 h, as blastoids had already reached their maximum size. During human blastocyst and blastoid development aquaporin 3 is initially expressed in all cells and then becomes restricted to trophoblast cells[5,19]. This restriction seemed to occur more rapidly with PRC2 inhibition (Extended Data Fig. 7e). These observations are consistent with an acceleration of trophoblast-fate induction, as observed in the earlier monolayer experiments (Fig. 5). The results show that PRC2 inhibition promotes the formation of functional trophoblast with some trophectoderm-like morphogenetic properties. Together, we conclude that PRC2 functions as a barrier to trophoblast formation in naive hPSCs (Fig. 7g).

## Discussion
Human naive PSCs and epiblast cells have the potential to generate trophoblast with high efficiency in response to inductive cues[6,11–14]. The molecular properties that enable this highly regulated developmental plasticity, however, have not been comprehensively defined. Here we have demonstrated that repressive chromatin pathways oppose trophoblast induction in naive hPSCs. We showed that PRC2-mediated H3K27me3 marks trophoblast regulators in naive hPSCs, including genes that are expressed in trophectoderm cells of human blastocysts and can promote trophoblast fate[4,53–55]. By establishing that PRC2 is a lineage gatekeeper stabilizing the undifferentiated naive state, these findings overturn the current assumption that naive hPSCs are epigenetically unrestricted. Protection of trophoblast genes against low-level or inappropriate transcriptional activation signals is anticipated to support robust growth of undifferentiated naive hPSCs while maintaining their broad developmental potential. Sustained exposure to strong trophoblast-inductive signals overcomes these repressive mechanisms and efficient trophoblast differentiation is initiated. By uncovering a role for this pathway in opposing trophoblast induction and finding that in naive hPSCs H3K27me3 also marks key regulators of additional cell types, such as primed pluripotency and primitive endoderm, our work lays the foundation for future studies to determine whether PRC2 could also control the specification of alternative lineages from naive pluripotency.

We have also shown that human naive and primed pluripotent cells have striking differences in the relative abundance and activities of chromatin-associated proteins. Integrating these datasets enabled a systems-level view of the chromatin proteomes and revealed that state-specific differences in the abundance of chromatin modifiers and their associated histone marks are not always concordant. These findings highlight the importance of regulating protein activities in addition to protein abundance in changing chromatin states in pluripotency and thereby raise caution in using methods like transcriptional or proteomic profiling to predict differences in chromatin states between cell types. Our work also resolved discrepancies in the literature. Different methodological approaches have previously reported conflicting results on the global level of H3K27me3 in naive hPSCs[31,32]. Our findings establish that global H3K27me3 levels are increased in naive compared with primed hPSCs and in multiple culture conditions, which is consistent with a previous report in hPSCs and with mouse pluripotent states[32,38]. These findings are in line with a possible need for high polycomb-protein activity in cells, such as naive hPSCs, to retain low levels of global DNA methylation[38]. More generally, because the changes in the relative abundance of most histone modifications were similar when human and mouse pluripotent states were compared, our results also suggest a general conservation of the histone code between human and mouse in these cell types. This raises the prospect of applying histone profiling to define mammalian PSC states. However, despite these general similarities, we also uncovered

---

**Fig. 5 | PRC2 inhibition promotes naive hPSC-to-trophoblast fate induction. a**, Schematic of the experimental design used to study the role of PRC2 and H3K27me3 in the conversion of naive human induced PSCs (hiPSCs) to trophoblasts. Inhibition of PRC2 was applied for 4 d before, during or throughout (before and during) trophoblast conversion. Created with BioRender.com. **b**, Levels of expression of the trophoblast marker genes *GATA3*, *GATA2* and *KRT7*, determined using quantitative PCR with reverse transcription. The expression values were normalized to *GAPDH*; experiments are shown as individual data points (squares, triangles and circles; n = 3 biologically independent samples); a.u., arbitrary unit. Two-sided Student's *t*-test with Bonferroni adjustment; *P < 0.05, **P < 0.01 and ****P < 0.0001. **c**, Levels of GATA3[+] and NANOG[+] nuclei, determined from immunofluorescence microscopy images (see Extended Data Fig. 5a), on day 4 of naive-to-trophoblast conversion; n = 2 biologically independent samples. **d,e**, scRNA-seq analysis. **d**, UMAP of single-cell transcriptomes coloured according to sample. Each dot represents a cell (n = 7,629 cells). **e**, UMAPs of the four treatment combinations shown separately. Grey dots indicate cells not belonging to the highlighted treatment. **f**, UMAP of single-cell transcriptomes coloured according to the cell clusters (C0–C3). Each dot represents a cell. **g**, Analysis (scRNA-seq) of pluripotency and trophoblast marker genes. Each dot represents a cell. Data are log-transformed normalized counts of gene expression. **h**, Expression of cell type-specific marker genes in two cell clusters (C1 and C2) with and without PRC2i, and in human embryo (epiblast, trophoblast and primitive endoderm) data from[3,74]. The size of the circles represents the proportion of cells in the cluster with the indicated gene expression enrichment. **a,b,d,e**, D, day. Source data are provided.

species-specific differences, particularly at the level of chromatin mediators. For example, the DNA methylation machinery seems to operate differently in human and mouse naive PSCs, with major state-specific differences in chromatin association of DNMT3A, DNMT3B and DNMT3L, which show opposite trends when human and mouse cells are compared. Curiously, the catalytically

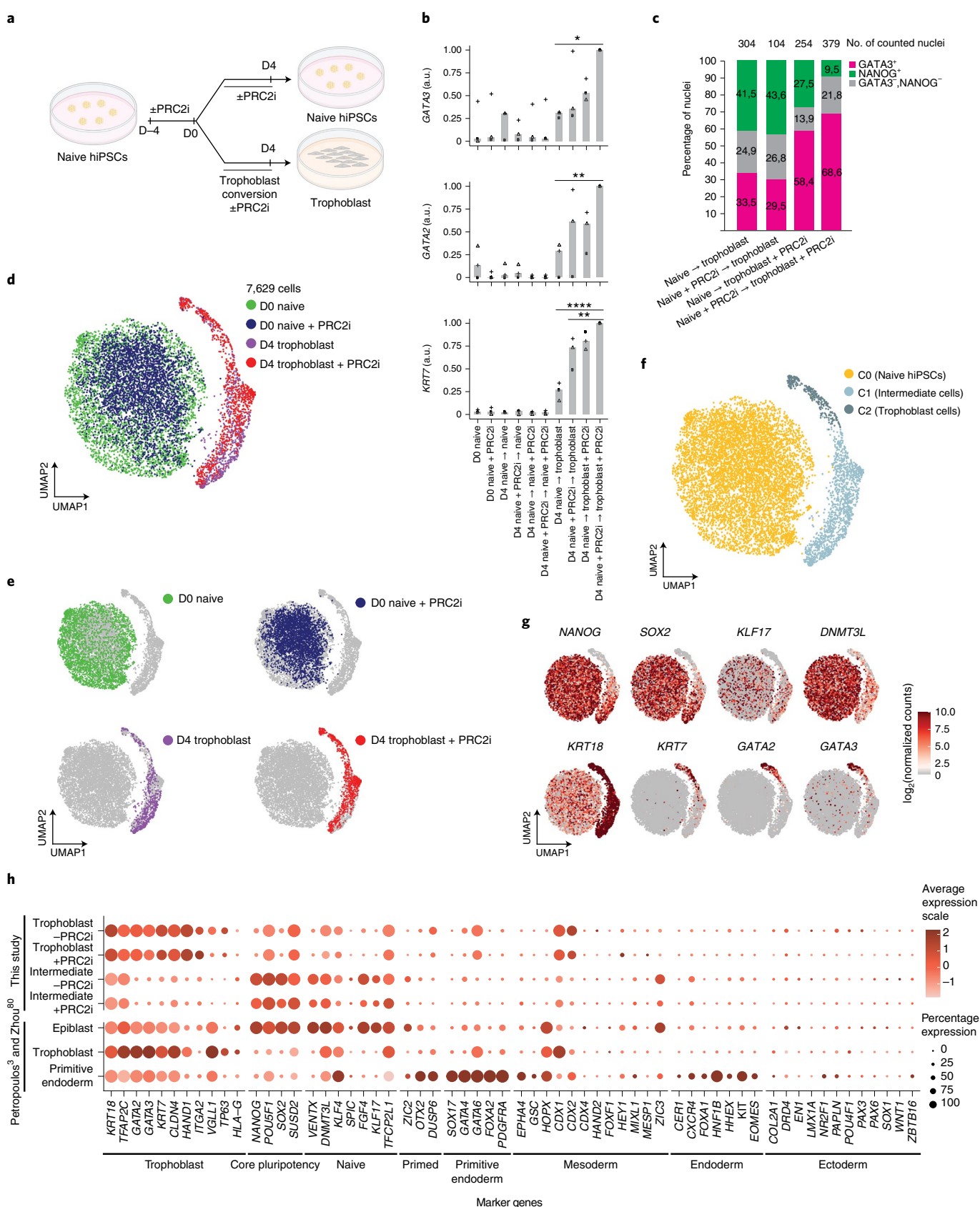

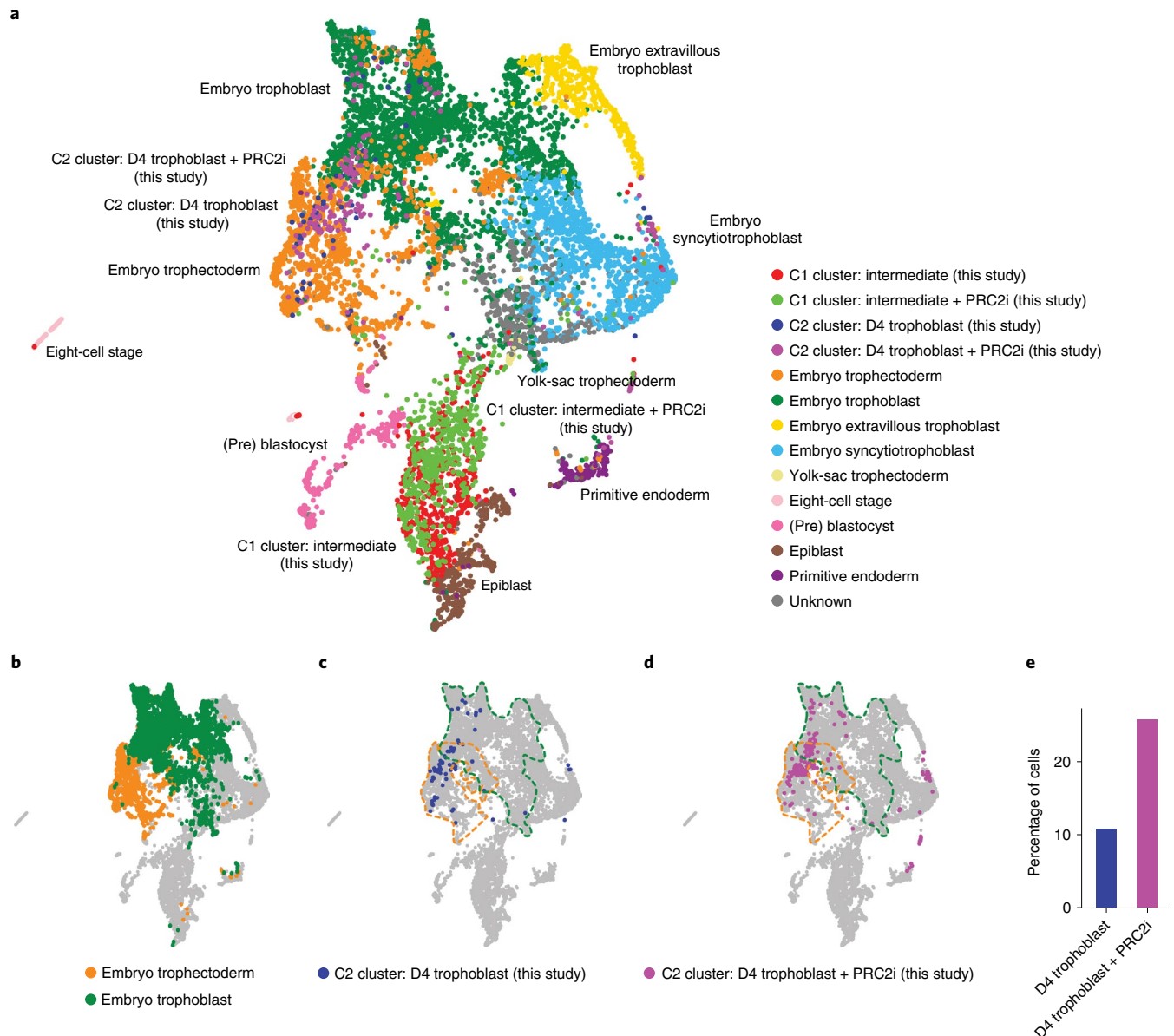

**Fig. 6 | Evaluation of differentiation by comparison with human embryo data. a**, UMAP projection of the human pre-implantation and postimplantation embryo integration with day 4 (D4) in vitro trophoblast conversion with and without PRC2i. Human embryo data from[3,74]. **b**, UMAP projection of embryo trophectoderm and embryo trophoblast on the UMAP from **a**. **c,d**, UMAP projection of D4 trophoblast cells in C2 with (**d**) and without (**c**) PRC2i treatment projected on the UMAP from **a**. The clusters correspond to those in Fig. 5d–f. Dotted lines represent the embryo trophectoderm (orange) and embryo trophoblast (green) as shown **b**. **e**, Proportion of D4 trophoblast conversion cells, with and without PRC2i, that were categorized as belonging to the C2 trophoblast cluster. Source data are provided.

inactive protein DNMT3L is strongly upregulated in naive hPSCs and its role in global hypomethylation is not intuitive as it might be expected to boost de novo methyltransferase activity by stimulating DNMT3A and DNMT3B. However, knockdown of *DNMT3L* during primed-to-naive hPSC resetting does not affect the levels of DNA methylation[64], and it is possible that DNMT3L might have roles in human naive pluripotency that are methylation-independent. It is of particular interest to establish whether DNMT3L might recruit chromatin-modifying repressor proteins to silence transposable elements and other target regions, as has been recently reported in mouse PSCs and fibroblasts[65,66].

The state-specific global differences in H3K27me3 prompted us to examine this modification in further detail. Using a quantitative CUT&RUN assay, we found that the levels of H3K27me3 in the

genome of naive hPSCs were substantially higher than previously shown[28,33], corroborating the global H3K27me3 quantification of our hPTM profiling. Importantly, the promoters of developmental regulators of multiple lineages are marked by H3K27me3 in naive hPSCs, thereby uncovering a more prominent role for H3K27me3 in these cells than recognized thus far[28,33]. This finding builds on our recent study showing that H3K27me3 tends to co-occur in naive hPSCs with the active histone marks H3K4me3 and H3K4me1 (refs. [25,26]), which is a signature of bivalent chromatin[67,68]. Whether these regions in naive hPSCs have other molecular hallmarks of bivalent chromatin is important to establish. In primed human and mouse PSCs, regions containing bivalent chromatin are connected through long-range chromatin interactions that are thought to constrain and coordinate transcriptional regulation[25,69–71]. In contrast, naive

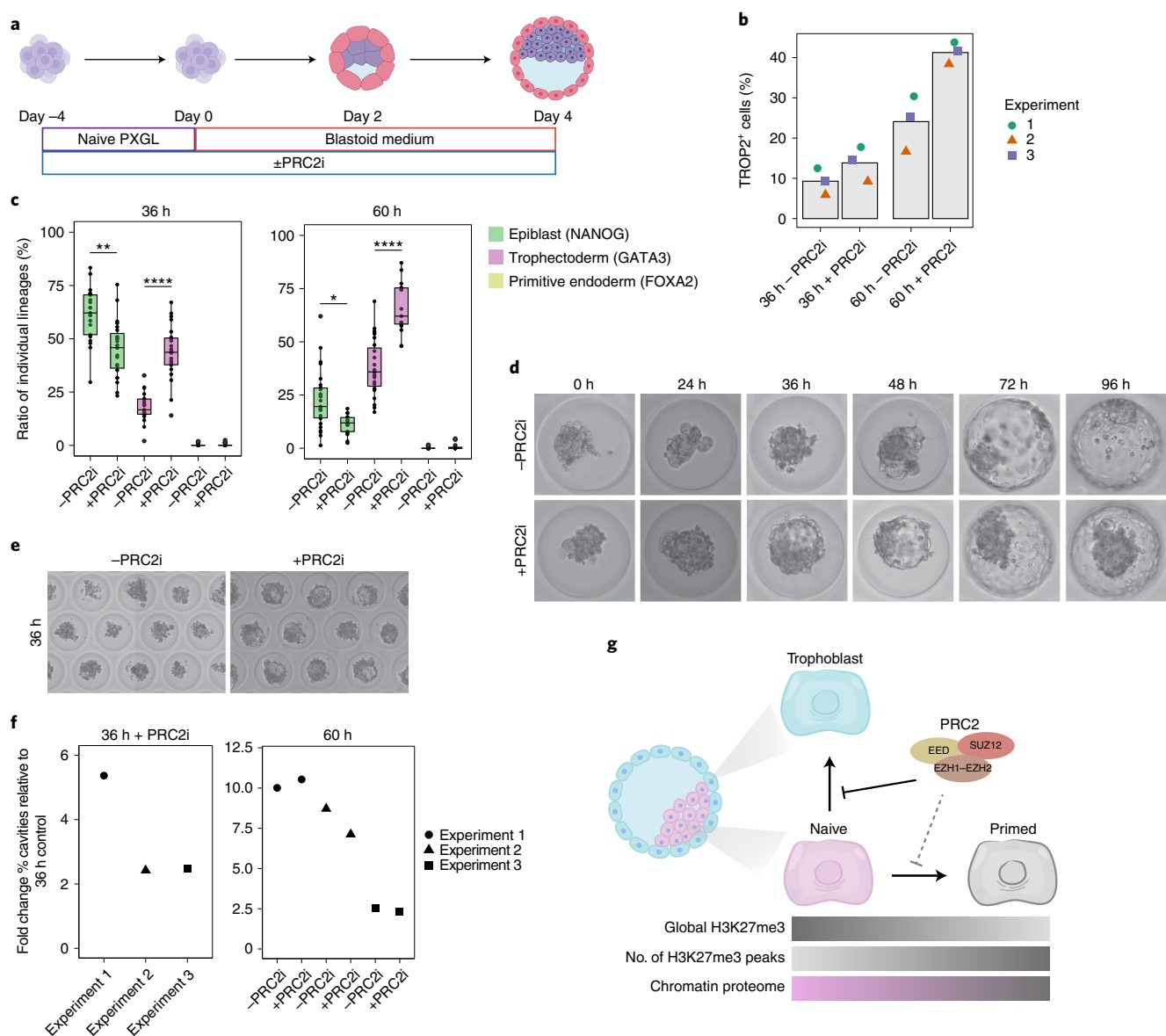

**Fig. 7 | PRC2 inhibition accelerates trophoblast development and cavity formation in human blastoids. a**, Schematic of the experimental set-up for studying the role of PRC2 in trophoblast formation in human blastoids. Blastoids are formed by aggregating naive hPSCs in microwells[19]. Created with BioRender.com. **b**, Proportion of TROP2+ trophoblast cells in human blastoids at 36 h and 60 h with and without PRC2i; $n = 3$ biologically independent samples. **c**, NANOG, GATA3 and FOXA2 expression in 36 h (left) and 60 h (right) blastoids with or without PRC2i, quantified from immunofluorescence images (Extended Data Fig. 7a–d). The boxplots show the interquartile range (box limits showing the 25th and 75th percentile) and median (centre line) of the ratio of cells belonging to individual lineages, represented as percentages of the total number of cells per blastoid. Whiskers indicate 1.5x the interquartile range; $n = 21$ (36 h without PRC2i (−PRC2i)), 23 (36 h + PRC2i), 27 (60 h − PRC2i blastoids) and 17 (60 h + PRC2i) blastoids were quantified from a single experiment. Two-sided Wilcoxon rank-sum test with Bonferroni correction; 36 h, **$P = 3.7 \times 10^{-3}$ and ****$P = 1.1 \times 10^{-7}$; 60 h, *$P = 1.1 \times 10^{-2}$ and ****$P = 2.5 \times 10^{-7}$. **d,e**, Bright-field images showing accelerated cavity formation during human blastoid formation (0–96 h) following PRC2i treatment (**d**) and at 36 h of human blastoid formation following PRC2i treatment (**e**). **f**, Fold change in cavitated human blastoids after 36 h with PRC2i (left) and 60 h with or without PRC2i (right). Data are shown as the FC normalized to 36 h without PRC2i; $n = 3$ biologically independent samples. **g**, Model showing that PRC2 restricts the induction of trophoblast fate from naive hPSCs. For color bars, darker colors indicate higher levels, except for the chromatin proteome, where pink represents naive chromatin proteome and grey represents primed chromatin proteome. Our findings establish that PRC2 acts as a barrier to lineage specification in naive hPSCs, opposing the formation of trophoblast cells in the presence of differentiation cues. In addition, our results uncover a potential role for PRC2 to safeguard the naive epigenome against adopting features of primed pluripotency, similar to observations in mice[38]. PRC2 activity establishes a higher global level of H3K27me3 in naive hPSCs compared with primed hPSCs, whereas the number of defined H3K27me3 peaks shows the opposite pattern. Our study also defined distinct chromatin proteomes that differ between naive and primed pluripotent states. Source data are provided.

human and mouse PSCs lack long-range connections between bivalent chromatin sites, suggesting that although developmental genes are marked by H3K27me3 in naive cells, the mode of regulation might differ[25,70].

Following the unexpected discovery of H3K27me3 at trophoblast-associated genes in naive hPSCs, we hypothesized that this repressive modification might functionally oppose the induction of trophoblast cell identity. We tested this prediction using

two different cellular models of naive-to-trophoblast specification and found that the acute inhibition of PRC2 activity promoted trophoblast-fate induction. Curiously, a recent study reported that PRC2-deficient primed hPSCs upregulate GATA3 and KRT7 when transferred into trophoblast stem cell medium[72]. However, the low efficiency and prolonged timing of these events suggest that the mechanisms and developmental relevance when starting from a primed state are distinct from the naive-to-trophoblast transition that we uncover here[14]. Our experiments showed that PRC2 inhibition was indeed not sufficient to increase the efficiency of trophoblast-fate induction in primed hPSCs to levels that are comparable to naive hPSCs.

Many of our conclusions are in line with another study published in this issue[73]. One of the few differences between the two studies relates to whether PRC2 inhibition causes naive hPSCs to differentiate in self-renewing conditions. We found there is no miscellaneous differentiation of naive hPSCs following PRC2 inhibition with UNC1999 in PXGL medium, whereas Kumar et al.[73] report significant levels of differentiation of naive hPSCs following PRC2 inhibition with EPZ-6438 in t2iLGö media. We tested whether this difference could be due to the different inhibitors used. Naive hPSCs treated with EPZ-6438 in PXGL medium also showed no change in naive hPSC differentiation (Extended Data Fig. 7g). We believe this difference can instead be attributed to the different naive hPSC media used, which alter the permissiveness of naive hPSCs to induce differentiation. In particular, PXGL medium contains a WNT antagonist (XAV939) whereas t2iLGö medium contains a WNT activator (CHIR99021). Shielding from WNT stimulation protects naive hPSCs against the induction of differentiation-associated genes[9].

We also examined the role of PRC2 in a second model of human trophoblast development. Our results in blastoids showed that PRC2 inhibition accelerates trophoblast induction and the appearance of a blastocoel-like cavity during blastoid formation. These findings raise the possibility that the controlled inhibition of PRC2 could be a way to improve the timing and efficiency of blastoid formation. Furthermore, because extended developmental plasticity is also a property of epiblast cells in human pre-implantation embryos[6], our results raise the possibility that PRC2 might also fulfil a similar role in human development. Whether PRC2 opposes trophoblast specification in human embryos is an exciting line of future research with important implications for understanding the causes of infertility and developmental disorders.

## Online content

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

## Methods

Our research complies with all relevant ethical regulations and guidelines. Experiments with hPSCs were approved by the UZ/KU Leuven Ethics Committee (S52426, S66185 and S64962), the Flemish Government (SBB 219 2020/0435) and the Steering Committee of the UK Stem Cell Bank (SCSC11-58). The WiCell line H9 (WA09) was used under the agreements 20-WO-341, 12-WO-202 and 18-WO-026. Blastoid generation was approved by the Commission for Science Ethics of the Austrian Academy of Sciences and the KU/UZ Leuven Ethics Committee (S66185 and S64962). This work did not exceed a developmental stage normally associated with 14 consecutive days in culture after fertilization. The animal work carried out in this study was covered by project licences (ECD_P003-2016 and ECD_P170/2019 to V.P. and to F.L., respectively) approved by the KU Leuven Animal Ethics Committee.

**Human cell lines.** Experiments were carried out using the following cell lines: H9 hESCs (obtained from WiCell) and ICSIG-1 IPSC0028 hiPSCs (obtained from Sigma-Aldrich). The H9 hESC line chemically reset to the naive state was provided by A. Smith[75] (with permission from WiCell) and was used for all experiments in Figs. 1–4,7 and Extended Data Figs. 1–3,4a–j. Other naive hPSC lines (H9 and IPSC0028) were newly reset to the naive state in the Pasque laboratory: chemically reset hiPSCs were used in Figs. 5,6 and Extended Data Figs. 4k, 5a–i,k, 7g, and newly chemically reset H9 hESCs were used in Extended Data Fig. 5j. The resetting protocol used is described below. Primed H9 hESCs were used in all experiments with the exception of Extended Data Fig. 5l, where primed IPSC0028 hiPSCs were used. None of the cell lines are on the Register of Misidentified Cell Lines. All cell lines used in this study were authenticated by RNA and protein expression analysis and were also confirmed to be mycoplasma-negative by PCR test.

**Primed hPSC culture.** Primed hPSCs were cultured under humidified conditions at 37 °C in an incubator with 5% $O_2$ and 5% $CO_2$. Primed H9 hPSCs were cultured in feeder-free conditions on plates precoated with 0.5 µg cm$^{-2}$ vitronectin (Thermo Fisher Scientific) in complete TeSR-E8 medium (Stem Cell Technologies). The cells were passaged using an incubation of 5 min with 0.5 mM EDTA in PBS at room temperature. UNC1999 (Abcam, ab146152) was applied at 2.5 µM for 4 d on the day after passaging; the medium was changed daily. Culture conditions for the primed-to-trophoblast conversion are described in the 'Primed hPSC-to-trophoblast fate conversion' section.

**Naive hPSC culture.** Naive hPSCs were cultured under humidified conditions at 37 °C in an incubator with 5% $O_2$ and 5% $CO_2$. Naive H9 hPSCs were cultured in feeder-free conditions using Geltrex (Thermo Fisher Scientific), diluted 1:300 in fresh medium; the ICSIG-1 and naive H9 hPSCs from Figs. 5,6 and Extended Data Figs. 5,6 were cultured on mitotically inactivated mouse embryonic fibroblasts (MEFs). Naive hPSCs were cultured in PXGL medium[9,76], consisting of a 1:1 mixture of DMEM/F12 and Neurobasal media supplemented with 0.5% N2 supplement, 1% B27 supplement, 2 mM L-glutamine, 0.1 mM β-mercaptoethanol and 1×penicillin–streptomycin (all from Gibco, Thermo Fisher Scientific) as well as 1 µM PD0325901 (Axon Medchem and Wellcome–MRC Cambridge Stem Cell Institute), 2 µM XAV939 (Sigma), 2 µM Gö6983 (Tocris), 20 ng ml$^{-1}$ human LIF (PeproTech and Wellcome–MRC Cambridge Stem Cell Institute) and 10 µM Y-27632 (Tocris and Cell Guidance Systems). The medium was changed daily, with freshly added XAV939 and Y-27632. Naive hPSCs were routinely passaged every 4 d at a ratio of 1:2 by single-cell dissociation with accutase (BioLegend), followed by filtering through a 40-µm cell strainer (Corning). Where indicated, some experiments were performed on naive H9 hPSCs cultured in t2iLGö medium[8] in a 1:1 mixture of DMEM/F12 and Neurobasal media supplemented with 0.5% N2 supplement, 0.5% B27 supplement, 2 mM L-glutamine, 50 U ml$^{-1}$ penicillin–streptomycin and 0.1 mM β-mercaptoethanol (all from Thermo Fisher Scientific); 1 µM PD0325901, 1 µM CHIR99021 and 20 ng ml$^{-1}$ human LIF (all from Wellcome–MRC Cambridge Stem Cell Institute); and 2 µM Gö6983 (Tocris) on Matrigel–coated plates (Corning). To inhibit PRC2, 1 µM UNC1999, or an equivalent volume of dimethylsulfoxide (DMSO) as a control, was freshly added to the medium. UNC1999 was used in all inhibitor experiments except in the inhibitor comparison experiment of Extended Data Fig. 7g, where EPZ-6438 (10 µM; EZSolution, Biovision, 2824-5) was also used. Here, naive hPSCs were cultured on feeders in PXGL supplemented with a PRC2 inhibitor (UNC1999 or EPZ-6438) for 7 d. The PXGL medium was changed daily. The cells were passaged 3 d before (day −3) treatment with PRC2 inhibitor as well as on days 0 and 4 of the PRC2-inhibitor treatment. Medium containing freshly added PRC2 inhibitors, UNC1999 (1 µM final concentration) or EPZ-6438 (10 µM final concentration), was added (1 µl ml$^{-1}$) to the media daily; 1 µl ml$^{-1}$ medium with DMSO was used as a control.

**Cell culture of MEFs.** Mouse embryonic fibroblasts were isolated from embryonic-day-14.5 male mouse embryos derived from WT C57B6/J mice (*Mus musculus musculus*, KU Leuven Animal Core Facility) and immortalized with mitomycin C (Bioconnect). The MEFs were cultured in filter-sterilized MEF medium—consisting of approximately 90% (vol/vol) DMEM medium (Thermo Fisher Scientific) supplemented with 10% fetal bovine serum (FBS; Gibco), 1×penicillin–streptomycin (Gibco), 1% Glutamax (Gibco), 1×non-essential amino acids (Gibco) and 0.1 mM β-mercaptoethanol (Gibco)—on 0.1% gelatin-coated plates. One day before use, the MEF feeders were plated and maintained at 37 °C under normoxic conditions (20% $O_2$ and 5% $CO_2$).

***Drosophila melanogaster* cells.** *Drosophila* S2 cells (obtained from Thermo Fisher Scientific) used for the cCUT&RUN were cultured in a non-humidified incubator at 28 °C without additional $CO_2$, in normoxic conditions. The *Drosophila* S2 cells were cultured in T75 flasks in Schneider's *Drosophila* medium (Thermo Fisher Scientific) supplemented with 10% heat-inactivated FBS (Sigma). The cells grew in a semi-adherent monolayer and were passaged by gently tapping the flasks and washing gently with medium, pipetting up and down to break up clumps.

**Primed-to-naive hPSC conversion.** Starting on day 1 or 2 after seeding primed IPSC0028 ICSIG-1 (Sigma) hiPSCs in E8Flex medium on Geltrex, the cells were switched to cRM-1 medium and moved to hypoxia[75]. The cRM-1 medium was comprised of N2B27 medium (50% (vol/vol) DMEM/F12 medium (Gibco, 31330-038), 50% Neurobasal medium (Gibco, 21103-049), 2 mM L-glutamine (Gibco, 25030-081), 0.5% N2 supplement (Gibco, 17502-048), 1% B27 supplement (Gibco, 17504-044), 1% penicillin–streptomycin (Gibco, 15140122) and 0.1 mM β-mercaptoethanol (Gibco, 31350010)) supplemented with 1 µM PD0325901 (Axon Medchem, 1408), 10 ng ml$^{-1}$ recombinant human LIF (PeproTech, 300-05) and 1 mM valproic acid (Sigma-Aldrich, V0033000). After 3 d in cRM-1 medium, the cells were switched to cRM-2 medium—N2B27 medium supplemented with 1 µM PD0325901, 10 ng ml$^{-1}$ recombinant human LIF, 2 µM Gö6983 (Tocris, 2285) and 2 µM XAV939 (Sigma-Aldrich, X3004)[75]. The cells were passaged onto MEFs on day 9 or 10. After this first passage, the cells were switched to t2iLGö medium supplemented with 2 µM XAV939. The t2iLGö medium comprised N2B27 medium supplemented with 1 µM PD0325901, 1 µM CHIR99021 (Axon Medchem, 1386), 2 µM Gö6983 and 10 ng ml$^{-1}$ human LIF[8]. These cells were passaged as single cells every 4–5 d through a 5-min incubation in accutase (Sigma-Aldrich, A6964-100ML) at 37 °C. Naive ICSIG-1 hPSCs were switched at passage 10 into PXGL medium for maintenance and expansion.

To convert primed H9 hESCs (used in Extended Data Fig. 5c,j) to naive hESCs, primed hPSCs were trypsinized and seeded onto gelatin-coated plates with MEFs in human KSR-primed medium along with 10 µM Y-27632 (Tocris, 1254) in humidified normoxia conditions (5% $CO_2$) for 2 d using a previously described protocol[28]. On day 3, after a PBS wash, the medium was changed to 5iLA medium—composed of 50% DMEM/F12 medium, 50% Neurobasal medium, 1% N2 supplement, 2% B27 supplement, 20 ng ml$^{-1}$ recombinant human LIF, 2 mM L-glutamine, 1% non-essential amino acids, 0.1 mM β-mercaptoethanol, 1×penicillin–streptomycin and 50 µg ml$^{-1}$ BSA (Sigma-Aldrich, A3059) supplemented with five inhibitors, that is, PD0325901 (Stemgent, 1 µM), IM-12 (Enzo, 1 µM), SB590885 (R&D Systems, 0.5 µM), WH-4-023 (A Chemtek, 1 µM), Y-27632 (Tocris, 10 µM) and activin A (Peprotech, 20 ng ml$^{-1}$)—in hypoxia conditions (5% $CO_2$ and 5% $O_2$) at 37 °C. Dome-shaped naive colonies were observed after 10–13 d. Naive cells were passaged as single cells every 4–5 d using accutase with an incubation of 5 min at 37 °C. The cells were switched into PXGL medium at passage 12.

**Naive hPSC-to-trophoblast fate conversion.** The conversion from naive hPSCs to trophoblast cells[11,13] was performed as follows. Cell culture plates were coated with 5 µg ml$^{-1}$ collagen IV (Corning, 354233) overnight at 37 °C. Naive colonies were dissociated to single cells with TrypLE (Thermo Fisher, 12605010; 10 min at 37 °C), followed by filtering through a 40-µm cell strainer. After washing the plates once with PBS, the naive hPSCs were seeded onto the collagen IV-coated plates in filter-sterilized trophoblast stem cell medium[13,62] comprising DMEM/F12 medium (Gibco, 11320033) supplemented with 0.3% BSA (Sigma-Aldrich, A3059), 0.2% FBS (Thermo Fisher, 10270106), 0.5% penicillin–streptomycin (Gibco, 15140122), 1% insulin-transferrin-selenium-ethanolamine supplement (ITS-X) (Thermo Fisher, 51500056), 8.5 µM L-ascorbic acid (Sigma-Aldrich, A4403), 0.5 µM A83-01 (PeproTech, 9094360), 1 µM SB431542 (Axon Medchem, 301836-41-9), 50 ng ml$^{-1}$ human epidermal growth factor (Miltenyi Biotec, 130-097-749), 2 µM CHIR99021 (Axon Medchem, HY-10182), 0.8 mM valproic acid (Merck, V0033000) and 0.1 mM β-mercaptoethanol (Gibco, 31350010). The medium was changed daily and supplemented with 5 µM Y-27632 (Tocris) and 1 µM UNC1999 or an equivalent volume of DMSO.

**Primed hPSC-to-trophoblast fate conversion.** Primed hiPSCs were cultured at 37 °C in filter-sterilized Essential 8 flex medium kit (Thermo Fisher, A2858501) under humidified, normoxic (20% $O_2$ and 5% $CO_2$) and feeder-free conditions. The conversion from primed hPSCs to trophoblast[72] was performed as follows. Cell culture plates were coated with Geltrex (Thermo Fisher, A1413302) and incubated at 37 °C overnight. The next day (day −4), primed hPSCs were washed with PBS and dissociated using versene (Thermo Fisher, 15040066) for 5 min at room temperature. The primed cells were collected and seeded in filter-sterilized Essential 8 flex medium kit supplemented with 1 µM UNC1999 or DMSO. The medium was changed daily and UNC1999 was freshly added every day. On day −1, the primed hPSCs were passaged at a 1:2 ratio. The following day (day 0), the medium was switched to trophoblast stem cell medium[62] as described earlier. On days 3 and 5 of conversion, the cells were passaged at a ratio of 1:2 using versene. Cells were fixed on days 0, 4 and 10 for immunofluorescence staining.

**Cell counting and viability.** Cells were counted and viability was assessed using a LUNA-FL dual fluorescence cell counter (Logos Biosystems) on days 0, 1, 2, 3 and 4 of naive hiPSC-to-trophoblast conversion. Cells were collected for cell count and viability measurements by collecting the supernatant and dissociating the attached cells to single cells using TrypLE. The cells were centrifuged for 5 min at 400g and the pellet was resuspended in 100 µl culture medium. The sample was prepared by adding 2 µl acridine orange–propidium iodide stain solution (Logos Biosystems, F23001) to 18 µl of sample (pellet diluted in culture medium). The sample preparation (10 µl) was loaded into a chamber of a PhotonSlide cell counter (Logos Biosystem) to count the total number of cells and measure cell viability.

**Human blastoids.** Naive hPSCs cultured on MEFs in PXGL medium were pre-treated with PRC2 inhibitor (1 µM UNC1999) for 4 d before blastoid induction. Blastoids were induced[19] as follows. Naive hPSCs cultured in untreated or pre-treated conditions were harvested using accutase (Biozym). The cells were resuspended in PXGL medium supplemented with 10 µM Y-27632 (MedChemExpress), seeded onto gelatin-coated plates and incubated at 37 °C for 70 min to deplete the MEFs. The unattached cells were collected, pelleted through centrifugation and resuspended in N2B27 medium containing 10 µM Y-27632 with or without PRC2 inhibitor (aggregation medium), after which 30,000 cells were seeded onto an array of 200-µm microwells inserted into a well of a 96-well plate. Note that microwell arrays comprising microwells were imprinted into 96-well plates[77,78]. After 24 h, the aggregation medium was replaced with N2B27 medium supplemented with 1 µM PD0325901, 1 µM A83-01 (MedChemExpress, HY-10432), 500 nM 1-oleoyl lysophosphatidic acid sodium salt (Tocris, 3854), 10 ng ml[−1] hLIF and 10 µM Y-27632, with DMSO for control or 1 µM UNC1999 for pre-treated samples. The medium was refreshed every 24 h.

**Immunofluorescence microscopy.** Immunofluorescence staining[79] was performed as follows. Cells were cultured on glass coverslips and washed with 1×PBS, fixed in PBS containing 4% paraformaldehyde (PFA; Life Technologies, 28908) for 10 min at room temperature, permeabilized with PBS containing 0.5% Triton X-100 for 5 min and washed twice with PBS containing 0.2% Tween 20 (PBST) for 5 min. The cells were stored at 4 °C in PBS, wrapped in parafilm until further staining, or used immediately. Primary and secondary antibodies (see below) were diluted in a blocking solution of PBST with 5% normal donkey serum (Sigma-Aldrich, S30-100) and 0.2% fish-skin gelatin. Following overnight incubation with primary antibodies in blocking solution at 4 °C, the cells were washed three times for 5 min with PBST, incubated with the appropriate fluorophore-labelled secondary antibodies in blocking solution for at least 30 min in the dark, washed with PBST, washed with a 1:50,000 DAPI (1 mg ml[−1]) solution in PBST, washed with PBST and mounted in ProLong Gold antifade reagent with DAPI (Invitrogen).

For human blastoids, aggregates were collected by gentle pipetting of blastoid medium on the microwells and transferred to U-bottomed 96-well plates (Merck, BR701330). Once the aggregates had settled, the aggregates were washed twice with PBS and fixed with 4% PFA for 30 min at room temperature with gentle shaking, followed by three 10 min washes with PBS containing 0.1% Triton X-100 (Sigma-Aldrich, X100-500). PBS containing 10% normal donkey serum and 0.3% Triton X-100 was used for blocking and permeabilization for 60 min at room temperature. The primary antibody was incubated at 4 °C in blocking/permeabilization solution and washed three times with PBS containing 0.1% Triton X-100 for 10 min. The secondary antibody was diluted in PBS containing 0.1% Triton X-100 and 1:10,000 DAPI (1 mg ml[−1]), and incubated at room temperature in the dark for 1 h. Following incubation with secondary antibodies, the blastoids were washed three times with PBS containing 0.1% Triton X-100 for 10 min and prepared for imaging. For imaging, the blastoids were placed in a glass-bottomed dish (MatTek, P35G-1.5-14-C) and directly imaged with 150 µl PBS containing 0.1% Triton X-100. Antibody information is provided in Supplementary Table 11.

Phase-contrast images were captured using a Nikon Eclipse Ti2 microscope and analysed using Nikon NIS-Elements and ImageJ. Immunofluorescence images were captured using a Zeiss AxioImager A1 inverted microscope coupled with an AxioCam MRc5 camera and the Axio Vision software. Confocal immunofluorescence images of human blastoids were acquired with a Nikon NiE upright microscope equipped with a Yokogawa CSU-X spinning-disk module with a Teledyne Photometrics Prime 95B camera and a ×20 Fluor (0.50) water-dipping objective. Optical sections with a thickness of 2 µm were collected. The images were denoised (DeNoise AI) and deconvolved (3D Richardson–Lucy algorithm), and nuclei were automatically detected and counted using NIS-Elements AR 5.30.01 via a GA3 script. Multi-channel images were processed in ImageJ.

**Flow cytometry.** For Fig. 4, naive and primed hPSCs were washed once with PBS and dissociated using accutase (BioLegend) with incubation for 5 min at 37 °C. The accutase was quenched 1:1 with medium, and the cells were passed through a 50-µm cell strainer (VWR) and centrifuged at 300g for 3 min. The cell pellets were washed once with PBS containing 2% FBS (flow buffer 1) and counted. Fluorophore-conjugated antibodies and eF780 fixable viability dye (eBioscience, 65-0865-14) were mixed with 50 µl Brilliant stain buffer (BD Biosciences) and applied to 500,000 cells in 50 µl flow buffer. Labelling occurred for 30 min at 4 °C in the dark. The cells were washed twice with flow buffer and analysed using a BD

LSR Fortessa cell analyser. Single-stained controls were used for compensation calculations and unstained cells were used in the cytometer and gating set-up. Data were analysed using the FlowJo v10.1 software (BD).

For Fig. 5, cells were dissociated using accutase (5 min incubation at 37 °C) and centrifuged at 200g for 5 min. The supernatant was removed, the cell pellets were resuspended in 300 µl FACS (PBS supplemented with 0,25–0,5% BSA) buffer per sample and centrifuged again under the same conditions. The cell pellets were incubated in 50 µl FACS buffer and antibody incubations were carried out at 4 °C for 30 min. Next, the cells were washed twice with 300 µl FACS buffer and centrifuged (200g for 5 min). The supernatant was removed and the pellet was resuspended in 300 µl PBS with 4% PFA. The samples were analysed using a BD FACSCanto II flow cytometer. For Fig. 7, blastoids were harvested from the microwell arrays and sequentially treated with 300 U ml[−1] collagenase type IV and 10×Trypsin–EDTA (Thermo Fisher) at 37 °C on a shaker. The blastoids were dissociated into single cells by pipetting. The cells were washed three times with flow buffer 2 (1% FBS in PBS) and incubated with TROP2 antibody (R&D Systems, MAB650) diluted in flow buffer and incubated for 30 min at 4 °C. The cells were centrifuged, washed twice with flow buffer and incubated with anti-mouse secondary antibody conjugated to AlexaFluor 488 (Invitrogen, A21202) for 30 min at 4 °C. The cells were centrifuged, washed twice with flow buffer and resuspended in fresh flow buffer for flow cytometry analysis. Antibody information is provided in Supplementary Table 11.

**Western blotting for histone proteins.** Cells were washed once with PBS and dissociated using accutase. Cell pellets were incubated with PBS containing complete EDTA-free protease inhibitor cocktail (Roche) for 10 min at 4 °C and centrifuged at 300g for 5 min at 4 °C. The cell pellet was resuspended in 0.2 M sulfuric acid, incubated for 30 min at 4 °C and centrifuged for 2 min at 12,000g at 4 °C. The supernatant was collected; one volume of trichloroacetic acid (Sigma) was added to the supernatant for every three volumes of sulfuric acid and incubated for 30 min at 4 °C. After centrifuging for 10 min at 12,000g at 4 °C, the supernatant was removed, and the pellets were washed with acetone and incubated for 10 min at 4 °C. After centrifuging at 1,200g for 10 min at 4 °C, a second acetone wash and centrifugation step was performed. Histone proteins were dissolved overnight in 100 mM Tris–HCl pH 8.0 containing protease inhibitors at 4 °C. The samples were centrifuged at 12,000g for 10 min at 4 °C and the supernatant was retained.

Histone proteins were quantified using the Bradford assay and denatured by heating at 95 °C in 5×Protein loading dye (4% SDS, 0.25 M Tris 6.8, 1 µM bromophenol blue, 0.5 mM dithiothreitol and 30% glycerol) for 5 min. The histones were separated by electrophoresis on a 15% SDS–PAGE gel alongside a pre-stained protein standard (Bio-Rad) to assess the protein molecular weights. The histones were transferred onto nitrocellulose membranes using an iBlot transfer system at 25 V for 10 min. The membranes were blocked for 3 h at room temperature in TBS-T (1×Tris-buffered saline and 0.05% Tween 20) containing 5% dried skimmed milk and hybridized overnight at 4 °C with primary antibodies diluted in TBS-T containing 5% milk. The membranes were washed three times with TBS-T for 10 min before incubation with fluorophore-conjugated secondary antibodies diluted in TBS-T containing 5% milk for 1 h at room temperature and protected from light. The membranes were washed three times with TBS-T for 10 min and then once with 1×TBS before detection using an Odyssey imaging system (LI-COR Biosciences) or Clarity western ECL reagent (Bio-Rad). Antibody information is provided in Supplementary Table 11.

**RNA extraction.** RNA extraction was performed using one of two methods—either TRIzol reagent or a RNeasy micro kit (Qiagen, 74004). RNA extraction with TRIzol reagent (Thermo Fisher, 15596-018) was performed according to the TRIzol reagent user guide. Briefly, cells were washed once with 1×PBS and dissociated with 400 µl TRIzol reagent for 15 min at room temperature. After collection, the samples were stored at −80 °C until further use. Chloroform (80 µl) was added to the sample, mixed, incubated for 2–3 min and centrifuged at 12,000g for 15 min at 4 °C in the presence of 1 µl glycogen. The aqueous phase containing the RNA was transferred to a new tube. Isopropanol (200 µl) was added to the sample and incubated for 10 min. Total RNA was precipitated by centrifugation at 12,000g for 10 min at 4 °C. The pellet was resuspended in 200 µl of 75% ethanol. The sample was briefly vortexed and centrifuged at 7,500g for 5 min at 4 °C. After discarding the supernatant, the RNA pellet was air dried for 10 min and resuspended in 20 µl MilliQ water. RNA extraction using the RNeasy micro kit was performed according to the RNeasy micro handbook.

**Reverse transcription.** Two reverse transcription protocols were used: (1) reverse transcription with homemade reverse transcriptase (RT) and (2) first-strand complementary DNA synthesis using SuperScript II RT (Thermo Fisher, 18064-022). The homemade RT was used for the samples that were collected with the TRIzol reagent and the SuperScript II RT for the samples collected using the RNeasy micro kit. RNA from the TRIzol-isolated samples (500 ng) was added to a mixture of 100 µg µl[−1] oligo(dT)12–18 (Thermo Fisher, 18418-012), 50 ng µl[−1] random hexamers (Thermo Fisher, S0142), 0.25 µM of each gene-specific primer (reverse), 0.2 mM of each dNTP (Thermo Fisher, 10297-018), 1×first-strand buffer (homemade), 2 mM dithiothreitol (Sigma, GE17-1318-01), 2 units µl[−1] RNaseOUT

(Thermo Fisher, 10777-019) and 1:10 M-MLV reverse transcriptase (homemade). The samples were gently mixed before incubation in a SimpliAmp thermal cycler for 10 min at 25 °C, 50 min at 42 °C, 15 min at 70 °C and then 4 °C. RNA from the RNeasy micro kit-isolated samples (50 ng) was added to a mixture of 0.5 µg µl⁻¹ oligo(dT)12–18, 1 µl gene-specific primers (reverse primer; primer sequences are provided in Supplementary Table 12) and 0.25 mM of each dNTP. This mixture was heated to 65 °C for 5 min and quickly chilled on ice. After brief centrifugation, 1×first-strand buffer, 10 mM dithiothreitol and 2 units µl⁻¹ RNaseOUT were added to the mix. After incubation at 42 °C for 2 min, 10 units of SuperScript II RT were added and mixed by gentle pipetting. The RT was activated at 42 °C for 50 min and inactivated by heating at 70 °C for 15 min. The cDNA samples were diluted with 180 µl water and stored at −20 °C.

**Quantitative PCR with reverse transcription.** Quantitative PCR was performed using a Platinum SYBR Green qPCR SuperMix-UDG kit (Invitrogen, 11733046) on an ABI ViiA7 real-time PCR system (Applied Biosystems) following the manufacturer's protocol. Each quantitative PCR well contained 1.25 ng cDNA together with the Platinum SYBR Green qPCR SuperMix-UDG, ROX reference dye (ABI, 7500) and a primer mix of the forward and reverse primers (0.25 µM each). Primer sequences are provided in Supplementary Table 12.

**Bulk RNA-seq library preparation.** RNA extraction from $0.5 \times 10^6$ cells per sample was performed using an RNeasy micro kit (Qiagen, 74004) following the manufacturer's protocol. Messenger RNA-seq libraries were prepared starting from 500 pg input RNA using a KAPA stranded mRNA-seq kit (Illumina, 07962193001) with KAPA single-indexed adapter kit sets A and B (Illumina, KR1317) following the manufacturer's protocol. Libraries were pooled with a final library concentration of 7 nM. The quality of the input RNA, cDNA and individual libraries was assessed using an Agilent 2100 Bioanalyzer system at the KU Leuven Nucleomics core. Sequencing was performed at the KU Leuven Genomics Core on a HiSeq4000 (Illumina) sequencer in single-end mode (50 bp), yielding an average of $29 \times 10^6$ reads per sample (Supplementary Table 1).

**Bulk RNA-seq analysis.** Quality assessment of the bulk RNA-seq data was performed using FastQC (v0.11.8; Babraham Bioinformatics). Samples were mapped to the human GRCh38.p12 reference genome with the corresponding GENCODE v31/Ensembl 97 using STAR (v2.7.1a)[80]. The count table was generated using featureCounts (v2.0.1)[81] with default parameters. Downstream analyses were performed using the R package DESeq2 (v1.26.0)[82]. Samples were filtered to keep genes that had more than one count in at least two conditions and the counts were transformed to the log₂-scale using the rlog function, which minimizes differences between samples for genes with small counts and normalizes with respect to library size. Differential gene expression analysis was performed using the DESeq2 function with unnormalized counts.

**Single-cell preparation and scRNA-seq.** Cells were washed with 1×PBS before dissociation from the culture dish via incubation with TrypLE express enzyme (Thermo Fisher, 12605-036) for 8 min at 37 °C. The single-cell suspensions were filtered through a Falcon 40-µm cell strainer and centrifuged at 200g for 5 min. After resuspension in 1×PBS with 0.04% BSA, the concentration of cells in the single-cell suspension was determined using a Luna-FL automated Fluorescence Cell Counter (Logos Biosystems).

Cells were loaded onto the 10X Chromium single-cell platform (10X Genomics) at a concentration of 1,000 cells µl⁻¹ (Next GEM single cell 3′ library and Gel Bead Kit v.3.1) according to the manufacturer's protocol (10X user guide, revision D). The cells were loaded targeting 4,000 cells for each run. Generation of gel beads in emulsion (GEM), barcoding, GEM-RT clean-up, complementary DNA amplification and library construction were all performed according to the manufacturer's protocol. Individual sample quality was assessed using a High sensitivity D5000 screen tape assay with the 4150 TapeStation system (Agilent). Qubit 2.0 (Thermo Fisher) and KAPA library quantification kit for Illumina Platform (KAPA Biosystems) were used for library quantification before pooling. The final library pool was sequenced on a NovaSeq6000 (Illumina) instrument using a NovaSeq SP kit (Illumina) for two lanes of 100-base-pair paired-end reads at the KU Leuven Genomics Core.

**scRNA-seq analysis.** 10X Genomics Cell Ranger (v4.0.0) was used to process, align and summarize unique molecular identifier counts for individual single-cell samples against the 10X Genomics pre-built human GRCh38 (hg38) GENCODE v32/Ensembl 98 and mouse GRCm28 (mm10) GENCODE vM23/Ensembl 98 reference genome datasets (version 2020-A, 7 July 2020). Gene Ensembl IDs were converted into gene symbols using the R package biomaRt (v2.64.3)[83]. Downstream analyses were performed with the R package Seurat (v4.0.1)[84]. Human cells were retained and mouse cells (MEFs) were filtered out by adjusting the number of counts per cell (nCount_RNA) and the number of mapped genes per cell (nFeature_RNA) to only keep cells that were mostly mapped to the human GRCh38 (hg38) genome (for naive cells: nCount_RNA < 40,000, nCount_RNA > 3,000, nFeature_RNA < 8000 and nFeature_RNA > 1,500; for day 4 of naive-to-trophoblast conversions:

nCount_RNA < 300,000, nCount_RNA > 10,000, nFeature_RNA < 12,000 and nFeature_RNA > 3,000). Naive cells with more than 25% of mitochondrial counts were filtered out. Day 4 trophoblast converted cells with more than 30% of mitochondrial counts were filtered out. Read-count tables for multiple samples were merged and cell counts were normalized using the Seurat global-scaling normalization method 'LogNormalize', which normalizes the feature expression measurements for each cell by the total expression, multiplies this by a 10,000-scale factor and log-transforms the result. Differential expression testing was performed using the FindMarkers function in Seurat based on the non-parametric Wilcoxon rank-sum test applying the log-transformed FC threshold of averaged log₂(FC) > 0.25. A graph-based cell-clustering approach was used to cluster cells with the FindClusters function in Seurat. Loom files were generated in R using build_loom and add_col_attr from SCopeLoomR (version 0.3.1).

**Integration of scRNA-seq data.** Datasets used for gene expression integration can be found in ArrayExpress under the accession number E-MTAB-3929 (ref. [3]) and in the Gene Expression Omnibus (GEO) database under the accession number GSE109555 (ref. [74]). Integration of published scRNA-seq embryonic datasets[3,74] with the day 4 trophoblast conversion ± PRC2i scRNA-seq dataset generated in this study was performed using Seurat's canonical correlation analysis integration tool. Anchors for integration were found using the FindIntegrationAnchors function with default arguments. Parameters and data were integrated across all features. An integration-based UMAP was constructed using the runUMAP function with dims: 1:30. Published scRNA-seq embryonic datasets[3,74] were annotated following[13] for Extended Data Fig. 6e. For Fig. 6a–d, annotations of[3,74] from[13] were simplified as follows: embryo trophectoderm (comprising early, medium and late trophectoderm), embryo trophoblast (comprising early, medium, late and apoptosis trophoblast), embryo extravillous trophoblast (comprising pre-extravillous trophoblast and extravillous trophoblast), embryo syncytiotrophoblast (comprising pre-syncytiotrophoblast and syncytiotrophoblast) and (pre)blastocyst (comprising morula and B1/B2 blastocyst).

**Calibrated CUT&RUN.** Concanavalin A-conjugated paramagnetic beads (EpiCypher, 21-1401) were resuspended and washed twice on a magnetic rack in bead activation buffer composed of 20 mM HEPES pH 7.9, 10 mM KCl, 1 mM CaCl₂ and 1 mM MnCl₂. Naive or primed hPSCs and *Drosophila* S2 cells were dissociated and counted. Cell pellets were washed twice with a wash buffer composed of 20 mM HEPES pH 7.5, 150 mM NaCl and 0.5 mM spermidine supplemented with EDTA-free protease inhibitor. The cells, 50,000 human cells and 20,000 *Drosophila* cells in wash buffer, were added to concanavalin A beads and incubated for 10 min at room temperature with rotation to immobilize the cells. The concanavalin A beads were collected with a magnetic rack, the supernatant was discarded and the beads were resuspended in antibody buffer (wash buffer supplemented with 0.08% digitonin (Millipore) and 2 mM EDTA). Antibody (1 µg) was added to the beads, which were incubated overnight at 4 °C with rotation. Antibody information is provided in Supplementary Table 11. The following day, the beads were washed twice with digitonin buffer (wash buffer supplemented with 0.08% digitonin). CUTANA pA/G MNase (2.5 µl; EpiCypher, 15-1016) was added to 50 µl digitonin buffer and incubated for 10 min at room temperature. The beads were washed twice with digitonin buffer, and the MNase was activated with 2 mM CaCl₂ and incubated for 2 h at close to 0 °C. MNase activity was terminated by adding 100 µl stop buffer (340 mM NaCl, 20 mM EDTA, 4 mM EGTA, 50 µg ml⁻¹ RNase A and 50 µg ml⁻¹ glycogen). Cleaved DNA fragments were released from nuclei by incubation for 10 min at 37 °C, centrifugation for 5 min at 16,000g at 4 °C and collection of the supernatant from the beads on a magnetic rack. The DNA was purified by incubation with 1 µl SDS (20%) and 1.5 µl proteinase K (20 mg ml⁻¹) at 70 °C for 10 min, followed by a 1.8×AMPure XP bead (Beckman Coulter) clean-up into DNA lo-bind tubes (Eppendorf) and elution in 50 µl of 0.1×TE. Libraries were prepared using a NEBNext ultra II DNA library preparation kit for Illumina (NEB) using the manufacturer's protocol, with libraries indexed using NEBNext multiplex oligos for Illumina (index primers sets 1 and 2; NEB). Following library preparation, the library fragment size and concentration were determined using a Qubit fluorometer double stranded DNA high sensitivity assay kit with an Agilent Bioanalyzer 2100 and using a KAPA library quantification kit (KAPA Biosystems). The samples were sequenced on an Illumina NextSeq500 instrument as HighOutput 75-base-pair paired-end reads at the Babraham Institute Next Generation Sequencing Facility (highest read count = 53,117,572, lowest read count = 20,861,009, and average read count = 32,213,678).

**Calibrated CUT&RUN analysis.** Raw FastQ data were trimmed using TrimGalore (v.0.6.6, Babraham Bioinformatics) and aligned to the GRCh38 genome or the *Drosophila* BDGP6 genome using Bowtie2 (v.2.3.2)[85] with the following parameters –very-sensitive -I 10 -X700. High-quality reads with a mapping quality value of >20 were retained by filtering using samtools view (v.1.11)[86]. Calibration factors for each sample were determined as the ratio of the sample with the lowest number of unique mapped *Drosophila* spike-in tags over the number of unique mapped *Drosophila* spike-in tags per sample. These calibration factors were then used to scale the human genome-mapped reads by random downsampling. Calibrated browser extensible data (BED) files were produced using BEDTools genome cov

scaling by these calibration factors and were used for peak calling (v.2.29.2)[87]. Peak calling was performed using the CUT&RUN optimized Sparse Enrichment Analysis for CUT&RUN (SEACR) algorithm and the top 1% peaks were retained (v.1.3)[51]. Peaks closer than 300bp were merged using BEDTools merge and peaks common to both replicates were determined by BEDTools intersect to generate final peak sets for naive and primed. Peaks called in naive and primed hPSC datasets were concatenated into a combined peak list and de-duplicated. Peaks that were differentially enriched between naive and primed hPSCs were then determined from this concatenated list using a DESeq2 implementation in SeqMonk (v.1.47.2; Babraham Bioinformatics) to identify differential regions with $P < 0.05$ after Benjamini–Hochberg multiple-testing correction. Common peaks were classified as peaks in the concatenated list that were not statistically enriched in either condition. These peaks were then filtered against the ENCODE GRCh38 exclusion list to remove coverage outliers. The fraction of reads in peak scores were calculated using tools from the deepTools API suite and processed using custom Python scripts (v.3.7.3). Peaks were annotated for genomic context using the R package ChIPseeker and a promoter cutoff of ±3kb of the transcription start site and a 'gene' level annotation (v.1.30.3)[88]. Peaks were annotated to the nearest genes using HOMER annotatePeak.pl and peaks within 10kb of the nearest transcription start site were retained as marked genes[89]. Calibrated bigWig files were produced using deepTools bamCoverage with scaling by the calculated calibration factors (v.3.43)[90]. Replicates were merged and processed using UCSC-tools bigWigMerge and bedGraphToBigWig and custom R scripts for visualization on the WashU epigenome browser (v.5)[91,92]. Heatmaps and profiles over called peaks were produced using deepTools computeMatrix with the following settings --missingDataAsZero and plotted with plotHeatmap or plotProfile.

For analysis of 1-kb windows of the genome, the coverage of 1-kb bins of sorted and indexed binary alignment map (BAM) files were processed by deepTools multiBamSummary scaling by the calibration normalization factors previously calculated. Replicate reproducibility was assessed by Pearson's correlation of signal across these 1-kb bins using deepTools plotCorrelation with the additional parameter --log1p for plotting. Principal component analysis plots were produced using signals at combined naive and primed peak sets on downsampled BAM files in Seqmonk (v.1.47.2; Babraham Bioinformatics). Scatter, violin, box, bar and density plots were produced using the R package ggplot2.

**Data processing and visualization.** Figures were produced in R using the R package ggplot2 (v.3.3.3), Microsoft Excel, BioRender and Adobe Illustrator. The online application for viewing the multi-omic datasets was developed using the Shiny package for R.

**Histone extraction, propionylation and digestion.** Histone preparation of naive (cultured in PXGL medium) and primed (cultured in E8 medium) hPSCs was performed starting from frozen cell pellets by isolating the nuclei through resuspension in hypotonic lysis buffer (10mM Tris–HCl pH 8.0, 1mM KCl and 1.5mM MgCl$_2$) complemented with 1mM dithiothreitol and complete protease inhibitors (Roche) at $4 \times 10^6$ cells per 800µl. To promote lysis, the cells were rotated for 30min at 4°C and centrifuged for 10min at 16,000g. Resuspension of the nuclei in 0.4N HCl (cell density of $8 \times 10^3$ cells µl$^{-1}$) was followed by incubation on a rotator for 30min at 4°C and centrifugation for 10min at 16,000g. The supernatant was transferred, and histones were precipitated using 33% trichloroacetic acid. The samples were incubated on ice for 30min, followed by centrifugation for 10min at 16,000g and 4°C. After removal of the supernatant, the samples were washed twice with ice-cold acetone, followed by centrifugation for 5min at 16,000g and 4°C to remove the remaining trichloroacetic acid. Of the resulting histone extracts, a fraction corresponding to $4 \times 10^5$ cells was isolated for histone quantification and normalization through one-dimensional SDS–PAGE on a 9–18% TGX gel (Bio-Rad). Propionylation and digestion was performed on the remaining $3,6 \times 10^6$ cells (22.5µg) of each sample as previously described[93,94]. Briefly, histones were resuspended in 20µl of 1M triethylammonium bicarbonate and 20µl propionylation reagent (isopropanol:propionic anhydride, 158:2). Following a 30-min incubation at room temperature, 20µl MilliQ water was added and the samples were incubated for 30min at 37°C. After vacuum drying, the samples were resuspended in 500mM triethylammonium bicarbonate, 1mM CaCl$_2$, 5% acetonitrile and trypsin (1:20 ratio) to a final volume of 50µl. The samples were incubated overnight at 37°C and vacuum dried. The propionylation reaction was carried out once more, identically to the first reaction, to cap newly formed peptide N termini. Overpropionylation of serine, threonine and tyrosine was reversed by adding 50µl of 0.5M NH$_2$OH and 15µl NH$_4$OH at pH 12 to the vacuum dried samples for 20min at room temperature. Finally, 30µl of 100% formic acid was added and the samples were vacuum dried.

Histone preparation of naive (cultured in t2iLGö medium) and primed (cultured in E8 medium) hPSCs was performed starting from frozen cell pellets[95]. The cells were thawed on ice and resuspended in Nuclei Isolation Buffer (15mM Tris–HCl pH 7.5, 15mM NaCl, 60mM KCl, 5mM MgCl$_2$, 1mM CaCl$_2$, 250mM sucrose, 100mM dithiothreitol, 0.5mM 4-(2-aminoethyl)benzenesulfonyl fluoride hydrochloride, 5nM microcystin and 10mM sodium butyrate). After centrifugation at 700g for 5min at 4°C, the pellets were resuspended in 0.1% (vol/vol) NP-40 Alternative in Nuclei Isolation Buffer for 5min on ice. The cells were

collected by centrifugation at 700g for 5min at 4°C and the pellets were washed three times in Nuclei Isolation Buffer. The washed nuclei pellets were resuspended in 0.4N H$_2$SO$_4$ and incubated with gentle shaking for 2h at 4°C. The samples were collected by centrifugation at 3,400g for 5min at 4°C. The supernatant (containing the acid-extracted histones) was transferred to a new tube and mixed with 100% trichloroacetic acid such that the final concentration was 30%. After incubation on ice overnight, histones were collected by centrifugation at 3,400g for 5min at 4°C, washed twice with 0.1% HCl in acetone and once with 100% acetone. The washed histones were collected by centrifugation at 20,000g for 5min at 4°C, washed once more with acetone and then left to air dry. The dried pellets were resuspended in mass spectrometry-grade water and adjusted to pH 8.0 with 100mM NH$_4$HCO$_3$. The histone concentration was estimated using the Bradford assay. About 20µg of purified histones in 20µl of 100mM NH$_4$HCO$_3$ (pH 8.0) were derivatized in 5µl propionylation reagent (mixture of propionic anhydride with acetonitrile at 1:3 (vol/vol)) for 15min at room temperature. This reaction was repeated to ensure complete propionylation of unmodified lysines. Following this, the histones were digested overnight with 1µg trypsin at room temperature. After digestion, the N termini of peptides were derivatized by two more rounds of propionylation. For injection into the mass spectrometer, the samples were desalted using C18 stage-tips.

**Liquid chromatography with MS/MS (hPTMs).** The propionylated naive (cultured in PXGL medium) and primed (cultured in E8 medium) samples, complemented with a β-galactosidase (Sciex) and MPDS (Waters) internal standard, were resuspended in 0,1% formic acid resulting in 1.5µg histones and 50fmol β-galactosidase and MPDS on column in a 9µl injection. A quality control mixture was created by combining 2µl of each sample. Data-dependent acquisition was performed on a TripleTOF 6600+ system (AB Sciex) operating in positive mode coupled to an Eksigent NanoLC 425 HPLC system operating in capillary flow mode (5µl min$^{-1}$). Trapping and separation of the peptides was carried out on a Triart C18 column (5mm × 0.5mm; YMC) and a Phenomenex Luna Omega Polar C18 column (150mm × 0.3mm, particle size 3µm), respectively, using a low pH reverse-phase gradient. Buffers A and B of the mobile phase consisted of 0,1% formic acid in water and 0,1% formic acid in acetonitrile, respectively. A 60-min gradient going from 3% to 45% Buffer B, with a total run time of 75min per sample, was applied. The samples were run in a randomized fashion and a quality control injection was incorporated every five samples. For each cycle, one full MS1 scan ($m/z$ 400–1,250) of 250ms was followed by an MS2 ($m/z$ 65–2,000, high-sensitivity mode) of 200ms. A maximum of ten precursors (charge state +2 to +5) exceeding 300c.p.s. were monitored, followed by an exclusion for 10s per cycle. A rolling collision energy with a spread of 15V and a cycle time of 2,3s was applied.

The propionylated naive (cultured in t2iLGö medium) and primed (cultured in E8 medium) samples were analysed using an EASY-nLC nanoHPLC (Thermo Scientific) fitted with a nano-column packed with inner diameter of 75µm × 17cm Reprosil-Pur C18-AQ (3µm; Dr. Maisch GmbH). Online mixing of solvents was as follows: 2–28% solvent B (solvent A, 0.1% formic acid; solvent B, 95% acetonitrile and 0.1% formic acid) over 45min, followed by 28–80% solvent B in 5min and 80% solvent B for 10min at a flow rate of 300nl min$^{-1}$, which allowed separation of analyte components and spray into the Q-Exactive mass spectrometer (Thermo Scientific). A data-independent acquisition method, consisting of a full-scan mass spectrometry spectrum ($m/z$ 300–1,100) at a resolution of 70,000 (at 200$m/z$) and tandem mass spectrometry (MS/MS) of windows of 50$m/z$ at a resolution of 15,000 was used. MS/MS data were acquired in centroid mode. The data-independent acquisition data were searched using EpiProfile[96].

**Analysis of hPTMs.** Analysis of the mass spectrometry data was performed as previously described[97]. For all runs, raw data were imported in Progenesis QIP 4.2. (Nonlinear Dynamics, Waters), followed by alignment, feature detection and normalization. Next, a mascot generic format (MGF) file was created based on the twenty MS/MS spectra closest to the elution apex and exported for searches using Mascot (Matrix Science). First, a standard search was performed on the exported MGF file to identify non-propionylated standards (β-galactosidase and MPDS) and to verify underpropionylation. Second, to identify the proteins present in the sample and to detect unexpected hPTMs, an error-tolerant search without biological modifications was carried out against a complete Human Swiss-Prot database (downloaded from UniProt and supplemented with contaminants from the cRAP database (https://www.thegpm.org/crap/)). Subsequently, a FASTA database was created based on the results of the error-tolerant search. In addition, the highest-ranking hPTMs that emerged from this search, complemented with the biologically most interesting hPTMs (acetylations and methylations), were selected to establish a set of nine hPTMs for further analysis. Next, the three MS/MS spectra closest to the elution apex per feature were merged into a single MGF file and exported for a Mascot search including the following parameters: (1) a mass error tolerance of 10ppm and 50ppm for the precursor and fragment ions, respectively; (2) Arg-C enzyme specificity, allowing for up to one missed cleavage site; (3) variable modifications included acetylation, butyrylation, crotonylation as well as trimethylation on lysine, methylation on arginine, dimethylation on both lysine and arginine, deamidation on asparagine, glutamine and arginine

(the latter representing citrullination), phosphorylation on serine and threonine, and oxidation of methionine; and (4) fixed modifications included N-terminal propionylation and propionylation on lysine. The search was performed against the above-mentioned custom-made FASTA database. This Mascot result file (extensible markup language (XML) format) was imported into Progenesis QIP 4.2 for annotation. To resolve isobaric near-coelution, features that were identified as histone peptidoforms were manually validated and curated by an expert. To correct for variations in sample loading, the samples were normalized against all histone peptides. Outlier detection and removal was done based on the principal component analysis. Finally, the deconvoluted peptide ion data of all histones were exported from Progenesis QIP 4.2 for further analysis.

**Acid extractome analysis after histone extraction.** Next to histones, other alkaline proteins remain in the HCl during acid extraction[32]. For this purpose, we used the MSqRob software[98], which was developed for relative protein quantification by implementing the peptide-level robust ridge-regression method. First, the deconvoluted peptide ion data of all identified peptides were exported from Progenesis QIP 4.2. (that is, the same project that was created in the section 'Analysis of hPTMs' was used). MSqRob requires an annotation file that contains the name of the runs included in the experiment as well as the condition to which each run belongs (that is, naive, naive + inhibitor, primed and primed + inhibitor). Log transformation and quantile normalization of the data were performed. Finally, pairwise comparisons (naive versus naive + inhibitor, primed versus primed + inhibitor and naive versus primed) were carried out and the result files were exported for further use.

**(Hydroxy)methylation measurements of genomic DNA.** Genomic DNA was isolated using the Wizard genomic DNA isolation kit (Promega). Mass spectrometry analysis of the nucleosides was performed on genomic DNA digested using DNA degradase plus (Zymo Research). The individual nucleosides were measured using a high-performance liquid chromatography–MS/MS system consisting of an Acquity UPLC (Waters) containing a Waters Atlantis Hilic column (3 μm, 2.1 mm × 100 mm) connected to a Micromass Quattro Premier XE (Waters). Quantification was performed using area-based linear regression curves derived from calibration standards containing internal standard solutions corresponding to 0.025, 0.05, 0.1, 0.2, 0.5, 1 and 2 mg of DNA. The 5mC and 5hmC levels were calculated as a concentration percentage ratio of the percentage of 5-methyl-20-deoxycytidine/20-deoxyguanosine (%mdC/dG) and the percentage of 5-hydroxymethyl-2′-deoxycytidine/2′deoxyguanosine (%hmdC/dG), respectively.

**ChEP.** Chromatin enrichment[99] was performed as follows. Cells were crosslinked in 1% PFA for 10 min at 37 °C, quenched with 0.25 M glycine, washed twice in PBS, scraped and transferred to 2 ml tubes. The cells were resuspended in 1 ml ice-cold cell lysis buffer (25 mm Tris pH 7.4, 0.1% Triton X-100, 85 mM KCl and 1×Roche protease inhibitor) and centrifuged at 2,300g for 5 min at 4 °C. The supernatant (cytoplasmic fraction) was removed and the cell pellets were resuspended in 500 μl SDS buffer (50 mM Tris pH 7.4, 10 mM EDTA, 4% SDS and 1×Roche protease inhibitor), incubated at room temperature for 10 min, topped up to 2 ml with urea buffer (10 mM Tris pH 7.4, 1 mM EDTA and 8 M urea) and centrifuged at 16,100g for 30 min at room temperature. The supernatant was discarded and this step was repeated once. Next, the pellet was resuspended in 500 μl SDS buffer, topped up to 2 ml with SDS buffer and centrifuged at 16,100g for 30 min at room temperature. The cell pellet was taken up in 250 μl storage buffer (10 mM Tris pH 7.4, 1 mM EDTA, 25 mM NaCl, 10% glycerol and 1×Roche protease inhibitor) and sonicated in an NGS Bioruptor (Diagenode) for five cycles (30 s on, 30 s off) to solubilize the pellet. The concentration of the resulting lysate was measured using a Qubit assay (Invitrogen). For sample preparation for mass spectrometry, 30 μg of protein extract was de-crosslinked for 30 min at 95 °C by adding 4×decrosslinking buffer (250 mM Tris pH 8.8, 8% SDS and 0.5 M 2-mercaptoethanol) to final 1×.

**Liquid chromatography with MS/MS analysis (ChEP).** De-crosslinked chromatin extracts (30 μg) were processed using filter-aided sample preparation[100] and digested overnight with trypsin. The digested samples were fractionated using strong anion exchange[101], where we collected and included three fractions for analysis per sample: flow through as well as pH 8 and pH 2 elutions. Peptides were subjected to Stage-Tip desalting and concentration[102] before liquid chromatography with mass spectrometry analysis. Three replicates for each sample were analysed using liquid chromatography with MS/MS. Peptides were applied to reverse-phase chromatography using a nanoLC-Easy1000 coupled online to a Thermo Orbitrap Q-Exactive HF-X. Using a 120-min gradient of Buffer B (80% acetonitrile and 0.01% TFA), peptides were eluted and subjected to MS/MS. The mass spectrometer was operated in Top20 mode and dynamic exclusion was applied for 30 s.

**Proteomic data analysis.** Raw mass spectrometry data were analysed using MaxQuant[103] (v.1.6.6.0) and searched against the curated UniProtKB human proteome (downloaded 27 June 2017) with default settings and LFQ, IBAQ and match between runs enabled. The identified proteins were searched against a decoy database from MaxQuant. Proteins flagged as 'reverse', 'potential contaminant' or 'only identified by site' were filtered from the final protein list. Biological triplicates

were grouped to calculate differential proteins. Data were filtered for three valid values in at least one group. Missing values were imputed using default settings in Perseus (v.1.5.0.0) based on the assumption that they were not detected because they were under or close to the detection limit. Differential proteins between triplicates were calculated using a Student's t-test ($P < 0.05$) and FC > 2. Generation of volcano plots and downstream analysis of proteomics data were performed using R. For data integration of the hPTM and the chromatin proteome, writers and erasers were selected from the HIstome database[104] and literature[105–107]. The quantified epigenetic modifiers and their target modification were integrated using Cytoscape v.3.7.2 (ref. [108]).

**Statistics and reproducibility.** Statistical tests and data processing were performed in R (v.4.0.3) or GraphPad Prism 9 (GraphPad, v.9.2.0). A two-sided $P < 0.05$ was considered statistically significant. No statistical method was used to pre-determine the sample size and the experiments were not randomized. Samples that failed standard quality control and filtering were excluded as described in the appropriate methods section and the accompanying Reporting Summary. The investigators were not blinded to allocation during experiments and outcome assessment. The RNA-seq, chromatin proteomics, histone proteomics and DNA methylation assays were performed at least three times. The naive hPSC-to-trophoblast conversion experiments and blastoid assays were performed three times unless otherwise specified in the legend. The cCUT&RUN assays (Figs. 3, 4b,c and Extended Data Figs. 3, 4e) were performed twice unless otherwise specified in the legend. The experiments shown in Fig. 5c and Extended Data Figs. 4k, 5c(left),e,j,l were performed twice. The experiments shown in Fig. 7c and Extended Data Figs. 5c(right),d,g,i, 7a–c,g were performed once.

**Reporting summary.** Further information on research design is available in the Nature Research Reporting Summary linked to this article.

## Data availability

The multi-omic data can be explored using the online resource https://www.bioinformatics.babraham.ac.uk/shiny/shiny_omics/Shiny_omics. The scRNA-seq loom files can be visualized on the SCope platform: https://scope.aertslab.org/#/HumanPluripotencyPRC2/*/welcome. The RNA-seq, cCUT&RUN and scRNA-seq datasets have been deposited in the GEO under the accession code GSE176175. The hPTM mass spectrometry proteomics datasets have been deposited in the ProteomeXchange Consortium via the PRIDE partner repository (https://www.ebi.ac.uk/pride/)[109] under the dataset identifiers PXD028162 and PXD032792. The project with the identifier PXD028162 (consultable via ProteomeXchange) was licensed on a single-run basis and is fully accessible and editable by the readership after free download of the Progenesis QIP 4.2 software (https://www.nonlinear.com/progenesis/qi-for-proteomics/). The ChEP mass spectrometry proteomics datasets have been deposited in the ProteomeXchange Consortium via the PRIDE partner repository (https://www.ebi.ac.uk/pride/)[109] under the dataset identifier PXD028111. Datasets were downloaded as provided by Petropoulos et al.[3] (ArrayExpress: E-MTAB-39293), Zhou et al.[74] (GEO: GSE10955580) and van Mierlo et al.[38] (GEO: GSE101675). Public databases used in this manuscript include Human Swiss-Prot database (https://www.uniprot.org/), cRAP database (https://www.thegpm.org/crap/), HIstome database (http://www3.iiserpune.ac.in/~coee/histome/) and UniProtKB human proteome (https://www.uniprot.org/). Source data are provided with this paper. All other data supporting the findings of this study are available from the corresponding authors on reasonable request.

## Code availability

Codes pertaining to analyses in this study are available from https://github.com/pasquelab/PRC2 (ref. [110]), https://github.com/laurabiggins/Shiny_omics (ref. [111]) and https://github.com/AndrewAMalcolm/Zijlmans-et-al.-2022 (ref. [112]).

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

## Acknowledgements

We thank J. Ooghe and the Vlaamse Supercomputer Center Leuven (https://www.vscentrum.be) for computing and the VIB/KU Leuven Center for Brain and Disease Research, in particular to K. Davie and S. Aerts for including the scRNA-seq data on Scope. We thank the UZ/KU Leuven Genomics Core (http://genomicscore.be) for high-throughput sequencing expertise, S. Schlenner and the KU Leuven/VIB FACS Core, the personnel of the KU Leuven Animal Facilities, J. C. Marine's laboratory for the use of Tapestation and 10X Chromium Controller and in particular G. Bervoets for helpful feedback. We thank A. Smith for providing naive reset H9 hESCs and C. Verfaillie for providing Sigma hiPSCs. We thank L. David for helpful discussions; and B. Thienpont, L. Roderick and T. Voet for helpful feedback during thesis committee meetings. The V.P. laboratory is part of the Leuven Single-Cell Omics Institute (LISCO) and Leuven Stem Cell Institute (SCIL). We thank F. Krueger from Babraham Bioinformatics for sequencing quality control and mapping analysis, P. Kokko-Gonzales and A. Edwards at the Babraham Institute Next Generation Sequencing Facility, the Wellcome–MRC Cambridge Stem Cell Institute Tissue Culture Facility for providing reagents and C. Semprich for help with the CUT&RUN protocol. We thank B. V. Puyvelde for his expertise and support in operating the liquid chromatography with mass spectrometry system. Research in the V.P. laboratory is supported by The Research Foundation–Flanders (FWO; Odysseus Return grant no. G0F7716N to V.P.; FWO grant nos G0C9320N and G0B4420N to V.P.; the Pandarome project 40007487, which received funding from the FWO and FRS-FNRS under the Excellence of Science (EOS) programme; the KU Leuven Research Fund (C1 grant no. C14/21/19 to V.P.) and FWO PhD fellowships to A.J. (grant no. 1158318N), I.T. (grant no. 1S72719N), R.N.A. (grant no. 11L0722N), L.V. (grant no. 1S29419N), S.K.T. (grant no. 1S75720N) and T.X.A.P. (grant no. 11N3122N). Work in the P.J.R.-G. laboratory is supported by grants from the BBSRC (grant nos BBS/E/B/000C0421 and BBS/E/B/000C0422, Core Capability Grant to P.J.R.-G. and Cambridge Biosciences DTP Studentship to A.B.), the MRC (grant nos MR/T011769/1, MR/V02969X/1 and MR/N018419/1 to P.J.R.-G., and MR/J003808/1 to A.J.C.) and the Wellcome Trust (grant nos 215116/Z/18/Z to P.J.R.-G. and 102160/Z/13/Z to A.A.M.). The H.M. laboratory is supported by an NWO-XS grant (grant no. OCENW.XS5.052 to H.M.). D.W.Z. and M.V. are part of the Oncode Institute, which is partly funded by the Dutch Cancer Society. Research in the ProGenTomics laboratory is supported by FWO mandates awarded to S.V. (grant no. 3S031319) and M.D. (grant no. 12E9716N). Research in the N.R. laboratory is supported by the European Research Council (ERC) under the European Union's Horizon 2020 research and innovation programme (ERC-Co grant agreement no. 101002317 'BLASTOID: a discovery platform for early human embryogenesis'). The B.A.G. laboratory is supported by NIH grant nos P01CA196539 and AG031862 to B.A.G.

## Author contributions

Conceptualization: M.D., H.M., P.J.R.-G. and V.P. Data curation: D.W.Z., I.T., A.A.M., S.V. and A.B. Formal analysis: D.W.Z., I.T., S.V., A.B., A.A.M. and N.V.B. Funding acquisition: F.L., D.D., J.H.J., B.A.G., M.V., N.R., M.D., H.M., P.J.R.-G. and V.P. Investigation (omics experiments): D.W.Z., I.T., S.V., A.B., A.J.C., N.V.B., S.K.T., A. Janiszewski, R.N.A., D.A. and R.K. Investigation (stem cell experiments): I.T., A.B., K.V.N., S.S.F.A.V.K., A.J.C., C.F., P.J.R.-G., B.P.B., A.P., J.C., T.X.A.P., M.O., C.E., V.P., L.V. and P.A. Investigation (blastoid experiments): A. Javali, I.T., G.G. and N.C. Methodology: D.W.Z., I.T., S.V., A.B., A. Javali, V.P., H.M., P.J.R.-G., M.D. and N.R. Project administration: D.W.Z., I.T., M.D., H.M., P.J.R.-G. and V.P. Resources: F.L., D.D., J.H.J., B.A.G., M.V., N.R., M.D., H.M., P.J.R.-G. and V.P. Software: L.B. Supervision: D.D., B.A.G., M.V., N.R., M.D., H.M., P.J.R.-G. and V.P. Validation: I.T., A. Javali, S.S.F.A.V.K., N.C., B.P.B., A.P., L.V., N.R., M.D., H.M., P.J.R.-G. and V.P. Visualization: D.W.Z., I.T., A.B., A.A.M., L.B., H.M., P.J.R.-G. and V.P. Writing, reviewing and editing of the manuscript: D.W.Z., I.T., S.V., A.B., A.A.M., M.D., H.M., P.J.R.-G. and V.P.

## Competing interests

The Institute for Molecular Biotechnology, Austrian Academy of Sciences has filed patent application no. EP21151455.9 describing the protocols for human blastoid formation and for the blastoid–endometrium interaction assay. A.J. and N.R. are the inventors on this patent. All other authors declare no competing interests.

## Additional information

**Extended data** is available for this paper at https://doi.org/10.1038/s41556-022-00932-w.

**Correspondence and requests for materials** should be addressed to Maarten Dhaenens, Hendrik Marks, Peter J. Rugg-Gunn or Vincent Pasque.

**a**

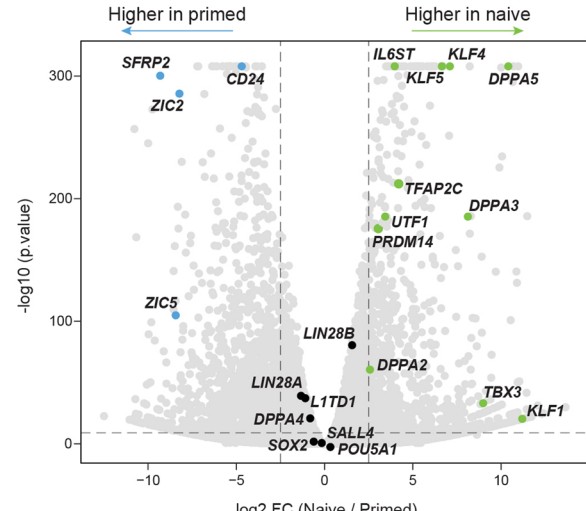

**b**

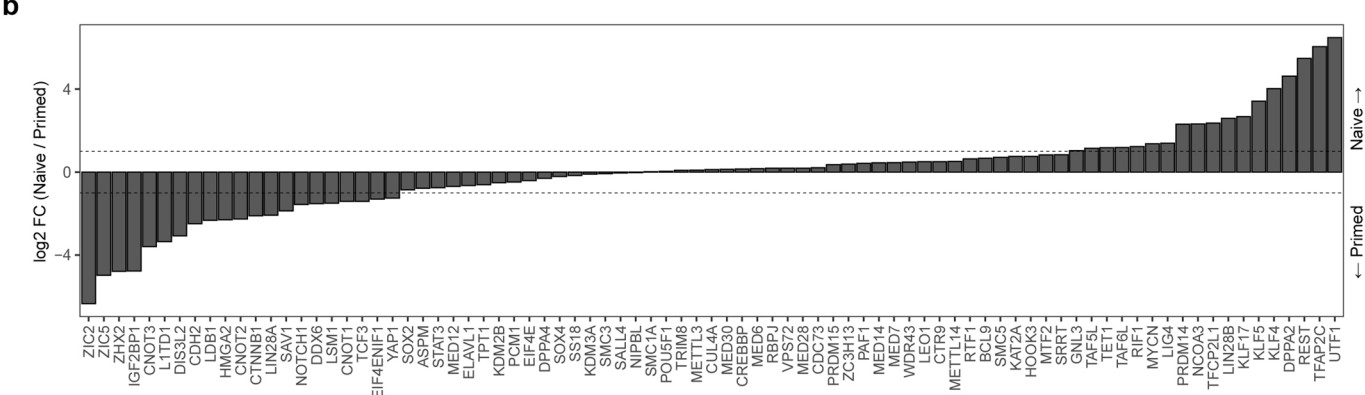

**c**

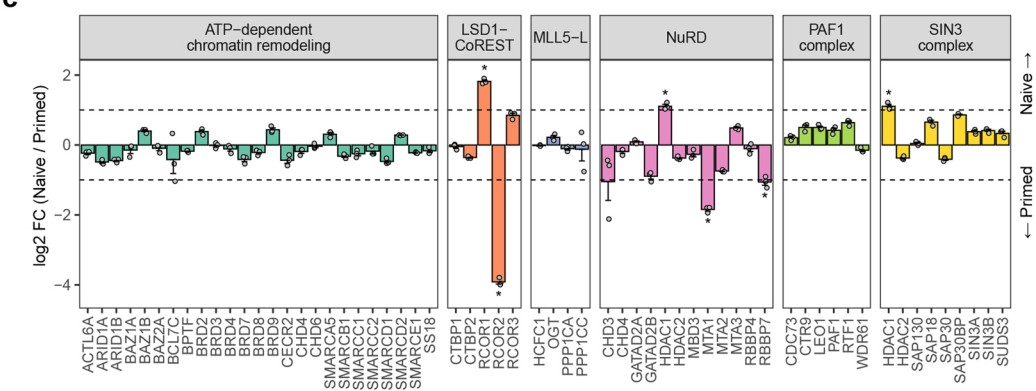

**d**

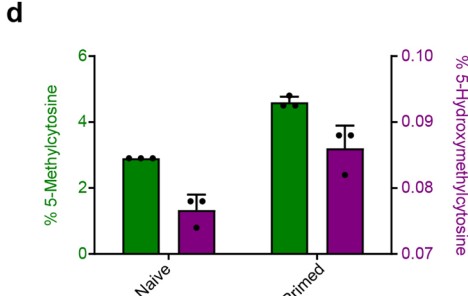

**Extended Data Fig. 1 | See next page for caption.**

**Extended Data Fig. 1 | Global analysis of chromatin proteome and transcriptome in naive and primed hPSCs. a**. Volcano plot showing differential gene expression as detected by RNA-seq between naive and primed hPSCs (n = 3 biologically independent samples). Core pluripotency factors (black) as well as factors specific to each pluripotent state (blue for primed; green for naive) are highlighted. Dashed lines indicate p-value < 0.05 and $log_2$fold change > 2.5 (two-sided student's t test). P-values can be found in Supplementary Table 8. **b**. Chromatin occupancy of all proteins with a potential role in stem cell maintenance in naive and primed hPSCs (n = 3 biologically independent samples). Protein names were collected from AmiGO (http://amigo.geneontology. org/; 'stem cell maintenance' GO:0019827)[38,50]. The following pluripotency-associated proteins were not detected in our dataset: ASCL2, BMP7, BMPR1A, DAZL, DLL1, ERAS, ESRRB, FANCC, FGF10, FGF4, FGFR1, FOXO3, FZD7, HES1, HES5, HESX1, ID1, ID2, ID3, JAG1, KIT, KLF10, KLF2, LBH, LDB2, LIF, LOXL2, LRP5, MCPH1, MED21, MED27, MMP24, MYC, NANOG, NANOS2, NODAL, NOG, NOTCH2, NR0B1, NR2E1, PADI4, PAX2, PAX8, PELO, PHF19, PIWIL2, PRDM16, PROX1, PRRX1, PTN, RAF1, SETD6, SFPI1, SFRP1, SIX2, SKI, SMO, SOX9, SPI1, TAL1, TBX3, TCF15, TCL1, TERT, TP63, TUT4, WNT7A, WNT9B, ZFP36L2, ZNF322, ZNF358, ZNF706. **c**. ChEP analysis of differential abundance of chromatin-associated complexes in naive and primed hPSCs (n = 3 biologically independent samples), supplementing Fig. 1d,e. Data are presented as mean values +/- SEM. Dashed lines represent 2-fold change. Asterisks indicate p-value < 0. 05 and fold change > 2 (two-sided student's t test). Low-change proteins (log2 FC < 0.5) involved in ATP-dependent chromatin remodelling (Fig. 1e) are shown. **d**. Mass spectrometry analyses of global levels of DNA methylation (green) and DNA hydroxymethylation (purple) in naive and primed hPSCs (n = 3 biologically independent samples) Data are presented as mean values +/- SD. Underlying source data is provided in Source Data Extended Data Fig. 1.

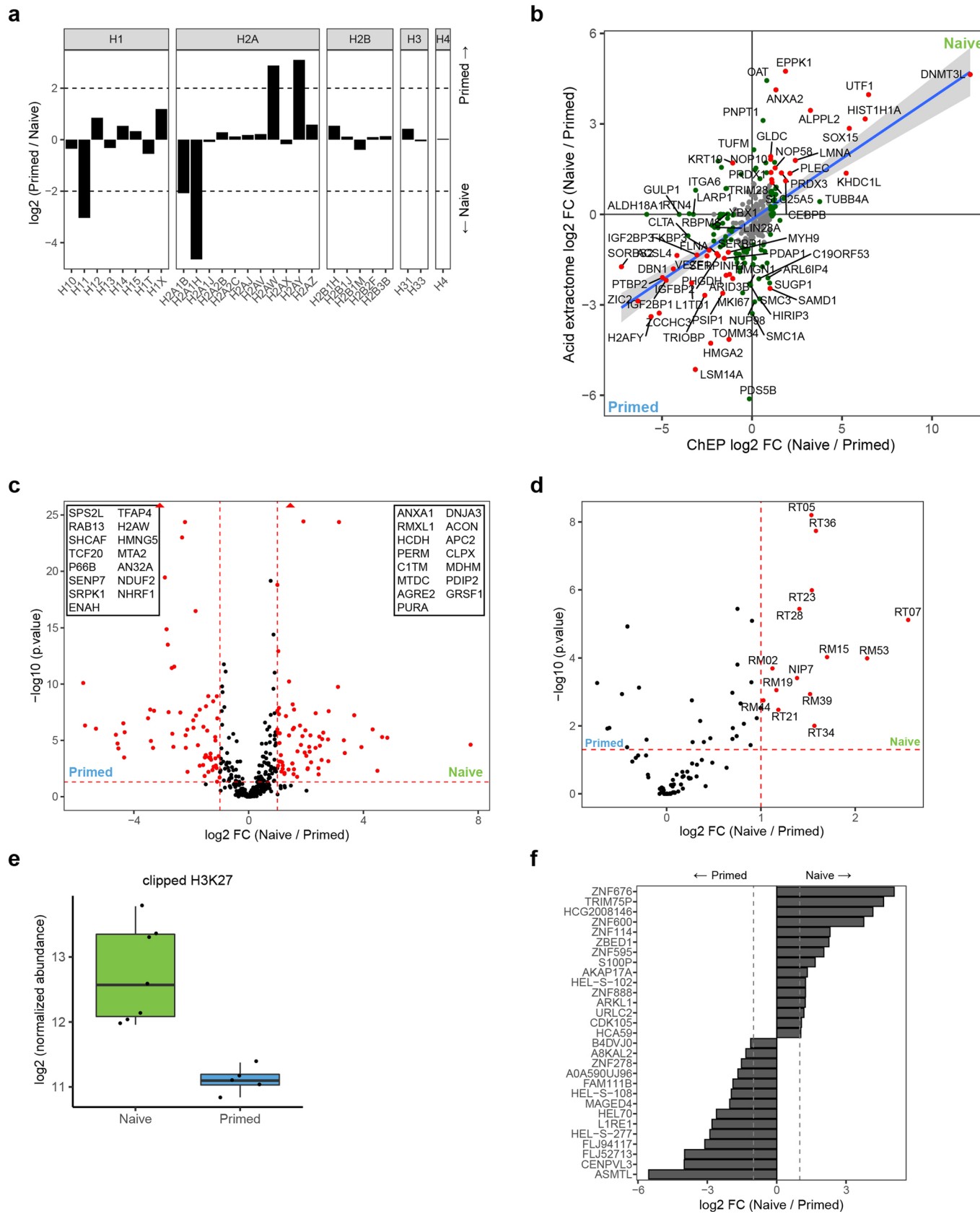

**Extended Data Fig. 2 | See next page for caption.**

**Extended Data Fig. 2 | Global analysis of acid extractome and hPTM clipping in naive and primed hPSCs. a**. Overview of histone variants identified in acid extractomes. Data are visualized as $\log_2$ transformed normalized expression of primed over naive hPSCs. Naive hPSCs, n = 7 biologically independent samples; primed hPSCs, n = 5 biologically independent samples. **b**. Comparison of proteins identified in the chromatin proteome (n = 4,576 proteins) and acid extractome (n = 894 proteins) in naive and primed hPSCs. Only proteins identified in both conditions were retained (n = 355 proteins). Proteins significantly changing (two-sided student's t test for chromatin proteome, moderated t test with Benjamini–Hochberg correction for acid extractome, p-value <0.05, > 2-fold change) in both datasets are indicated with red dots, while proteins significantly changing in only one dataset are highlighted in green dots. Strongest changing proteins (p-value < 0.05 & > 4-fold change in one dataset) are labelled by name. Blue line indicates best fit linear regression, while the shaded grey area indicates 95% confidence interval. **c**. Quantification of proteins uniquely identified in the acid extractome of naive and primed hPSCs (n = 539 proteins). Significantly changing proteins (moderated t test with Benjamini–Hochberg correction, p-value <0.05 & >2-fold change) are indicated with red dots. The 15 most strongly changing proteins for each pluripotent state are labelled by name. **d**. Quantification of ribosomal and nucleolar proteins identified in the acid extractome of naive and primed hPSCs (n = 128 proteins). Significantly changing proteins (moderated t test with Benjamini–Hochberg correction, p-value <0.05 & >2-fold change) are indicated with red dots and labelled by name. **e**. $\log_2$ transformed abundance of H3 tail clipping events (normalized against all histone peptidoforms) as identified by mass spectrometry. The boxplots show the interquartile range (box limits) and the median (centre line) of the abundance of clipping events. Naive hPSCs, n = 7 biologically independent samples; primed hPSCs, n = 5 biologically independent samples. **f**. Comparison of chromatin-associated proteins uniquely identified in human between naive and primed pluripotent states. The listed proteins have no known mouse ortholog or homologue. Underlying source data is provided in Source Data Extended Data Fig. 2.

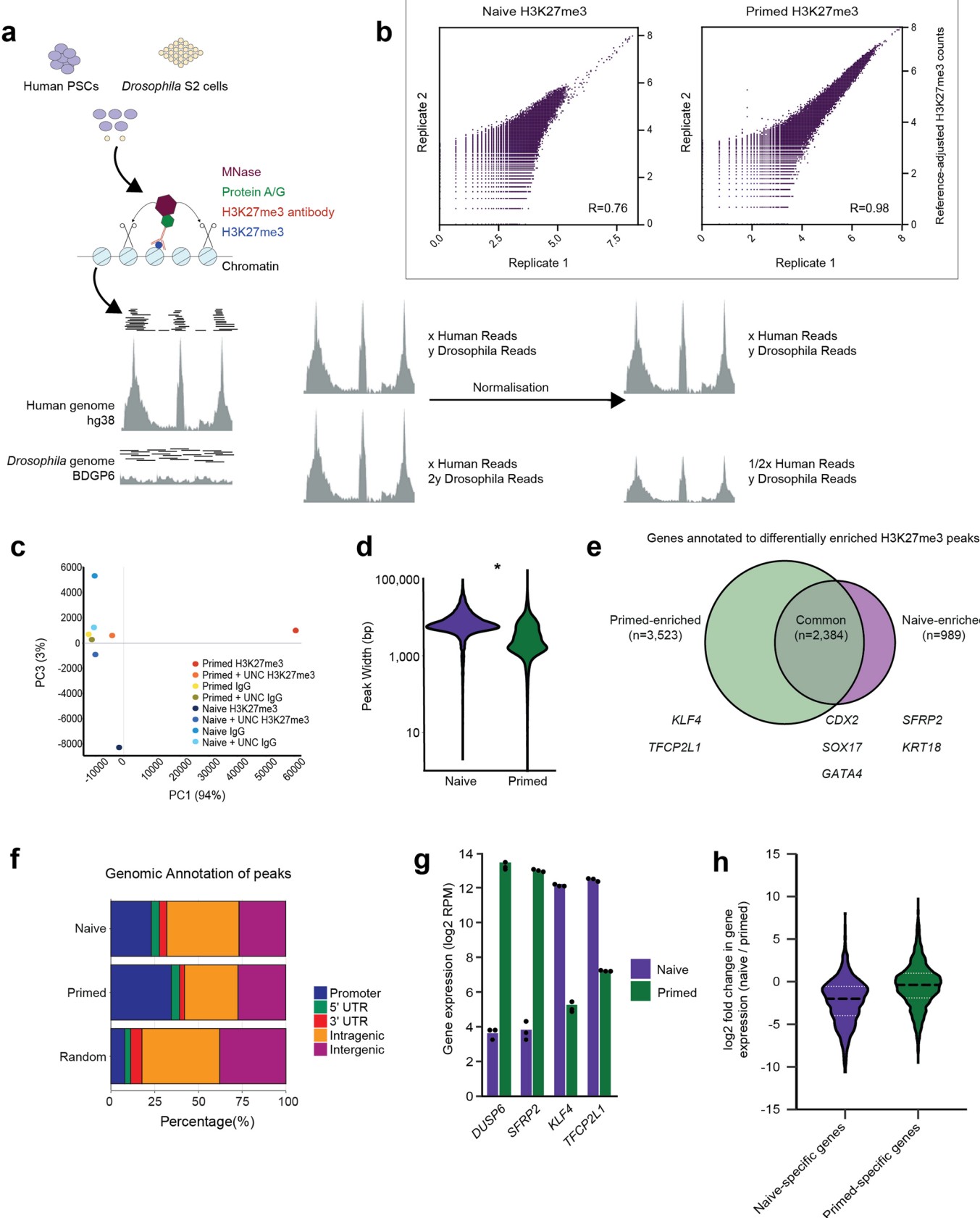

**Extended Data Fig. 3 | See next page for caption.**

**Extended Data Fig. 3 | Chromatin profiling of H3K27me3 in naive and primed hPSCs. a**. Schematic of the cCUT&RUN method and normalization strategy. **b**. Scatterplots comparing the log2 transformed H3K27me3 normalized read count across 1 kb windows for naive (left) and primed (right) hPSCs. Correlation *r* is determined by Pearson correlation prior to transformation. 2 biologically independent experiments for primed and naive H3K27me3 and IgG calibrated CUT&RUNs, with the exception of naive IgG cCUT&RUN which was performed once. **c**. Principal Component Analysis of normalized cCUT&RUN data. **d**. Violin plots showing the H3K27me3 peak width of normalized cCUT&RUN data in naive and primed hPSCs. The difference in width between primed and naive hPSCs is statistically significant, indicated by * (two-sided student's t test, $p < 2.2 \times 10^{-16}$; n = 10,187 peaks for naive PSCs and n = 17,626 peaks for primed PSCs). **e**. Venn diagram showing the extent of overlap of nearest genes (within 10 kb of promoter) to H3K27me3 peaks in naive and primed hPSCs, with example genes added for each category. **f**. Stacked bar plot describing genomic annotation of H3K27me3 peaks in untreated naive and primed hPSCs, compared to a background sample of 10,000 randomly generated peaks. **g**. Gene expression values of primed-specific (*DUSP6* and *SFRP2*) and naive-specific (*KLF4* and *TFCP2L1*) transcripts from bulk RNA-sequencing data. Individual data points from n = 3 biologically independent samples are shown, and the bar indicates the mean expression value. **h**. Violin plots show the fold change in gene expression between naive and primed hPSCs for naive-specific and primed-specific H3K27me3-marked genes from bulk RNA-sequencing data. The plots show the 25th and 75th quartiles (white dotted lines) and the median (black dashed lines). Underlying source data is provided in Source Data Extended Data Fig. 3.

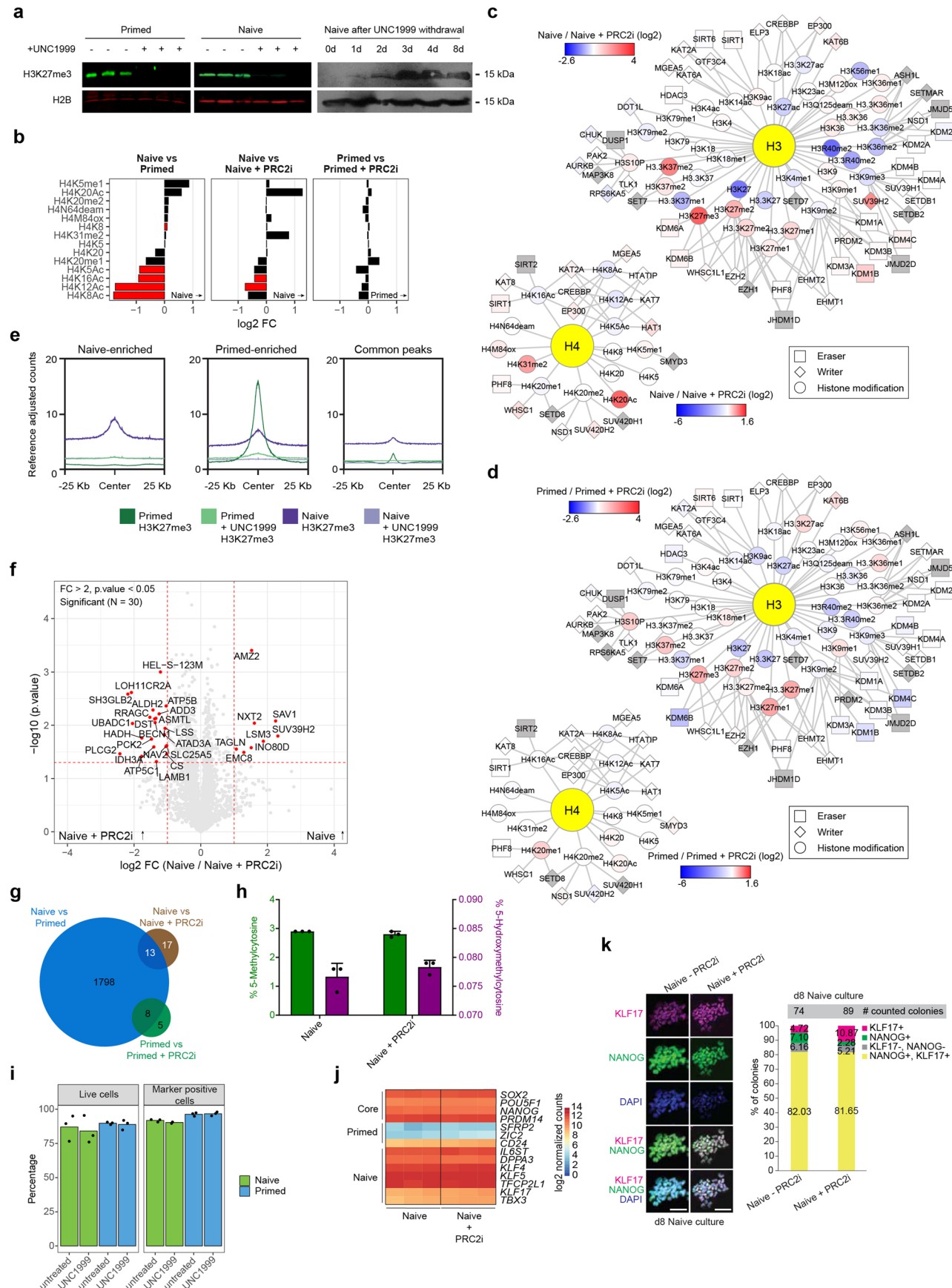

**Extended Data Fig. 4 | See next page for caption.**

**Extended Data Fig. 4 | Four-day PRC2 inhibition by UNC1999 efficiently removes H3K27me3, with minor alterations in chromatin proteome and hPTM landscape. a**. Western blot validation of H3K27me3 removal in naive and primed hPSCs upon treatment with UNC1999 (Data shown represent 3 biologically independent samples [AU: modified as "n = X" statement is retained for cases where statistics are derived]). **b**. Histone PTM quantification for H4 in naive and primed hPSCs, with and without PRC2i. Data are visualized as $\log_2$ fold changes between two conditions, which are listed on top of each panel. Data are ordered according to the first panel. Red bars indicate significantly changing hPTMs (two-sided student's t test with Benjamini–Hochberg correction, p-value < 0.05). Naive hPSCs, n = 7 biologically independent samples; primed hPSCs, n = 5 biologically independent samples; naive hPSCs + inhibitor n = 6 biologically independent samples; primed hPSCs + inhibitor, n = 8 biologically independent samples. **c-d**. Integration of the chromatin proteome and hPTM measurements for naive (**c**) and primed (**d**) hPSCs with and without PRC2i. Nodes represent chromatin modifiers and hPTMs, and are coloured by log2 fold change in abundance. Edges indicate functional connection (write or erase) between the nodes. Chromatin modifiers in grey nodes were not detected. **e**. Metaplots showing average profiles of normalized H3K27me3 reads across peaks, with relative abundance and distribution within 25 kb either side of the peak centre for primed-enriched, common and naive-enriched peaks in naive and primed hPSCs, with or without PRC2i. Two biologically independent experiments were used for primed and naive H3K27me3 and IgG cCUT&RUN experiments, both with and without PRC2i, with the exception of naive IgG cCUT&RUN which was performed once. Non-inhibitor treated samples are replicated from Extended Data Fig. 3. **f**. Specific changes in chromatin-associated proteins induced by PRC2i in naive hPSCs. Volcano plot of chromatin-associated proteins (n = 3,784 proteins) quantified in n = 3 biologically independent samples for naive hPSCs with or without PRC2i. Significantly changing (two-sided student's t test, p-value <0.05, >2-fold change) proteins are indicated with red dots and labelled by name. **g**. Numbers of significantly changing chromatin proteins (two-sided student's t test, p-value < 0.05 & >2-fold change) between naive and primed hPCSs and after PRC2i. N = 3 biologically independent samples. **h**. Global DNA methylation (green) and DNA hydroxymethylation (purple) levels in naive hPSCs, with and without PRC2i (n = 3 biologically independent samples). Data are presented as mean values +/- SD. **i**. Flow cytometry analysis of cell viability (left) and state-specific pluripotency markers (right) in naive and primed hPSCs treated with and without UNC1999 for four days. Cell viability was assessed using a live-dead dye, and the values shown represent the percentage of live cells in the total cell population. For the protein marker analysis, naive hPSCs were assayed for the expression of cell-surface markers SUSD2 and CD75, and primed hPSCs for the cell-surface markers SSEA4 and CD24. The values shown are the percentage of double-positive cells (SUSD2 and CD75, or SSEA4 and CD24) out of the total population of live cells. N = 3 biologically independent samples. **j**. Heatmap of normalized counts from RNA-seq of naive hPSCs with and without PRC2i for naive, primed and core pluripotency marker genes. Data are visualized as log2 normalized counts. N = 3 biologically independent samples. **k**. Immunofluorescence analysis for KLF17 (magenta), NANOG (green) and DAPI (blue) in naive hiPSCs after 8 days with or without PRC2i in PXGL medium. Right panel: quantification of KLF17 and/or NANOG positive colonies. Images are representative of 2 experiments. Scale bar = 100 mm. Underlying source data is provided in Source Data Extended Data Fig. 4.

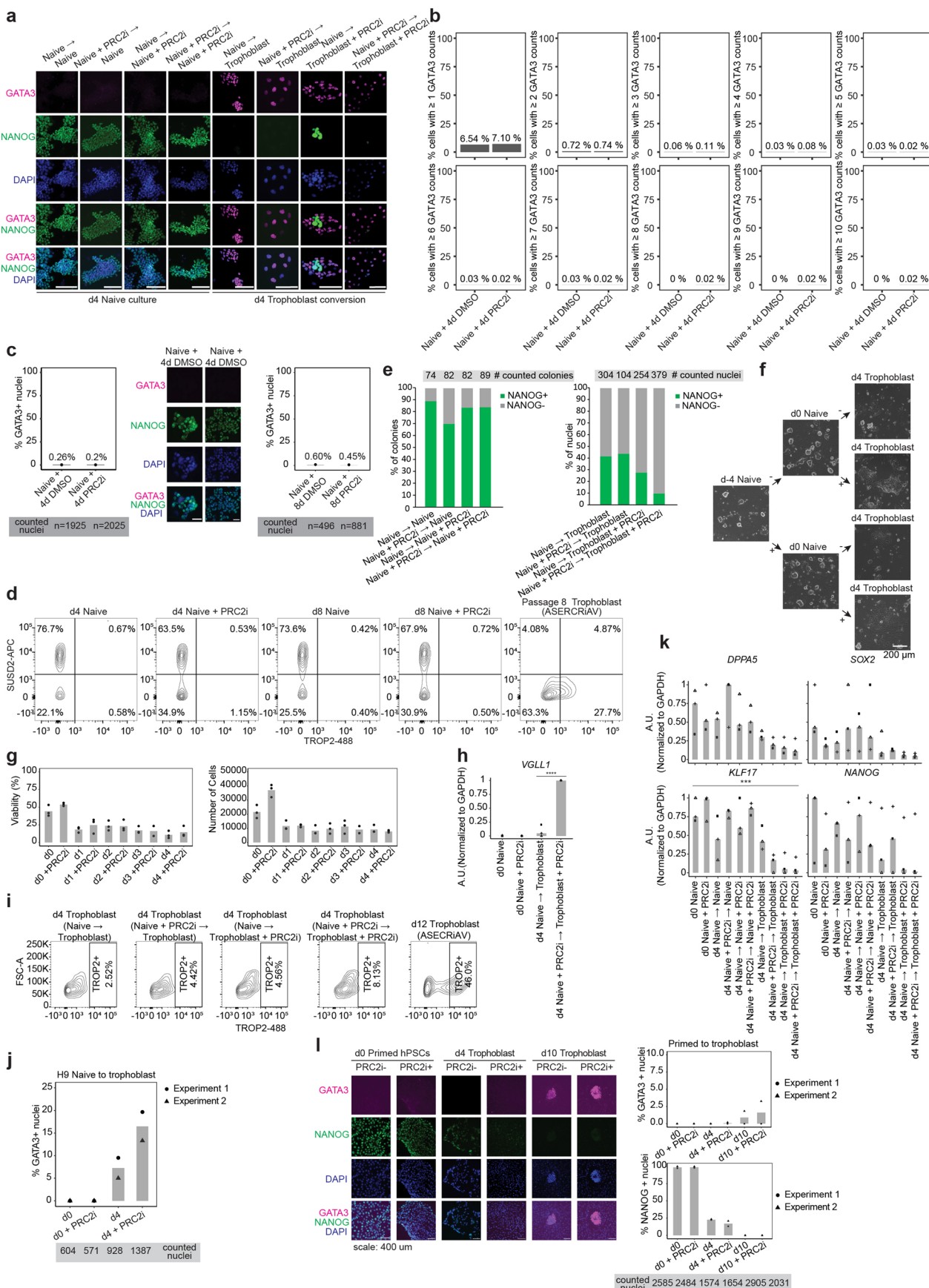

**Extended Data Fig. 5 | See next page for caption.**

**Extended Data Fig. 5 | PRC2 inhibition promotes naive to trophoblast induction. a**. Immunofluorescence analysis of trophoblast and pluripotency markers with or without PRC2 inhibition. Naive hiPSCs were stained for GATA3 (magenta), NANOG (green) and DAPI (blue) at day 4 of naive to trophoblast conversion with PRC2 inhibition (PRC2i) during the 4 days preceding trophoblast conversion only (Naive + PRC2i → Trophoblast), or during trophoblast conversion only (Naive → Trophoblast + PRC2i), or throughout the experiment (Naive + PRC2i → Trophoblast + PRC2i). Representative images from 3 experiments. Scale bar = 100 mm. **b**. Proportion of naive hiPSCs with GATA3 expression using different thresholds of scRNA-seq data counts to deem a cell GATA3 positive after 4 days of PRC2i in PXGL or in DMSO control conditions. **c**. Immunofluorescence analysis of naive hPSC with and without PRC2i. Cells were stained for GATA3 and DAPI at day 4 (H9 hPSCs; n = 1,925 cells for day 4 Naïve; n = 2,025 cells for day 4 Naive + PRC2i, in two experiments) and day 8 (ICSIG-1 hiPSCs; n = 496 cells for day 8 Naïve; n = 881 cells for day 8 Naive + PRC2i, in one experiment). Middle panel shows representative images for day 4 (H9 hPSCs). Left and right panels show quantification of GATA3 + nuclei for day 4 and day 8, respectively. Scale bar = 200 mm. **d**. Flow cytometry analysis of TROP2 and SUSD2 in naive hiPSCs at 4 and 8 days with or without PRC2i treatment. This experiment was performed once. **e**. Quantification of the proportion of colonies expressing or lacking NANOG protein expression in 2 experiments as evaluated by immunofluorescence. The number of colonies counted is shown on top of the panels. **f**. Phase contrast images of representative cells at day -4 (d -4), 0 (d0) and 4 (d4) of naive hiPSC to trophoblast conversion. Representative images of 3 experiments. **g**. Cell viability (left) and cell number (right) during naive hiPSC to trophoblast conversion with and without PRC2i. Cells have been pretreated for 4 days with PRC2i. N = 3 biologically independent samples. **h**. RT-qPCR assay for trophoblast marker *VGLL1*. Expression was normalized to *GAPDH*. Squares, triangles and circles represent n = 3 biologically independent samples, each with averaged biological triplicates or duplicates in each experiment. A.U. = arbitrary unit. Two-sided t test with Bonferroni adjustment, *p < 0.05, **p < 0.01, ***p < 0.001, ****p < 0.0001. **i**. Flow cytometry analysis of TROP2 and SUSD2 markers during naive hiPSC to trophoblast conversion with and without PRC2i treatment or pre-treatment at day 4. Day 12 trophoblast cells were included as a positive control. This experiment was performed once. **j**. Quantification of immunofluorescence analysis of H9 naive hPSC to trophoblast conversion with and without PRC2i from 2 experiments. Cells were stained for GATA3 at day 0 and day 4. **k**. RT-qPCR assay for pluripotency-associated genes, *DPPA5, SOX2, KLF17 and NANOG* (normalized to *GAPDH*) at day 0 and 4 of naive hiPSC to trophoblast conversion with and without PRC2 pre-treatment, treatment during trophoblast conversion alone, or PRC2i throughout the experiment. Squares, triangles and circles represent independent experiments (n = 3 experiments). For each experiment, the average of 3 or 2 biologically independent samples is shown. A.U. = arbitrary unit. Two-sided t-test with Bonferroni adjustment, *p < 0.05, **p < 0.01, ***p < 0.001, ****p < 0.0001. **l**. Immunofluorescence analysis of primed hPSC to trophoblast conversion with and without PRC2i. Cells were stained for GATA3, NANOG and DAPI at day 0, day 4 and day 10. Right panel shows quantification of GATA3 + and NANOG + nuclei. Representative images of 2 experiments. Underlying source data is provided in Source Data Extended Data Fig. 5.

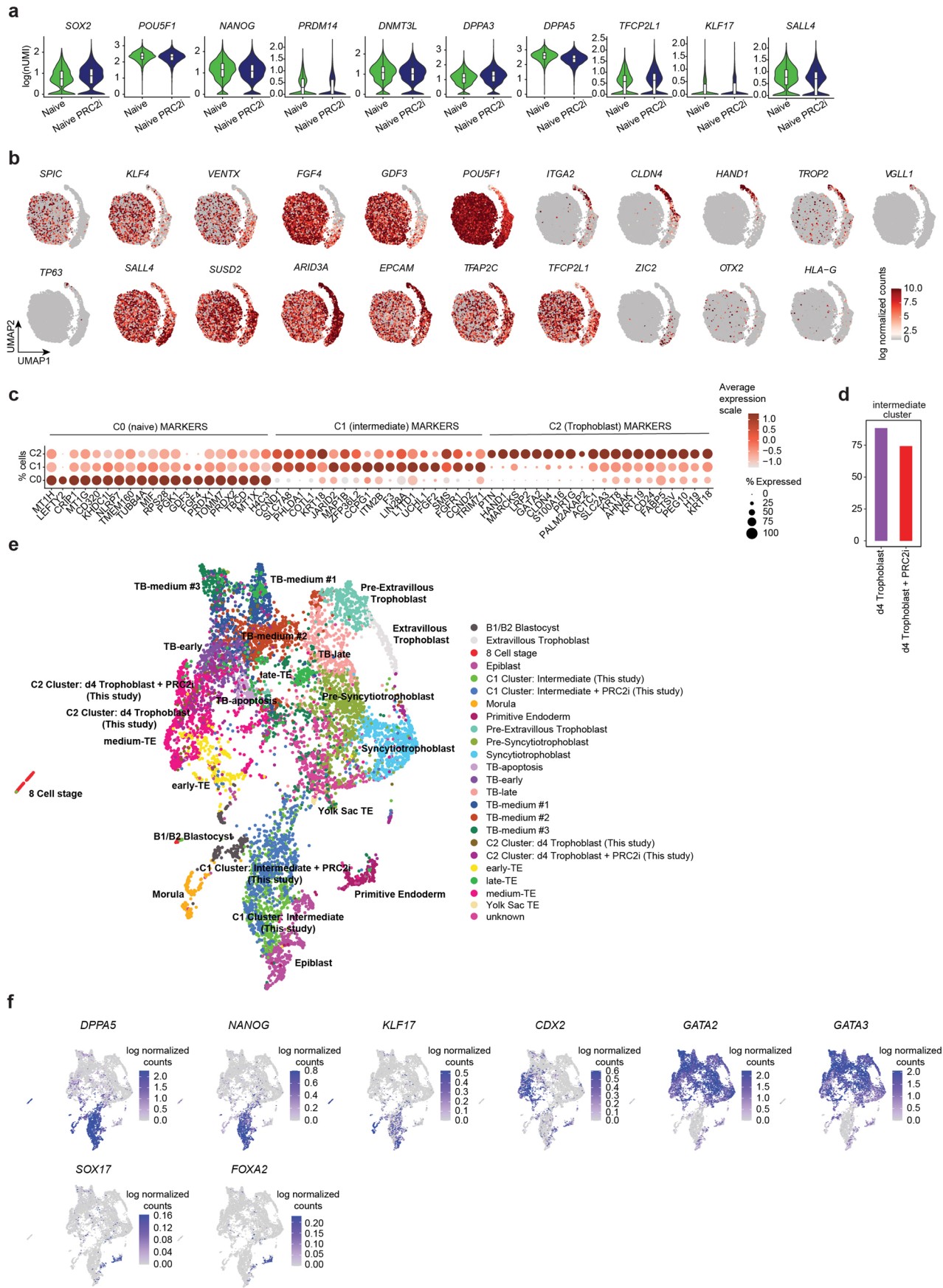

**Extended Data Fig. 6 | See next page for caption.**

**Extended Data Fig. 6 | Integration with human embryo data. a**. Violin plots with single cell expression distributions combined with boxplots in naive hiPSCs with and without PRC2i of pluripotency-associated genes. Data are visualized as log(nUMI). The boxplots show the interquartile range (box limits) and median (centre line) of gene expression levels. Number of single cells measured: n = 2,903 cells for the naive sample, and n = 3,338 cells for the naive + PRC2i sample. **b**. scRNA-seq analysis of pluripotency-associated and trophoblast-associated genes (UMAPs). Data are visualized as log normalized counts. Darker red intensity represents higher levels of gene expression, while lower red represents lower gene expression levels. **c**. scRNA-seq analysis showing the 20 most differentially expressed genes between the naive, intermediate and trophoblast cell clusters. Point size represents the proportion of cells in the cluster with the indicated gene expression enrichment. Data are visualized as average expression scale. Darker red intensity represents higher levels of gene expression, while softer red represents lower gene expression levels. **d**. Proportion of day 4 converted cells with intermediate cell identity. Purple indicates the day 4 trophoblast and red indicates the day 4 + PRC2i samples. **e**. Single-cell UMAP representation comparing in vitro day 4 trophoblast and day 4 trophoblast + PRC2i with human pre-implantation[3] and postimplantation[74] by data integration. Annotations from[13]. **f**. Single-cell UMAP representation of pluripotency, trophoblast and primitive endoderm marker genes from data in **e**. Data are visualized as log normalized counts. Underlying source data is provided in Source Data Extended Data Fig. 6.

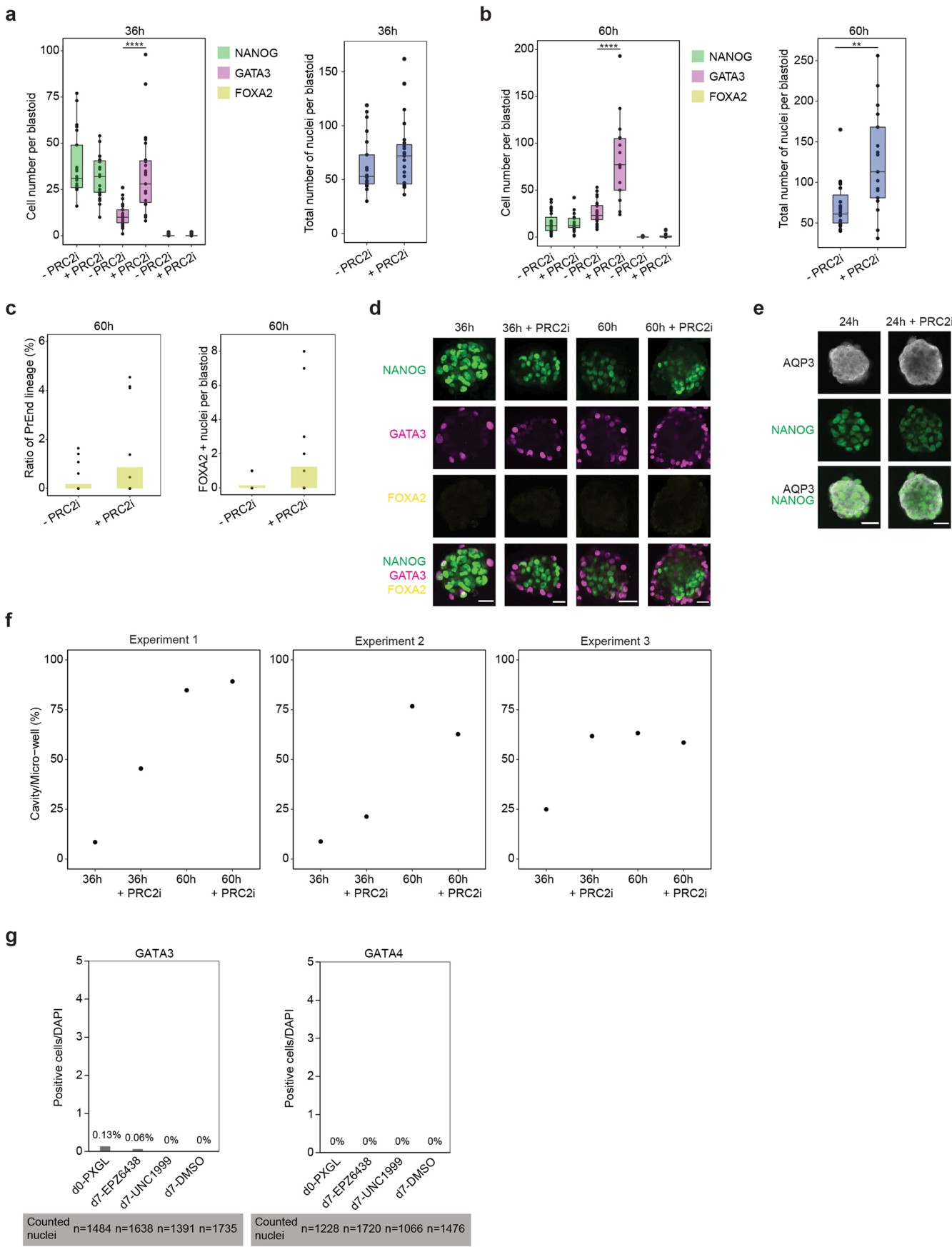

**Extended Data Fig. 7 | See next page for caption.**

**Extended Data Fig. 7 | Human blastoids. a–b**. Boxplots with the immunofluorescence quantification at 36 h (**a**) and 60 h (**b**) blastoids with or without PRC2i. Blastoids were stained for NANOG, GATA3 and FOXA2. The boxplots show the interquartile range (box limits) and median (centre line) of the total number of positive cells per blastoid (left panel). The right panel indicates the total number of cells per blastoid. N = 21 blastoids for 36 h -PRC2i blastoids; n = 23 blastoids for 36 h + PRC2i; n = 27 blastoids for 60 h -PRC2i; and n = 17 blastoids for 60 h + PRC2i, quantified from n = 1 experiment. A two-sided Wilcoxon rank-sum test with Bonferroni correction was used for significance testing. 36 h: ****$P = 1 \times 10\text{-}6$; 60 h: ****$P = 3.9 \times 10\text{-}5$, **$P = 1.6 \times 10\text{-}3$. **c**. Close-up for average number of FOXA2 + cells as found in (**b**). N = 27 blastoids for -PRC2i, and n = 17 blastoids for +PRC2i, quantified from n = 1 experiment. PrEnd = Primitive Endoderm. **d**. Representative immunofluorescence images for Fig. 7c and Extended Data Fig. 7a-b. NANOG is shown in green, GATA3 in magenta and FOXA2 in yellow. Scale bar: 200 mm. The experiment was performed once. **e**. Immunofluorescence analysis of human blastoids with and without PRC2i. Cells were stained for AQP3 (white) and NANOG (green) after 24 h. Representative image from 1 experiment. Scale bar: 200 mm. **f**. Quantification of cavitated human blastoids after 36 and 60 h with and without PRC2i. N = 3 biologically independent samples.

**g**. Quantification of immunofluorescence in naïve human pluripotent stem cells cultured in PXGL treated with PRC2i EPZ-6438 or UNC1999 for 7 days. Cells were stained for GATA3 and GATA4. The experiment was performed once. Underlying source data is provided in Source Data Extended Data Fig. 7.

Peter J. Rugg-Gunn
Hendrik Marks
Maarten Dhaenens

# Reporting Summary

## Statistics

For all statistical analyses, confirm that the following items are present in the figure legend, table legend, main text, or Methods section.

| n/a | Confirmed | |
|---|---|---|
| ☐ | ☒ | The exact sample size (*n*) for each experimental group/condition, given as a discrete number and unit of measurement |
| ☐ | ☒ | A statement on whether measurements were taken from distinct samples or whether the same sample was measured repeatedly |
| ☐ | ☒ | The statistical test(s) used AND whether they are one- or two-sided<br>*Only common tests should be described solely by name; describe more complex techniques in the Methods section.* |
| ☒ | ☐ | A description of all covariates tested |
| ☐ | ☒ | A description of any assumptions or corrections, such as tests of normality and adjustment for multiple comparisons |
| ☐ | ☒ | A full description of the statistical parameters including central tendency (e.g. means) or other basic estimates (e.g. regression coefficient) AND variation (e.g. standard deviation) or associated estimates of uncertainty (e.g. confidence intervals) |
| ☐ | ☒ | For null hypothesis testing, the test statistic (e.g. *F*, *t*, *r*) with confidence intervals, effect sizes, degrees of freedom and *P* value noted<br>*Give P values as exact values whenever suitable.* |
| ☒ | ☐ | For Bayesian analysis, information on the choice of priors and Markov chain Monte Carlo settings |
| ☒ | ☐ | For hierarchical and complex designs, identification of the appropriate level for tests and full reporting of outcomes |
| ☐ | ☒ | Estimates of effect sizes (e.g. Cohen's *d*, Pearson's *r*), indicating how they were calculated |

*Our web collection on statistics for biologists contains articles on many of the points above.*

## Software and code

Policy information about availability of computer code

| Data collection | CUT&RUN (SEACR) algorithm (v1.3) |
|---|---|
| Data analysis | FlowJo V10.1 software<br><br>Nikon NIS-Elements AR with DeNoise AI and 3D Richardson-Lucy algorithm (v5.30.01)<br>Axio Vision software (v4.9.1.0)<br>ImageJ (v1.37a)<br><br>Microsoft Excel (Microsoft Office Professional Plus 2016)<br>R (v4.0.3)<br>ggplot2 (v3.3.3)<br>FastQC (v0.11.8; Babraham Bioinformatics)<br>STAR (v2.7.1a)<br>featureCounts (v2.0.1)<br>DESeq2 (v1.26.0)<br>10X Genomics Cell Ranger (v4.0.0)<br>biomaRt (v2.64.3)<br>Seurat (v4.0.1)<br><br>TTrimGalore (v.0.6.6: Babraham Bioinformatics)<br>Bowtie2 (v.2.3.2)<br>DESEQ2 implementation within SeqMonk (v1.47.2; Babraham Bioinformatics)<br>ChIPseeker (v1.30.3) |

samtools (v1.11)
bedtools (v2.29.2)
deeptools (v3.43)
WashU epigenome browser (v5)
Seqmonk (v1.47.2: Babraham Bioinformatics)

MaxQuant (v1.6.6.0)
Perseus (v1.5.0.0)
Cytoscape (v3.7.2)

Progenesis QIP 4.2. (Nonlinear Dynamics, Waters)
Mascot Daemon (v2.6.0)
MSqRob (v0.7.6)

Codes pertaining to analyses in this study are available from GitHub webpages:
https://github.com/pasquelab/PRC2 (DOI: 10.5281/zenodo.6398543),
https://github.com/laurabiggins/Shiny_omics (DOI: 10.5281/zenodo.6400749), and
https://github.com/AndrewAMalcolm/Zijlmans-et-al.-2022 (DOI: 10.5281/zenodo.6399297).

For manuscripts utilizing custom algorithms or software that are central to the research but not yet described in published literature, software must be made available to editors and reviewers. We strongly encourage code deposition in a community repository (e.g. GitHub). See the Nature Portfolio guidelines for submitting code & software for further information.

# Data

Policy information about availability of data

All manuscripts must include a data availability statement. This statement should provide the following information, where applicable:
- Accession codes, unique identifiers, or web links for publicly available datasets
- A description of any restrictions on data availability
- For clinical datasets or third party data, please ensure that the statement adheres to our policy

Data generated during this study has been deposited in public repositories as follows:

The multi-omic data can be explored using the online resource: https://www.bioinformatics.babraham.ac.uk/shiny/shiny_omics/Shiny_omics
Figures 1, 2, 3, 4 and Extended Data Figures 1, 2, 3, 4.

Raw and processed sequencing data for RNA-seq, cCUT&RUN and scRNA-seq (including scRNA-seq loom files to be visualised on the SCope platform: https://scope.aertslab.org/#/HumanPluripotencyPRC2/*/welcome) have been submitted to the NCBI GEO (http://www.ncbi.nlm.nih.gov/geo/) under accession number GSE176175.
Figures 3, 4, 5, 6 and Extended Data Figures 3, 4, 5, 6.

Raw and processed sequencing data generated in this study (ChEP mass spectrometry proteomics data):
PRIDE partner repository, https://www.ebi.ac.uk/pride/
(dataset identifier PXD028111)
Figures 1, 2, 4 and Extended Data Figures 1, 2, 4.

Raw and processed sequencing data generated in this study (hPTM mass spectrometry proteomics):
PRIDE partner repository, https://www.ebi.ac.uk/pride/
(dataset identifier PXD028162, 10.6019/PXD028162 and PXD032792)
Figures 2, 4 and Extended Data Figures 2, 4.

Public databases/datasets used in this manuscript:
Human Swissprot database (https://www.uniprot.org/)
 cRAP database (https: //www.thegpm.org/crap/)
HIstome database (http://www3.iiserpune.ac.in/~coee/histome/)
UniProtKB human proteome (https://www.uniprot.org/)
ArrayExpress under the accession number E-MTAB-39293 (Petropoulos et al., 2016)
GEO database under accession number GSE10955589 (Zhou et al., 2019)
van Mierlo et al., 2019; PRIDE partner repository, https://www.ebi.ac.uk/pride/ (dataset identifier PXD007154).

DATA AVAILABILITY
The multi-omic data can be explored using the online resource: https://www.bioinformatics.babraham.ac.uk/shiny/shiny_omics/Shiny_omics. The scRNA-seq loom files can be visualised on the SCope platform: https://scope.aertslab.org/#/HumanPluripotencyPRC2/*/welcome.
RNA sequencing, cCUT&RUN, and single-cell RNA sequencing datasets have been deposited in the Gene Expression Omnibus (GEO) under the accession code of GSE176175.
hPTM mass spectrometry proteomics datasets have been deposited in the ProteomeXchange Consortium via the PRIDE partner repository (https://www.ebi.ac.uk/pride/) under the dataset identifiers PXD028162, 10.6019/PXD028162 and PXD032792. The project with identifier PXD028162 (consultable via ProteomeXchange) was licensed on a single run basis and is fully accessible and editable by the readership after free download of the Progenesis QIP 4.2 software (https://www.nonlinear.com/progenesis/qi-for-proteomics/). ChEP mass spectrometry proteomics datasets have been deposited in the ProteomeXchange Consortium via the PRIDE partner repository (https://www.ebi.ac.uk/pride/) under the dataset identifier PXD028111. Furthermore, datasets were downloaded as provided by Petropoulos et al. (ArrayExpress: E-MTAB-39293), Zhou et al. (Zhou et al., 2019) (GEO: GSE10955580) and van Mierlo et al. (GEO: GSE101675). Public databases used in this manuscript include Human Swissprot database (https://www.uniprot.org/),  cRAP database (https: //www.thegpm.org/crap/), HIstome database (http://www3.iiserpune.ac.in/~coee/histome/), UniProtKB human proteome (https://www.uniprot.org/). Source data are provided with this paper. All other data supporting the findings of this study are available from the corresponding authors on reasonable request.

# Field-specific reporting

Please select the one below that is the best fit for your research. If you are not sure, read the appropriate sections before making your selection.

☒ Life sciences      ☐ Behavioural & social sciences      ☐ Ecological, evolutionary & environmental sciences

For a reference copy of the document with all sections, see nature.com/documents/nr-reporting-summary-flat.pdf

# Life sciences study design

All studies must disclose on these points even when the disclosure is negative.

| | |
|---|---|
| Sample size | All sample size are indicated on the figure legends or in the manuscript.<br>In stem cells, qPCR, flow cytometry, western blotting and immunostaining, experiments were performed in two or three independent replicates unless stated in figure legends.<br>In blastoids experiments, flow cytometry and cavity formation experiments were performed in three independent experiments. Immunofluorescence in blastoids were performed in several blastoids from one independent experiment.<br>In bulk RNAseq, ChEP,  and cCUT&RUN experiments were performed in two or three biological replicates.<br>hPTM mass spectrometry was performed for at least five biological replicates per condition.<br>No statistical method was used to determine sample size. Sample sizes were chosen based on previous experience from similar studies. |
| Data exclusions | For ChEP, proteins flagged as 'reverse', 'potential contaminant' or 'only identified by site' were filtered from the final protein list. Biological triplicates were grouped to calculate differential proteins. Data were filtered for 3 valid values in at least 1 group. Missing values were imputed using default settings in Perseus, based on the assumption that they were not detected because they were under or close to detection limit.<br>For bulk RNAseq analysis samples were filtered to keep genes that had more than 1 count in at least 2 conditions.<br>For single-cell RNAseq analysis human cells were retained and mouse cells (MEFs) were filtered out by adjusting the number of counts per cell (nCount_RNA) and the number of mapped genes per cell (nFeature_RNA) to only keep cells that were mostly mapped to the human GRCh38 (hg38) genome (for naive cells: nCount_RNA < 40000, nCount_RNA > 3000, nFeature_RNA < 8000 and nFeature_RNA > 1500; for day 4 of naive to trophoblast conversions: nCount_RNA < 300000, nCount_RNA > 10000, nFeature_RNA < 12000 and nFeature_RNA > 3000). Naive cells with more than 25% of mitochondrial counts were filtered out. Day 4 trophoblast converted cells with more than 30% of mitochondrial counts were filtered out.<br>For the hPTM analysis, outliers were removed first on the basis of the normalisation factor in Progenesis QIP 4.2: a normalisation factor less than 0,5 or more than 2 were filtered out. Secondly, outliers were removed based on the principal component analysis (PCA).<br>For calibrated CUT&RUN analysis, low quality reads with a MAPQ value > 20 were removed by filtering with samtools view.<br>All exclusion criteria used in this manuscript were pre-established. |
| Replication | All of the main experiments listed on this manuscript have been successfully replicated at least twice as independent experiments. When this is not the case, this has been indicated in figure legends. |
| Randomization | For scRNA-seq we used canonical correspondance analysis to remove potential technical/sample-to-sample effects and variation.<br>For the mass spectrometry experiments, all samples were run in a randomized fashion. Sample allocation was random in all other experiments |
| Blinding | ChEP, Acid Extractome, RNA-seq, Western blot, scRNA-seq, imaging<br>Samples from naive versus prime hPSCs, untreated or treated with UNC1999 inhibitor, were used, hence the analyses could not be performed with blinding. No blinding was performed because none of the analyses reported involved procedures that could be influenced by investigator bias.<br><br>Imaging.<br>Quantification of immunostaining experiments were performed by different researchers to confirm scoring.<br><br>For blastoids lineage analysis, counting of nuclei was performed blindly using automated nuclei scoring based on fluorescence using NIS-Elements AR 5.30.01 via a GA3 script. |

# Reporting for specific materials, systems and methods

We require information from authors about some types of materials, experimental systems and methods used in many studies. Here, indicate whether each material, system or method listed is relevant to your study. If you are not sure if a list item applies to your research, read the appropriate section before selecting a response.

## Materials & experimental systems

| n/a | Involved in the study |
|---|---|
| ☐ | ☒ Antibodies |
| ☐ | ☒ Eukaryotic cell lines |
| ☒ | ☐ Palaeontology and archaeology |
| ☐ | ☒ Animals and other organisms |
| ☒ | ☐ Human research participants |
| ☒ | ☐ Clinical data |
| ☒ | ☐ Dual use research of concern |

## Methods

| n/a | Involved in the study |
|---|---|
| ☐ | ☒ ChIP-seq |
| ☐ | ☒ Flow cytometry |
| ☒ | ☐ MRI-based neuroimaging |

# Antibodies

| | |
|---|---|
| Antibodies used | Immunostaining:<br>Anti-NANOG Mouse, 1 in 50, BD Pharmingen, 560482 (clone N31-355); Anti-mouse-Alexa Fluor 488; 555; 647 Donkey, 1 in 500, Invitrogen, A21202; A31570; A31571.<br>Anti-KLF17 Rabbit, 1 in 333, Abcam, HPA0246329; Anti-rabbit-Alexa Fluor 647 Donkey, 1 in 500, Invitrogen, A31573<br>Anti-GATA3 Rabbit, 1 in 250, Abcam, ab199428 [clone EPR16651]; Anti-rabbit-Alexa Fluor 647 Donkey, 1 in 500, Invitrogen, A31573<br>Anti-GATA3 Rat, 1 in 100, Thermo Fisher, 14-9966-82 (clone TWAJ); Anti-rat-Alexa Fluor 488 Donkey, 1 in 500, Invitrogen, A21208<br>Anti-AQP3 Rabbit, 1 in 200, Antibodies-online.com, ABIN863208; Anti-rabbit-Alexa Fluor 488 Donkey, 1 in 500, Invitrogen, A21206<br>Anti-FOXA2 Goat, 1 in 40, r&D Systems biotechne, AF2400; Anti-goat-Alexa 546 Donkey, 1 in 500, Invitrogen, A11056<br>Ant-GATA4 Rat, 1 in 400, eBioscience, 14-9980-82 (clone eBioEvan); Anti-rat-Alexa Fluor 488 Donkey, 1 in 500, Invitrogen, A21208<br><br>Flow Cytometry:<br>Anti-TROP2 Mouse, 1 in 100, R&D Systems, MAB650 (clone 77220); Anti-mouse-Alexa Fluor 488 Donkey, 1 in 500, Invitrogen, A21202<br>Anti-TROP2-488 Mouse, 1 in 25, R&D systems, FAB650G (clone 77220)<br>Anti-SUSD2-APC Mouse, 1 in 50, Miltenyi Biotec, 130-121-134 (clone W5C5)<br>Anti-SUSD2-PE Mouse, 1 in 100, Miltenyi Biotec, 130-111-641 (clone REA795)<br>Anti-CD75-eF660 Mouse, 1 in 20, eBioscience, 15519896 (clone LN-1)<br>Anti-SSEA4-APC Mouse, 1 in 40, R&D Systems, FAB1435A (clone MC-813-70)<br>Anti-CD24-BUV395 Mouse, 1 in 40, BD Bioscience, 563818 (clone ML5)<br><br>Calibrated CUT&RUN:<br>Tri-Methyl-Histone H3 Lysine 27 Rabbit, 1 in 50, Cell Signaling Technology, 9733 (clone C36B11)<br>Anti-IgG Rabbit, 1 in 50, Invitrogen, 31188<br>Anti-Histone H2Av spike-in antibody, 1 in 50, Active Motif, 61686<br>Western Blotting:<br>Anti-Tri-Methyl-Histone H3 Lysine 27 Rabbit, 1 in 1000, Cell Signaling Technology, 9733 (clone C36B11); Anti-Rabbit IgG (H+L)-DyLight-800 Donkey, 1 in 10000, Invitrogen, SA5-10044<br> Anti-Rabbit IgG (H+L)-HRP Goat, 1 in 10000, Bio-Rad, 1706515<br>Western Blotting Anti-Histone H2B Mouse, 1 in 1000, Abcam, ab64165 (clone mAbcam 64165); Anti-Mouse IgG (H+L)-DyLight-680 Donkey, 1 in 10000, Invitrogen, SA5-10170<br> Anti-Mouse IgG (H+L)-HRP Goat, 1 in 10000, Bio-Rad, 1706516 |
| Validation | Immunostaining:<br>Anti-NANOG Mouse, 1 in 400, BD Pharmingen, 560482 (clone N31-355)<br>Anti-mouse-Alexa Fluor 488; 555; 647 Donkey, 1 in 500, Invitrogen, A21202; A31570; A31571<br>Anti-KLF17 Rabbit, 1 in 200-1 in 500, Abcam, HPA0246329<br>Anti-rabbit-Alexa Fluor 647 Donkey, 1 in 200-1 in 2000, Invitrogen, A31573<br>Anti-GATA3 Rabbit, 1 in 250, Abcam, ab199428 [clone EPR16651]<br>Anti-GATA3 Rat, 1 in 100, Thermo Fisher, 14-9966-82 (clone TWAJ)<br>Anti-rat-Alexa Fluor 488 Donkey, 1 in 2000, Invitrogen, A21208<br>Anti-AQP3 Rabbit, 1 in 200, Antibodies-online.com, ABIN863208<br>Anti-rabbit-Alexa Fluor 488 Donkey, 1 in 1000, Invitrogen, A21206<br>Anti-FOXA2 Goat, 1 in 13,3-1 in 40, r&D Systems biotechne, AF2400<br>Anti-goat-Alexa 546 Donkey, 1 in 200-1 in 2000, Invitrogen, A11056<br>Ant-GATA4 Rat, 1 in 200-1 in 400, eBioscience, 14-9980-82 (clone eBioEvan)<br><br>Flow Cytometry:<br>Anti-TROP2 Mouse, 0.25 µg/10^6 cells, R&D Systems, MAB650 (clone 77220)<br>Anti-mouse-Alexa Fluor 488 Donkey, no validation for Flow Cytometry, Invitrogen, A21202<br>Anti-TROP2-488 Mouse, 10 µL/10^6 cells, R&D systems, FAB650G (clone 77220)<br>Anti-SUSD2-APC Mouse, 1 in 50, Miltenyi Biotec, 130-121-134 (clone W5C5)<br>Anti-SUSD2-PE Mouse, 1 in 50, Miltenyi Biotec, 130-111-641 (clone REA795)<br>Anti-CD75-eF660 Mouse, 5µl/Test, eBioscience, 15519896 (clone LN-1)<br>Anti-SSEA4-APC Mouse, 10 µL/10^6 cells, R&D Systems, FAB1435A (clone MC-813-70)<br>Anti-CD24-BUV395 Mouse, 5µl/Test, BD Bioscience, 563818 (clone ML5)<br><br>Calibrated CUT&RUN:<br>Tri-Methyl-Histone H3 Lysine 27 Rabbit, 1 in 50, Cell Signaling Technology, 9733 (clone C36B11) |

Anti-IgG Rabbit, no validation for CUT&RUN, Invitrogen, 31188
Anti-Histone H2Av spike-in antibody, 1 in 50, Active Motif, 61686
Western Blotting:
Anti-Tri-Methyl-Histone H3 Lysine 27 Rabbit,  1 in 1000, Cell Signaling Technology, 9733 (clone C36B11)
Anti-Rabbit IgG (H+L)-DyLight-800 Donkey, 1 in 5000-1 in 20000, Invitrogen, SA5-10044
Anti-Rabbit IgG (H+L)-HRP Goat, 1 in 3000, Bio-Rad, 1706515
Western Blotting Anti-Histone H2B Mouse, 1 in 200-1 in 1000, Abcam, ab64165 (clone mAbcam 64165)
Anti-Mouse IgG (H+L)-DyLight-680 Donkey, 1 in 1000-1 in 20000, Invitrogen, SA5-10170
Anti-Mouse IgG (H+L)-HRP Goat, 1 in 10000, Bio-Rad, 1706516

## Eukaryotic cell lines

Policy information about cell lines

| Cell line source(s) | H9 hESCs (WiCell).<br>ICSIG-1 IPSC0028 hiPSCs (Sigma).<br>Wild-type IPSC0028 hiPSCs (Sigma).<br>Male mouse embryonic fibroblasts (MEFs) isolated from wild-type mouse B6 embryos.<br>Drosophila S2 cells (ThermoFisher Scientific) |
|---|---|
| Authentication | All cell lines in this study were authenticated via gene expression analysis. |
| Mycoplasma contamination | Periodic mycoplasma contamination testing was carried out and confirmed the absence of mycoplasma contamination. |
| Commonly misidentified lines<br>(See ICLAC register) | No commonly identified cell lines were used in this study. |

## Animals and other organisms

Policy information about studies involving animals; ARRIVE guidelines recommended for reporting animal research

| Laboratory animals | Wild-type male mouse embryos C57B6/J E14.5. Animals were kept with a maximum of 5 animals per cage and separated by sex.<br>Illumination was controlled on a 14h light, 10h dark light cycle from 7h to 21h .<br>Temperature was checked daily and should be 22±2°C<br>Humidity in mouse rooms was checked daily and should be between 45-70% but can vary with weather conditions, especially in winter. |
|---|---|
| Wild animals | No wild animals were used in this study. |
| Field-collected samples | No field-collected samples were used in this study. |
| Ethics oversight | Animal work carried out in this study was covered by project licences (ECD_P003-2016 and ECD_P170/2019 to V.P and to F.L, respectively) approved by the KU Leuven Animal Ethics Committee. |

Note that full information on the approval of the study protocol must also be provided in the manuscript.

## ChIP-seq

### Data deposition

☒ Confirm that both raw and final processed data have been deposited in a public database such as GEO.

☒ Confirm that you have deposited or provided access to graph files (e.g. BED files) for the called peaks.

| Data access links<br>*May remain private before publication.* | Calibrated CUT&RUN data provided in GEO (http://www.ncbi.nlm.nih.gov/geo/) under the accession number GSE176175.<br> http://ftp1.babraham.ac.uk/ftpusr79/ |
|---|---|
| Files in database submission | GSM5569915 Primed_H3K27me3_Replicate_1<br>GSM5569916 Primed_H3K27me3_Replicate_2<br>GSM5569917 Primed_UNC1999_H3K27me3_Replicate_1<br>GSM5569918 Primed_UNC1999_H3K27me3_Replicate_2<br>GSM5569919 Primed_IgG_Replicate_1<br>GSM5569920 Primed_IgG_Replicate_2<br>GSM5569921 Primed_UNC1999_IgG_Replicate_1<br>GSM5569922 Primed_UNC1999_IgG_Replicate_2<br>GSM5569923 Naïve_H3K27me3_Replicate_1<br>GSM5569924 Naïve_H3K27me3_Replicate_2<br>GSM5569925 Naïve_UNC1999_H3K27me3_Replicate_1<br>GSM5569926 Naïve_UNC1999_H3K27me3_Replicate_2<br>GSM5569927 Naïve_IgG_Replicate_1<br>GSM5569928 Naïve_UNC1999_IgG_Replicate_1<br>GSM5569929 Naïve_UNC1999_IgG_Replicate_2 |

| Genome browser session<br>(e.g. UCSC) | http://ftp1.babraham.ac.uk/ftpusr79/Zijlmans_CutandRun_0122_DataHub.json - openable as a remote data track URL in WashU genome browser. |
|---|---|

## Methodology

| Replicates | Two independent biological replicates for all samples except for naive IgG which has one biological replicate. Calibrated CUT&RUN performed for H3K27me3 for experimental samples and IgG, with an anti-Drosophila H2Av spike-in antibody, as a control used for peak calling. Strong replicate agreement between samples of the same condition - Pearson correlation coefficient of 0.76 between two naive H3K27me3 samples and 0.98 for two primed H3K27me3 samples. |
|---|---|
| Sequencing depth | Total number of sequencing reads: 483,205,175. Highest read count = 53,117,572, Lowest read count = 20,861,009, Average read count = 32,213,678. Uniquely mapping human reads = 275,974,550. Highest uniquely mapping read count = 35,268,030, Lowest uniquely mapping read count = 6,450,381. Average uniquely mapping read count = 18,398,303. Samples were sequenced as 150 bp paired-end sequencing. |
| Antibodies | Tri-Methyl-Histone H3 Lysine 27 - Cell Signaling Technology, 9733 - (C36B11) Rabbit mAb #9733<br>Rabbit anti-Mouse IgG (H+L) Secondary Antibody - Invitrogen, 31188 - # 31188<br>Anti-Histone H2Av spike-in antibody - Active Motif, 61686 - #61686 |
| Peak calling parameters | Peak calling was performed using the CUT&RUN optimised Sparse Enrichment Analysis for CUT&RUN (SEACR) algorithm, using H3K27me3 samples as the sample and their corresponding IgG as a control, against which peaks were called. The top 1% of peaks were retained (v1.3). Peaks closer than 300bp were merged using bedtools merge and peaks common to both replicates were determined by bedtools intersect to generate final peak sets for naive and primed. Peaks called in naive and primed hPSC data sets were concatenated into a combined peak list and deduplicated. Differentially enriched peaks between naive and primed hPSCs were then determined from this concatenated list using a DESEQ2 implementation within SeqMonk (v1.47.2; Babraham Bioinformatics) to identify differential regions with a p-value < 0.05 after Benjamini-Hochberg Multiple-Testing Correction. Common peaks were classified as peaks in the concatenated list that were not statistically-enriched in either condition. These peaks were then filtered against the ENCODE GRCh38 exclusion list to remove coverage outliers. |
| Data quality | Raw FastQ data were trimmed with Trim Galore and aligned to GRCh38 human genome or Drosophila BDGP6 genome using Bowtie2 with the following parameters –very-sensitive -I 10 -X700. High quality reads with a MAPQ value > 20 were retained by filtering with samtools view. Peak calling was performed using the Sparse Enrichment Analysis for CUT&RUN (SEACR) algorithm and the top 1% peaks were retained. Differentially enriched peaks were determined from a concatenated peak list using a DESEQ2 implementation within SeqMonk to identify differential regions with a p-value < 0.05 after Benjamini-Hochberg Multiple-Testing Correction to ensure a false discovery rate of less than 5%. |
| Software | Trim Galore (v.0.6.6: Babraham Bioinformatics)<br>Bowtie2 (v.2.3.2)<br>samtools (v1.11)<br>bedtools (v2.29.2)<br>Sparse Enrichment Analysis for CUT&RUN (SEACR) algorithm (v1.3)<br>DESEQ2 package within SeqMonk (v1.47.2; Babraham Bioinformatics)<br> deeptools API suite<br>Python (v3.7.3)<br>ChIPseeker package within R (v1.30.3:93)<br>HOMER<br>deeptools bamCoverage (v3.43)<br>UCSC-tools bigWigMerge<br>USCS-tools bedGraphToBigWig<br>WashU epigenome browser (v5)<br>deeptools computeMatrix<br>deeptools MultiBamSummary<br>deeptools plotCorrelation<br>ggplot2 package within R |

# Flow Cytometry

## Plots

Confirm that:

☒ The axis labels state the marker and fluorochrome used (e.g. CD4-FITC).

☒ The axis scales are clearly visible. Include numbers along axes only for bottom left plot of group (a 'group' is an analysis of identical markers).

☒ All plots are contour plots with outliers or pseudocolor plots.

☒ A numerical value for number of cells or percentage (with statistics) is provided.

## Methodology

| Sample preparation | For Figure 7, blastoids were harvested from the microwell arrays, sequentially treated with 300 U/ml collagenase type-IV and 10x Trypsin-EDTA (ThermoFisher) at 37C on a shaker. The blastoids were dissociated into single cells by pipetting. Cells were washed 3 times with flow buffer (1% FBS in PBS) and incubated with primary antibody diluted in flow buffer and incubated |
|---|---|

for 30 minutes at 4°C. The cells were centrifuged, washed with flow buffer twice, and combined with secondary antibody and incubated for 30 minutes at 4°C. The cells were centrifuged, washed twice with flow buffer, and resuspended in fresh flow buffer for flow cytometry analysis.

For Figure 4, naive and primed hPSCs were washed once with PBS and dissociated using Accutase (BioLegend) by incubation for 5 min at 37 °C. Accutase was quenched 1:1 with medium and cells were passed through a 50 μm cell strainer (VWR) and centrifuged at 300 x g for 3 minutes. Cell pellets were washed once with PBS containing 2% FBS (flow buffer) and counted. Fluorophore-conjugated antibodies and eF780 fixable viability dye (eBioscience, 65-0865-14) were mixed with 50 μL Brilliant stain buffer (BD Biosciences) and applied to 500,000 cells in 50 μL flow buffer. Labelling occurred for 30 min at 4 °C in the dark. Cells were washed twice with flow buffer and analysed.

For Figure 5, cells were dissociated using Accutase (5 min incubation at 37°C) and centrifuge at 1000 rpm for 5 min. Supernatant was removed and cell pellets were resuspended in 300 ul of FACS (PBS supplemented with 0,25-0,5% BSA) buffer per sample and centrifuged again in same conditions. Cell pellets were resuspended in 50 ul of FACS buffer and antibody incubations were carried out at 4°C for 30 min. Afterwards, cells were washed twice with 300 ul of FACS buffer and centrifuged (1000 rpm for 5 min). Supernatant was removed and pellet was resuspended in 300 ul of PBS with 4% PFA. Samples were analysed using a BD FACSCanto II Flow Cytometer.

| Instrument | BD FACSCanto II Flow Cytometer
BD LSR Fortessa |
| Software | FlowJo software |
| Cell population abundance | No cell populations were sorted in this work. |
| Gating strategy | Single-stained controls were used for compensation calculations and unstained cells were used in cytometer and gating setup. |

☒ Tick this box to confirm that a figure exemplifying the gating strategy is provided in the Supplementary Information.

