## [Peer Review File · Nature Cell Biology]

Peer Review Information

Journal: Nature Cell Biology

Manuscript Title: Integrated Multi-Omics Reveal Polycomb Repressive Complex 2 Restricts Human Trophoblast Induction

Corresponding author names: Maarten Dhaenens, Hendrik Marks, Peter J. Rugg-Gunn, Vincent Pasque

Reviewer Comments & Decisions:

Decision Letter, initial version:

Subject: Decision on Nature Cell Biology submission NCB-P46415A
Message:

*Please delete the link to your author homepage if you wish to forward this email to co-authors.

Dear Professor Pasque,

Your manuscript, "Polycomb Repressive Complex 2 shields naïve human pluripotent cells from trophoblast differentiation", has now been seen by 3 referees, who are experts in trophoblasts (referee 1) and polycomb complex and stem cells (referees 2 and 3) and. As you will see from their comments (attached below) they find this work of potential interest, but have raised substantial concerns, which in our view would need to be addressed with considerable revisions before we can consider publication in Nature Cell Biology.

Nature Cell Biology editors discuss the referee reports in detail within the editorial team, including the chief editor, to identify key referee points that should be addressed with priority, and requests that are overruled as being beyond the scope of the current study. To guide the scope of the revisions, I have listed these points below. We are committed to providing a fair and constructive peer-review process, so please feel free to contact me if you would like to discuss any of the referee comments further.

In particular, it would be essential to:

a) strengthen the major claim that PRC2-mediated H3K27me3 specifically restricts the trophoblast lineage in naïve ESCs, as noted by:

Reviewer #1:

-Fig. 5a-c: These are interesting and important experiments, but the figures suggest that PRC2i does not robustly induce trophoblast. For example, in Fig. 5c, naïve->trophoblast induced 33.5% GATA3 positive cells, but naïve PRC2i ->trophoblast induced only 29.5% GATA3 positive cells. If PRC2 functions as a barrier of trophoblast formation in naïve hPSCs, the naïve PSCs with PRC2i treatment from day -4 to 0 should differentiate trophoblast more efficiently than normal naïve PSCs. I suggest the authors evaluate

2the efficiency by flowcytometry. I wonder if many naïve PSCs die during the trophoblast induction in this manuscript. The cell number during induction is also important information.

Reviewer #3:

That said, some of the interpretation needs to be adjusted or further strengthened. The major claim is the specific restriction of the trophoblast lineage via PRC2 mediated H3K27me3 (title, abstract, main text and model in Figure 6g), however the data do not support this as currently presented. PRC2 clearly regulates trophoblast genes, but also many other loci in naïve and primed cells as well as subsequent lineages. Hence it is not clear on what the specific claim towards trophoblast restriction is based. The specific assignment in Figure 6g of PRC2 as a result of the data shown seems not fitting. H3K27me levels decrease globally (as indicated in 6g) but the number of targeted loci notably increase, which is again something crucial that is not represented in the current simplified model.

Specific comments:

The authors state: 'Importantly, we observed that genes associated with embryonic and extraembryonic lineage specification were marked by H3K27me3 in naive hPSCs, similar to primed hPSCs (Figure 3G).'

As noted above this doesn't match the major claim.

Then they continue: 'The presence of H3K27me3 at the promoters of key trophoblast regulators, therefore, raises the possibility that PRC2-mediated H3K27me3 might restrict the trophoblast specification programme in naive hPSCs.'

Figure 3g shows clearly that PRC2 mediated H3K27me3 is not specific to extraembryonic nor naïve cells, and in fact its slightly lower in naïve.

It needs to be clarified and shown what supports the authors assertions of a specific role rather than the expected general repression of ALL lineages?

Along those lines, the effect of PRC2 inhibition (EZHi) is only 2-fold (and only with the TSC induction conditions) and hence unclear if that supports a major role of PRC2 repression in that lineage.

b) analyze the scRNA-seq data more thoroughly, as noted by Reviewer #2:

As presented the single cell analysis doesn't add much beyond presenting nice data as a resource. More should be done here or it could be left out in favor of more detailed analysis at the chromatin level including more functional investigation.

c) All other referee concerns pertaining to strengthening existing data, providing controls, methodological details, clarifications and textual changes, should also be addressed.

d) Finally please pay close attention to our guidelines on statistical and methodological reporting (listed below) as failure to do so may delay the reconsideration of the revised manuscript. In particular please provide:

We would be happy to consider a revised manuscript that would satisfactorily address these points, unless a similar paper is published elsewhere, or is accepted for publication in Nature Cell Biology in the meantime.

- ensure that it conforms to our format instructions and publication policies (see below and <https://www.nature.com/nature/for-authors>).

- provide a point-by-point rebuttal to the full referee reports verbatim, as provided at the end of this letter.

- provide the completed Reporting Summary (found here <https://www.nature.com/documents/nr-reporting-summary.pdf>). This is essential for reconsideration of the manuscript will be available to editors and referees in the event of peer review. For more information see <http://www.nature.com/authors/policies/availability.html> or contact me.

When submitting the revised version of your manuscript, please pay close attention to our [Digital Image Integrity Guidelines](https://www.nature.com/nature-research/editorial-policies/image-integrity). and to the following points below:

Nature Cell Biology is committed to improving transparency in authorship. As part of our efforts in this direction, we are now requesting that all authors identified as 'corresponding author' on published papers create and link their Open Researcher and Contributor Identifier (ORCID) with their account on the Manuscript Tracking System (MTS), prior to acceptance. ORCID helps the scientific community achieve unambiguous attribution of all scholarly contributions. You can create and link your ORCID from the home page of the MTS by clicking on 'Modify my Springer Nature account'. For more information please visit www.springernature.com/orcid.

This journal strongly supports public availability of data. Please place the data used in your paper into a public data repository, or alternatively, present the data as Supplementary Information. If data can only be shared on request, please explain why in your Data Availability Statement, and also in the correspondence with your editor. Please note that for some data types, deposition in a public repository is mandatory - more information on our data deposition policies and available repositories appears below.

[REDACTED]

We would like to receive a revised submission within six months.

We hope that you will find our referees' comments, and editorial guidance helpful. Please do not hesitate to contact me if there is anything you would like to discuss.

Best wishes,

Jie Wang

Jie Wang, PhD
Senior Editor
Nature Cell Biology

Tel: +44 (0) 207 843 4924
email: jie.wang@nature.com

Reviewers' Comments:

Reviewer #1:

Remarks to the Author:

In their manuscript, Zijlmans et al. examined the wide lineage propensity of naïve human pluripotent stem cells by investigating chromatin modifications. By analyzing proteins bound to chromatin in naïve and primed H9 ES cells by ChEP, they found core components of the PRC2 complex, including EED, EZH2, and SUZ12, in naïve PSCs were upregulated. An analysis of 43 histone post-translational modifications revealed that 23 of them were different between naïve and primed PSCs. Notably, PRC2-mediated H3K27me_{2/3} was higher in naïve PSCs than in primed PSCs. The level of H3K27me_{2/3} erasers KDM6A/B was also higher in naïve PSCs. cCUT&RUN confirmed that H3K27me₃ in naïve PSCs is globally higher than in primed PSCs but that there are more specific H3K27me₃ peaks in primed PSCs. Interestingly, the authors observed peaks of H3K27me₃ in trophoblast markers, including HAND1, CDX2, GATA2, GATA3 and KRT8/18, both in naïve and primed PSCs, and a PRC2 inhibitor reduced H3K27me levels. From these

5observations, the authors concluded that PRC2 functions as a barrier of trophoblast formation in naïve PSCs. This is a novel and important finding. However, prior to publication, I have the following comments about authors claims that I ask they resolve.

- Although the authors showed common H3K27me3 peaks in Fig. 4b, 4c and Fig. S4f, 4g after PRC2 inhibitor treatment, there are no data about primed and naïve specific peaks. Primed and naïve specific peaks also need to be shown. In addition, the authors should show primed peaks in Fig 4d.

- How many days did the authors culture naïve cells in Fig 4i-l. Fig. 4l and S4k show very different gene expression profiles of naïve and primed PSCs after PRC2i treatment. The authors should provide a reason and show the data of primed PSCs in Fig. 4i-k.

-Fig. 5a-c: These are interesting and important experiments, but the figures suggest that PRC2i does not robustly induce trophoblast. For example, in Fig. 5c, naïve->trophoblast induced 33.5% GATA3 positive cells, but naïve PRC2i->trophoblast induced only 29.5% GATA3 positive cells. If PRC2 functions as a barrier of trophoblast formation in naïve hPSCs, the naïve PSCs with PRC2i treatment from day -4 to 0 should differentiate trophoblast more efficiently than normal naïve PSCs. I suggest the authors evaluate the efficiency by flowcytometry. I wonder if many naïve PSCs die during the trophoblast induction in this manuscript. The cell number during induction is also important information.

-To evaluate the differentiated cells (trophectoderm), scRNA-seq data should be compared with human embryo data by tSNE or UMAP.

-How long does the effect of PRC2i continue? Does H3K27me3 continue to decrease after the withdrawal of PRC2i?

-Fig. 6: In L374-377, the authors refer to improved conditions for blastoid formation. However, I could not find the method, which prevents judgement of the claim. Blastoids should be evaluated by immunofluorescence and scRNA-seq. Does PRC2i affect only trophoblast lineage in blastoids? The authors should count the number of blastoids+/-PRC2i cells in epiblast, hypoblast, and trophoblast. AQP3 should be stained in blastoids+/-PRC2i at 24 hours.

-Other ES or iPS cell lines should be checked.

Minor points

-Fig. S2b: A Supplemental Table showing the expression level of each gene is recommended.

- Fig. 6c and 6f do not show N or error bars.

Reviewer #2:

Remarks to the Author:

In this manuscript, Zijlmans and collaborators aim to challenge the current assumption that human naïve pluripotent cells are “epigenetically unrestricted”. For that, the authors perform a massive multi-omic approach at the proteomic and genomic levels. By integrating these data, the authors postulate a potential role of the Polycomb Repressive Complex 2 (PRC2), responsible for H2K27me3 deposition, as a barrier for trophoblast induction. The authors test this hypothesis by treating naïve hESCs with a specific EZH2 inhibitor (UNC1999), and evaluating at multiple levels (gene expression, protein expression by IF, scRNAseq, and human blastoid formation), its impact in trophoblast induction. Using this approach, the authors conclude that PRC2 limits the trophoblast induction, thus indicating the existence of an epigenetic restriction in naïve hESCs.

Overall, this study aims to clarify an open question of high interest for both fundamental and biomedical research. In this sense, the findings from this study are very timely and relevant. The analysis is exhaustive and technically well-performed, and the overall conclusions are experimentally well-supported. We believe that the scientific community will benefit very much from the large data-set provided at multiple levels from the proteomic, epigenomic, and transcriptomic levels. Finally, the functional conclusions raised by the authors set the foundations for exploring the implications of PRC2 in developmental disorders.

However, we think that the final model would be strengthened by additional experiments. According to Fig. 3G, trophoblast markers are similarly decorated with H3K27me3 both in naïve and in primed hPSCs. The deposition of this mark at these genes suggested to the authors that “PRC2-mediated H3K27me3 might restrict the trophoblast specification programme in naïve hPSCs” (a hypothesis tested by the subsequent functional analysis). We are wondering whether this functionality of PRC2 as a barrier for trophoblast differentiation from naïve hPSCs is conserved in primed hPSCs. As already reported in other studies (PMID: 33831366, 32048992, 32619492), trophoblast-like induction is achieved with much less efficiency from primed hPSCs. Can this reduced efficiency, in the induction of trophoblast from primed PSCs, be overcome in the presence of UNC1999?

Extra comments on the comparison of the two co-submissions:

At the functional level, although both studies point towards an implication of PRC2 in pluripotent-to-trophoblast transition, there are relevant discrepancies among both studies. Zijlmans et al. findings,

7well-supported at multiple experimental levels, indicate a function of PRC2 as a barrier during trophoblast induction. This is because, in the absence of instructive signals, PRC2 inhibition does not cause spontaneous differentiation or loss of pluripotency marks in naïve hPSCs. This dispensable role of PRC2 in naïve culture conditions is an observation that seems in line with previously published studies (PMID: 28864533; PMID: 28939884). Only when cultured under trophoblast differentiation media, hPSCs transient more efficiently towards trophoblast cells in the presence of the EZH2 inhibitor. In contrast, Kumar et al. study shows that the treatment with EPZ-6438 for 7 days results in the spontaneous differentiation of a fraction of naïve hESCs (yet to be quantified) towards trophoblast cells, in the absence of inductive signals. This would suggest that PRC2 functions as the active blocker of trophoblast differentiation. We agree that the discrepancies might result from technical differences between both studies.

Reviewer #3:

Remarks to the Author:

The manuscript by Zijlmans et al, Integrated Multi-Omics Analyses Reveal Polycomb Repressive Complex 2 Restricts Naïve Human Pluripotent Stem Cell to Trophoblast Fate Induction, uses a multi-layered approach to investigate the epigenetic landscape in naïve and primed pluripotent stem cells to assign regulatory relevance to the observations. The data appear solid, lots of very interesting techniques are being utilized making it clearly a very comprehensive and informative resource for the community.

That said, some of the interpretation needs to be adjusted or further strengthened. The major claim is the specific restriction of the trophoblast lineage via PRC2 mediated H3K27me3 (title, abstract, main text and model in Figure 6g), however the data do not support this as currently presented. PRC2 clearly regulates trophoblast genes, but also many other loci in naïve and primed cells as well as subsequent lineages. Hence it is not clear on what the specific claim towards trophoblast restriction is based. The specific assignment in Figure 6g of PRC2 as a result of the data shown seems not fitting. H3K27me levels decrease globally (as indicated in 6g) but the number of targeted loci notably increase, which is again something crucial that is not represented in the current simplified model. Lastly, selected genes are listed but their roles have not been functionally tested and hence it is difficult to position them in a model.

Specific comments:

The authors state: 'Importantly, we observed that genes associated with embryonic and extraembryonic lineage specification were marked by H3K27me3 in naïve hPSCs, similar to primed hPSCs (Figure 3G).'

8As noted above this doesn't match the major claim.

Then they continue: 'The presence of H3K27me3 at the promoters of key trophoblast regulators, therefore, raises the possibility that PRC2-mediated H3K27me3 might restrict the trophoblast specification programme in naive hPSCs.'

Figure 3g shows clearly that PRC2 mediated H3K27me3 is not specific to extraembryonic nor naive cells, and in fact its slightly lower in naive.

It needs to be clarified and shown what supports the authors assertions of a specific role rather than the expected general repression of ALL lineages?

Along those lines, the effect of PRC2 inhibition (EZHi) is only 2-fold (and only with the TSC induction conditions) and hence unclear if that supports a major role of PRC2 repression in that lineage.

As presented the single cell analysis doesn't add much beyond presenting nice data as a resource. More should be done here or it could be left out in favor of more detailed analysis at the chromatin level including more functional investigation.

Dnmt3l is most upregulated (Figure 1d) and its role in the global hypomethylation context is not intuitive as it should boost de novo activity. More could be done here or at least discussed beyond: 'However, in naive hPSCs, we detected a decrease in DNMT1 and its known interactor UHRF146, and an increase in TET1 and the catalytically inactive DNMT3L47, which are differences that could potentially reinforce the hypomethylated state of naive hPSCs.' I would agree that decrease in DNMT1 and UHRF1 as well as increase in TET1 fit the hypomethylation, but 3L is curious. Ref 47 that is cited here does not say anything relevant for this point regarding 3L and their knockdown of 3L had little effect in the transition.

The figures in general are very busy with lots of small labels. Much of this could be condensed in the main Figures and less relevant parts shown in the supplement. It currently is very much designed and presented has a resource data summary. Data could be analyzed further and more biological insights could be presented.

Methods should be written concisely, but should contain all elements necessary to allow interpretation and replication of the results. As a guideline, Methods sections typically do not exceed 3,000 words. The Methods should be divided into subsections listing reagents and techniques. When citing previous methods, accurate references should be provided and any alterations should be noted. Information must be provided about: antibody dilutions, company names, catalogue numbers and clone numbers for monoclonal antibodies; sequences of RNAi and cDNA probes/primers or company names and catalogue numbers if reagents are commercial; cell line names, sources and information on cell line identity and authentication. Animal studies and experiments involving human subjects must be reported in detail, identifying the committees approving the protocols. For studies involving human subjects/samples, a statement must be included confirming that informed consent was obtained. Statistical analyses and

information on the reproducibility of experimental results should be provided in a section titled “Statistics and Reproducibility”.

All Nature Cell Biology manuscripts submitted on or after March 21 2016 must include a Data availability statement as a separate section after Methods but before references, under the heading "Data Availability". For Springer Nature policies on data availability see <http://www.nature.com/authors/policies/availability.html>; for more information on this particular policy see <http://www.nature.com/authors/policies/data/data-availability-statements-data-citations.pdf>. The Data availability statement should include:

- Accession codes for primary datasets (generated during the study under consideration and designated as "primary accessions") and secondary datasets (published datasets reanalysed during the study under consideration, designated as "referenced accessions"). For primary accessions data should be made public to coincide with publication of the manuscript. A list of data types for which submission to community-endorsed public repositories is mandated (including sequence, structure, microarray, deep sequencing data) can be found here <http://www.nature.com/authors/policies/availability.html#data>.
- Unique identifiers (accession codes, DOIs or other unique persistent identifier) and hyperlinks for datasets deposited in an approved repository, but for which data deposition is not mandated (see here for details <http://www.nature.com/sdata/data-policies/repositories>).
- At a minimum, please include a statement confirming that all relevant data are available from the authors, and/or are included with the manuscript (e.g. as source data or supplementary information), listing which data are included (e.g. by figure panels and data types) and mentioning any restrictions on availability.
- If a dataset has a Digital Object Identifier (DOI) as its unique identifier, we strongly encourage including this in the Reference list and citing the dataset in the Methods.

We recommend that you upload the step-by-step protocols used in this manuscript to the Protocol Exchange. More details can found at www.nature.com/protocolexchange/about.

All imaging data should be accompanied by scale bars, which should be defined in the legend. Cropped images of gels/blots are acceptable, but need to be accompanied by size markers, and to retain visible background signal within the linear range (i.e. should not be saturated). The boundaries of panels with low background have to be demarked with black lines. Splicing of panels should only be considered if unavoidable, and must be clearly marked on the figure, and noted in the legend with a statement on whether the samples were obtained and processed simultaneously. Quantitative comparisons between samples on different gels/blots are discouraged; if this is unavoidable, it should only be performed for samples derived from the same experiment with gels/blots were processed in parallel, which needs to be stated in the legend.

- For line art, graphs, charts and schematics we prefer Adobe Illustrator (.AI), Encapsulated PostScript (.EPS) or Portable Document Format (.PDF). Files should be saved or exported as such directly from the application in which they were made, to allow us to restyle them according to our journal house style.
- We accept PowerPoint (.PPT) files if they are fully editable. However, please refrain from adding PowerPoint graphical effects to objects, as this results in them outputting poor quality raster art. Text used for PowerPoint figures should be Helvetica (preferred) or Arial.

SUPPLEMENTARY INFORMATION – Supplementary information is material directly relevant to the conclusion of a paper, but which cannot be included in the printed version in order to keep the manuscript concise and accessible to the general reader. Supplementary information is an integral part of a Nature Cell Biology publication and should be prepared and presented with as much care as the main display item, but it must not include non-essential data or text, which may be removed at the editor's discretion. All supplementary material is fully peer-reviewed and published online as part of the

HTML version of the manuscript. Supplementary Figures and Supplementary Notes are appended at the end of the main PDF of the published manuscript.

The total number of Supplementary Figures (not including the “unprocessed scans” Supplementary Figure) should not exceed the number of main display items (figures and/or tables (see our Guide to Authors and March 2012 editorial <http://www.nature.com/ncb/authors/submit/index.html#suppinfo>; <http://www.nature.com/ncb/journal/v14/n3/index.html#ed>). No restrictions apply to Supplementary Tables or Videos, but we advise authors to be selective in including supplemental data.

GUIDELINES FOR EXPERIMENTAL AND STATISTICAL REPORTING

REPORTING REQUIREMENTS – We are trying to improve the quality of methods and statistics reporting in our papers. To that end, we are now asking authors to complete a reporting summary that collects information on experimental design and reagents. The Reporting Summary can be found here <https://www.nature.com/documents/nr-reporting-summary.pdf> If you would like to reference the guidance text as you complete the template, please access these flattened versions at <http://www.nature.com/authors/policies/availability.html>.

15STATISTICS – Wherever statistics have been derived the legend needs to provide the n number (i.e. the sample size used to derive statistics) as a precise value (not a range), and define what this value represents. Error bars need to be defined in the legends (e.g. SD, SEM) together with a measure of centre (e.g. mean, median). Box plots need to be defined in terms of minima, maxima, centre, and percentiles. Ranges are more appropriate than standard errors for small data sets. Wherever statistical significance has been derived, precise p values need to be provided and the statistical test used needs to be stated in the legend. Statistics such as error bars must not be derived from $n < 3$. For sample sizes of $n < 5$ please plot the individual data points rather than providing bar graphs. Deriving statistics from technical replicate samples, rather than biological replicates is strongly discouraged. Wherever statistical significance has been derived, precise p values need to be provided and the statistical test stated in the legend.

We strongly recommend the presentation of source data for graphical and statistical analyses as a separate Supplementary Table, and request that source data for all independent repeats are provided when representative experiments of multiple independent repeats, or averages of two independent experiments are presented. This supplementary table should be in Excel format, with data for different figures provided as different sheets within a single Excel file. It should be labelled and numbered as one of the supplementary tables, titled “Statistics Source Data”, and mentioned in all relevant figure legends.

Author Rebuttal to Initial comments

Point-by-point Response to the Reviewers' Comments:

Manuscript ID: NCB-P46415B

We thank the reviewers for their constructive comments and helpful suggestions of our manuscript “*Integrated Multi-Omics Analyses Reveal Polycomb Repressive Complex 2 Restricts*

16Naive Human Pluripotent Stem Cell to Trophoblast Fate Induction". We appreciate their interest and the acknowledgement of novelty, importance, relevance, and timeliness of the work. We are grateful for the comment that our analysis is exhaustive and technically well-performed, and that the overall conclusions are experimentally well supported.

We are pleased to read that the reviewers think that the scientific community will benefit from the large data-set provided at multiple levels and that the conclusions of the work set the foundations for exploring the implications of PRC2 in developmental disorders. The reviewers also provided insightful feedback and excellent suggestions aimed at improving the manuscript.

We have addressed the points raised by the reviewers in our revised manuscript, and incorporating these suggestions has further improved our paper. We have provided additional computational and experimental evidence to better support our claims, as well as improved the discussion and interpretation of the data. In summary, we have now performed additional analyses on human blastoids and analysed several lineages including the primitive endoderm and AQP3 by immunofluorescence analysis, and have also conducted additional flow cytometry analyses of blastoids and trophoblast conversion experiments. Moreover, we have conducted additional analyses of scRNA-seq data, integrated our data set with human embryo data, and performed additional gene expression analyses. We have improved our bioinformatics analyses of the CUT&RUN data and added analysis of additional genes from different lineages. We have carried out additional experiments and analyses on the effects of PRC2 inhibition in naive human pluripotent stem cells and during trophoblast fate induction. We have repeated trophoblast fate induction experiments in independent cell lines and also in primed human pluripotent stem cells. Finally, to maximise accessibility of our data, we have created an online platform to explore the "acid extractome" protein abundance, chromatin-associated protein abundance, gene expression and histone modifications data sets (https://www.bioinformatics.babraham.ac.uk/shiny/shiny_omics/Shiny_omics ; Username: test; Password: justtesting).

We have also provided two loom files to explore UMAP-clustering and gene expression from our scRNA-seq data set and integration with human embryo data, which can be visualised here: https://scope.aertslab.org/#/HumanPluripotencyPRC2/*/welcome.

We therefore believe that the reviewers' suggestions have strengthened our manuscript significantly and we hope that the reviewers are satisfied with the changes and will be able to support publication of the work.

Reviewer 1

17In their manuscript, Zijlmans et al. examined the wide lineage propensity of naive human pluripotent stem cells by investigating chromatin modifications. By analysing proteins bound to chromatin in naive and primed H9 ES cells by ChEP, they found core components of the PRC2 complex, including EED, EZH2, and SUZ12, in naive PSCs were upregulated. An analysis of 43 histone post-translational modifications revealed that 23 of them were different between naive and primed PSCs. Notably, PRC2-mediated H3K27me_{2/3} was higher in naive PSCs than in primed PSCs. The level of H3K27me_{2/3} erasers KDM6A/B was also higher in naive PSCs. cCUT&RUN confirmed that H3K27me₃ in naive PSCs is globally higher than in primed PSCs but that there are more specific H3K27me₃ peaks in primed PSCs. Interestingly, the authors observed peaks of H3K27me₃ in trophoblast markers, including HAND1, CDX2, GATA2, GATA3 and KRT8/18, both in naive and primed PSCs, and a PRC2 inhibitor reduced H3K27me levels. From these observations, the authors concluded that PRC2 functions as a barrier of trophoblast formation in naive PSCs. This is a novel and important finding. However, prior to publication, I have the following comments about authors claims that I ask they resolve.

- 1) Although the authors showed common H3K27me₃ peaks in Fig. 4b, 4c and Fig. S4f, 4g after PRC2 inhibitor treatment, there are no data about primed and naive specific peaks. Primed and naive specific peaks also need to be shown. In addition, the authors should show primed peaks in Fig 4d.

Response 1.

We agree with the reviewer to include the primed- and naive-specific peaks. We have now updated Figure 4 to which we added these data (Fig. 4b, Extended Data Fig. 4e). We have also added tracks for primed hPSCs to Fig. 4c (previously Fig 4d), as suggested. These new data show that, like for the common peaks, H3K27me₃ is absent on primed- and naive-specific peaks following four days of PRC2 inhibition.

- 2) How many days did the authors culture naive cells in Fig 4i-l. Fig. 4l and S4k show very different gene expression profiles of naive and primed PSCs after PRC2i treatment. The authors should provide a reason and show the data of primed PSCs in Fig. 4i-k.

Response 2.

For Fig 4i-l (now Fig.4d-f, Extended Data Fig. 4j), naive cells were grown with the PRC2 inhibitor for four days. Primed cells were also grown in the inhibitor for four days. Therefore, the difference in gene expression changes upon PRC2 inhibition in primed and naive cells in these figures is not explained by a difference in the duration of PRC2 inhibition between the two cell types. We envision that the difference in changes in chromatin landscape between naive and primed hPSCs as we identified (shown in Fig 1-3), combined with the distinct gene regulatory

programs within the two different cell types, underlies the different gene expression responses, as we now indicate in the manuscript (page 14, lines 360-362). In the initial submission, we focused the main figure on naive cells and hence we opted to include the primed cells in the supplementary. We agree with the reviewer that these are noteworthy observations relevant to our study, and we have now included the data for primed cells in the main figure, as requested (Fig. 4g).

3) Fig. 5a-c: These are interesting and important experiments, but the figures suggest that PRC2i does not robustly induce trophoblast. For example, in Fig. 5c, naive->trophoblast induced 33.5% GATA3 positive cells, but naive PRC2i ->trophoblast induced only 29.5% GATA3 positive cells. If PRC2 functions as a barrier of trophoblast formation in naive hPSCs, the naive PSCs with PRC2i treatment from day -4 to 0 should differentiate trophoblast more efficiently than normal naive PSCs. I suggest the authors evaluate the efficiency by flow cytometry. I wonder if many naive PSCs die during the trophoblast induction in this manuscript. The cell number during induction is also important information.

Response 3.

To avoid confusion, we firstly would like to point out that, in our study, PRC2 inhibition does not induce the trophoblast fate robustly in the absence of trophoblast differentiation cues. In our hands, only PRC2 inhibition during trophoblast fate conversion leads to a significant increase in the proportion of trophoblast cells induced (Fig. 5b,c, 6e). As the reviewer rightfully points out, inhibition of PRC2 for 4 days in the naive state followed by PRC2 inhibitor removal during trophoblast conversion does not seem to have a strong effect on the efficiency with which trophoblast cells are induced in this assay.

As suggested by the reviewer, for further quantification, we have carried out flow cytometry analysis to assess the efficiency for trophoblast induction. We used TROP2, an established trophoblast stem cell surface marker (Lipinski et al., 1981; Kagawa et al., 2021), and analysed the effect of PRC2 inhibition. Although the efficiency of TROP2 induction at day 4 of conversion is low, as TROP2 appears to be a marker of late-stage trophoblast cell induction in this trophoblast fate induction protocol (Extended data Fig. 6b), we did corroborate the finding that PRC2 inhibition leads to an increase in trophoblast fate induction, in line with our previous findings (Extended Data Fig. 5i). When the inhibitor was applied in the naive state only or during conversion only, we observed an intermediate effect for both (Extended Data Fig. 5i). These new experiments now show that applying the inhibitor in the naive state only has a positive effect on trophoblast fate induction after trophoblast fate is induced, in line with PRC2 functioning as a barrier of trophoblast formation in naive hPSCs. The difference between the immunofluorescence and flow

cytometry data likely originate from differences in sensitivities, differences in protein abundance (GATA3 and TROP2), or differences in conversion efficiency between experiments.

*To further investigate the intermediate effect after temporal PRC2 inhibition, we reasoned that the dynamics by which H3K27me3 recovers after inhibitor removal would provide further insight (also see point 5/response 5 below). A relatively quick recovery of H3K27me3 might thereby have a less pronounced effect as compared to a slow recovery. To address this, we tested the effect of PRC2 inhibitor removal on H3K27me3 levels, which showed that H3K27me3 is reacquired starting 24h after inhibitor removal and to full levels by 72 hours after inhibitor removal (**Extended Data Fig. 4a**). These results not only suggest that PRC2 inhibition is reversible within a short time period but also that such reversibility likely accounts for the lack or reduced effect of PRC2 removal during trophoblast fate induction compared to continuous inhibitor treatment. These results are now mentioned in the manuscript (**page 13, line 348**).*

*Based on the suggestion of the reviewer, we assayed additional possible confounding effects that could explain differences in trophoblast conversion within the various conditions: cell death and growth. Although we do see moderate cell death one day after transfer of naive hPSCs grown in PXGL conditions into trophoblast medium (ASECRiAv), which is in line with observations previously reported for this protocol (Castel et al., 2020), viability and cell number were similar irrespective of whether cells were treated with a PRC2 inhibitor during trophoblast conversion. These data have been added to the revised manuscript (**Extended Data Fig. 5g; page 14, lines 376-378**). In addition, we note that it is the proportion of GATA3+ cells that increases in the presence of PRC2 inhibition during trophoblast fate induction (**Fig. 5c**). Therefore, the effect of PRC2 inhibition on trophoblast fate induction in the conversion experiments cannot be easily explained by differences in viability and cell number between control and inhibitor treated cells. Additionally, there is no appreciable cell death in the blastoid assay, where PRC2 inhibition also leads to an increase in trophoblast fate induction (**Fig 7 and Extended Data Fig 7**). Interestingly, the total number of cells per blastoid is significantly higher upon PRC2 inhibition at 60h of blastoid formation (**Extended Data Fig. 7b**), suggesting an advanced developmental time point of blastoids treated with the PRC2 inhibitor.*

4) To evaluate the differentiated cells (trophectoderm), scRNA-seq data should be compared with human embryo data by tSNE or UMAP.

Response 4.

We agree with the reviewer that it is important to validate our data with human embryo data, and we have now conducted additional integrative analyses and compared our scRNA-seq data to

human embryo scRNA-seq data (**Fig. 5h, 6 & Extended Data Fig. 6e-f**). For these analyses, we included scRNA-seq data from human embryo preimplantation data (Petropoulos et al., 2016) and from embryos grown ex vivo up to day 14 (Zhou et al., 2019). Cells assigned as trophoblast in our data align closely with the human embryo early trophoblast and medium/late trophectoderm lineage (**Fig. 6**), corroborating our previous conclusion that trophoblast fate is induced in our experiments. These results suggest that the trophoblast identity induced in our experiments is similar to that of the human embryo. The UMAPs also show an increase in the proportion of cells with trophoblast identities in the PRC2 inhibitor treated conditions (**Fig. 6e**), which is in line with our conclusion that PRC2 inhibition promotes the induction of trophoblast fate.

These results have now been added to **Fig. 5h, 6 and Extended Data Fig. 6e-g**. And in the Results section on **pages 15-16, lines 409-411**.

Additionally, next to making our scRNA-seq data publicly available via GEO, we have now made the scRNA-seq data available via the open access SCoPe platform developed by the laboratory of Dr. Stein Aerts. This allows inspecting and quantifying individual genes in a highly convenient and user-friendly browser, so as to facilitate and simplify usage of our unique resource data by other researchers. Both the integrated and non-integrated data are available providing an additional resource for the community.

https://scope.aertslab.org/#/HumanPluripotencyPRC2/*/welcome

We have also uploaded the loom files on the publicly available GEO platform, enabling download and reuse of our data for future studies.

5) How long does the effect of PRC2i continue? Does H3K27me3 continue to decrease after the withdrawal of PRC2i?

Response 5.

To address this issue (for which we discuss the biological implications in response 3), we have treated naive hPSCs with UNC1999 for 4 days and then removed the inhibitor to determine the dynamics of H3K27me3 recovery. We observed a low H3K27me3 signal by Western blot at 24h after withdrawal (as compared to no signal at 0h after withdrawal), while H3K27me3 is completely recovered after 72h of inhibitor withdrawal. These results suggest that the depletion of H3K27me3 by PRC2 inhibition is reversible and that recovery is relatively fast. We have now added these results to **Extended Data Fig. 4a** and in the text on **page 13, line 348**.

6) Fig. 6: In L374-377, the authors refer to improved conditions for blastoid formation.

21However, I could not find the method, which prevents judgement of the claim.

Response 6.

We apologise for this omission. We now included a section in the methods, where the conditions used for blastoid formation are present in more details (pages 26-27, lines 703- 706). Also, we now refer to the original study where the protocol that we used to create human blastoids was developed, compared to other methods and described in great detail (Kagawa et al. 2021).

7) Blastoids should be evaluated by immunofluorescence and scRNA-seq. Does PRC2i affect only trophoblast lineage in blastoids? The authors should count the number of blastoids+/- PRC2i cells in epiblast, hypoblast, and trophoblast. AQP3 should be stained in blastoids+/- PRC2i at 24 hours.

Response 7.

To address this comment, we have counted the number of epiblast, hypoblast and trophoblast cells in blastoids treated with and without PRC2 inhibition. We inhibited PRC2 during blastoid formation and analysed the epiblast, trophoblast and primitive endoderm lineages using immunofluorescence (NANOG for epiblast, GATA3 for trophoblast, and FOXA2 for hypoblast). In line with our previous results, we found that PRC2 inhibition significantly increased the induction of trophoblast cells at both 36h and 60h timepoints (Fig. 7c and Extended Data Fig. 7a, b, d). In contrast to the trophoblast lineage, we detected a decrease in the proportion of epiblast cells at 36h and 60h in PRC2 inhibitor-treated blastoids (Fig. 7c). Interestingly, we also observed a slight, albeit non-significant, increase in primitive endoderm cells at 60h in PRC2 inhibitor-treated blastoids (Extended Data Fig. 7c), suggesting that PRC2 may be involved in opposing primitive endoderm induction. Altogether, these results strengthen our previous conclusion that PRC2 opposes induction of the trophoblast lineage. These results have now been added to the manuscript (page 16, lines 429-433).

As we were able to address the key question of whether the three cell lineages of blastocysts/blastoids are altered in the same way following PRC2 inhibition using immunofluorescence microscopy, we have not performed the suggested scRNA-seq analysis of blastoids, which would be an endeavour by its own.

We thank the reviewer for the very interesting question on AQP3. In human embryos and in human blastoids, AQP3 is initially expressed in all cells and then becomes restricted to outer cells (Kagawa et al., 2021; Meistermann et al., 2021). We asked if this change in protein localisation is promoted by PRC2 inhibition by examining AQP3 by immunofluorescence at 24h of blastoid formation as suggested by the reviewer. We observed an expression pattern that is

*consistent with the accelerated restriction of APQ3 to the outer cells of blastoids following PRC2 inhibition. These results have been added to **Extended Data Fig. 7e** and in the manuscript (**page 17, lines 443-446**).*

8) Other ES or iPS cell lines should be checked.

Response 8.

*We agree with the reviewer on the importance of validating our main findings in other cell lines, to exclude a cell line specific effect. We therefore performed our naive to trophoblast conversion experiments in H9 naive ESCs, to complement our original data using iPSCs. In the new experiments, we assessed trophoblast fate induction efficiency by immunofluorescence for GATA3 (**Extended Data Fig. 5j**). We observed that PRC2 inhibition promoted trophoblast fate induction in the widely-used H9 hESCs in a similar fashion as for the iPSCs, further validating our original observations. These results have been added to the manuscript (**page 15, lines 384-385**). In the revised manuscript we specify which experiments were performed with which cell line (**page 21, lines 560-567**).*

Minor points

9) Fig. S2b: A Supplemental Table showing the expression level of each gene is recommended.

Response 9.

*We fully agree. We would like to clarify that Fig. S2b showed proteome data, not gene expression data. As suggested by the reviewer, we have added a new Supplemental Table providing the abundance of each protein in the acid extractome data (**Supplementary Table 5**). A Supplemental Table showing the expression level of each gene is also included (**Supplementary Table 2**). In addition, we created an online searchable tool to visualise all omics datasets presented in this paper, including the acid extractome data (https://www.bioinformatics.babraham.ac.uk/shiny/shiny_omics/Shiny_omics ; Username: test; Password: justtesting). Finally, all scRNA-seq data are available to browse online using the following link: https://scope.aertslab.org/#/HumanPluripotencyPRC2*/welcome.*

10) Fig. 6c and 6f do not show N or error bars.

Response 10.

*We have now added the results of three independent experiments in **Fig. 7b, f** (previously Fig. 6c, f).*

Reviewer 2

In this manuscript, Zijlmans and collaborators aim to challenge the current assumption that human naive pluripotent cells are “epigenetically unrestricted”. For that, the authors perform a massive multi-omic approach at the proteomic and genomic levels. By integrating these data, the authors postulate a potential role of the Polycomb Repressive Complex 2 (PRC2), responsible for H2K27me3 deposition, as a barrier for trophoblast induction. The authors test this hypothesis by treating naive hESCs with a specific EZH2 inhibitor (UNC1999), and evaluating at multiple levels (gene expression, protein expression by IF, scRNAseq, and human blastoid formation), its impact in trophectodermal induction. Using this approach, the authors conclude that PRC2 limits the trophoblast induction, thus indicating the existence of an epigenetic restriction in naive hESCs.

Overall, this study aims to clarify an open question of high interest for both fundamental and biomedical research. In this sense, the findings from this study are very timely and relevant. The analysis is exhaustive and technically well-performed, and the overall conclusions are experimentally well-supported. We believe that the scientific community will benefit very much from the large data-set provided at multiple levels from the proteomic, epigenomic, and transcriptomic levels. Finally, the functional conclusions raised by the authors set the foundations for exploring the implications of PRC2 in developmental disorders.

1) However, we think that the final model would be strengthened by additional experiments. According to Fig. 3G, trophectodermal markers are similarly decorated with H3K27me3 both in naive and in primed hPSCs. The deposition of this mark at these genes suggested to the authors that “PRC2-mediated H3K27me3 might restrict the trophoblast specification programme in naive hPSCs” (a hypothesis tested by the subsequent functional analysis). We are wondering whether this functionality of PRC2 as a barrier for trophoblast differentiation from naive hPSCs is conserved in primed hPSCs. As already reported in other studies (PMID: 33831366, 32048992, 32619492), trophectodermal-like induction is achieved with much less efficiency from primed hPSCs. Can this reduced efficiency, in the induction of trophoblast from primed PSCs, be overcome in the presence of UNC1999?

Response 11.

To follow up on this interesting question about whether the reduced efficiency of trophoblast fate induction from primed hPSCs compared to naive hPSCs can be overcome in the presence of UNC1999, we performed new experiments that exposed primed hPSCs to trophoblast stem cell culture conditions with and without PRC2 inhibition. We used the exact same strategy as in the naive hPSC conversion experiments, but starting from primed hPSCs instead. Our experiments

24

confirmed the very low efficiency of trophoblast cell induction when starting from primed hPSCs compared to naive hPSCs, with a maximum of 2.5% GATA3-positive cells at day 10 of conversion (compared with ~33% at day 4 of conversion starting from naive cells, Fig. 5c). Although we observed slight differences in the efficiency and dynamics of NANOG downregulation and GATA3 upregulation with or without PRC2 inhibition, the number of GATA3-positive cells remained very low throughout (Extended Data Fig. 5I). Therefore, the reduced efficiency in induction from primed PSCs cannot be overcome by UNC1999 alone. This suggests that, in addition to H3K27me3, which is present at trophoblast genes in primed cells (Fig. 4c), we believe that there are likely more prominent barriers in primed PSCs that oppose trophoblast conversion. These results have been included in the manuscript (page 15, lines 387-390).

2) Extra comments on the comparison of the two co-submissions:

At the functional level, although both studies point towards an implication of PRC2 in pluripotent-to-trophoblasts transition, there are relevant discrepancies among both studies. Zijlmans et al. findings, well-supported at multiple experimental levels, indicate a function of PRC2 as a barrier during trophoblast induction. This is because, in the absence of instructive signals, PRC2 inhibition does not cause spontaneous differentiation or loss of pluripotency marks in naive hPSCs. This dispensable role of PRC2 in naive culture conditions is an observation that seems in line with previously published studies (PMID: 28864533; PMID: 28939884). Only when cultured under trophoblast differentiation media, hPSCs transient more efficiently towards trophoblast cells in the presence of the EZH2 inhibitor. In contrast, Kumar et al. study shows that the treatment with EPZ-6438 for 7 days results in the spontaneous differentiation of a fraction of naive hESCs (yet to be quantified) towards trophoblast cells, in the absence of inductive signals. This would suggest that PRC2 functions as the active blocker of trophoblast differentiation. We agree that the discrepancies might result from technical differences between both studies.

Response 12.

We re-investigated the extent to which PRC2 inhibition potentially induces the trophoblast fate in naive conditions in the absence of trophoblast inductive cues. As before, we found that PRC2 inhibition using four days of UNC1999 (PRC2 inhibitor) treatment in PXGL media does not substantially increase the proportion of trophoblast cells. By scRNA-seq, we identified very few GATA3-positive cells in PXGL-cultured naive hPSCs, irrespective of whether PRC2 was inhibited or not (Extended Data Fig. 5b). By immunofluorescence analysis, we detected GATA3-positive cells in cultures grown with and without PRC2 inhibition, but, in each case, there were fewer than 0.3% GATA3-positive cells in the cultures and the number of GATA3-positive cells between untreated and PRC2-inhibited cultures was similar (Extended Data Fig. 5c). Additionally, using flow cytometry analysis for TROP2, there were ~0.5-1% TROP2-positive cells

25in PXGL cultures, and again this was irrespective of four or eight days of UNC1999 treatment or not (Extended Data Fig. 5d). Although these results reveal the (surprising) presence of a very small proportion (<1%) of GATA3-positive cells in PXGL-grown naive hPSC cultures, they also show that treatment of PXGL-cultured naive hPSCs treated with UNC1999 for four days does not induce the trophoblast fate in the absence of differentiation cues. These results strengthen our previous observations.

We hypothesised that the difference between our results and those of Kumar et al. might be due to the length of treatment (four days of PRC2 inhibition in our study versus seven days in the Kumar et al. study). We tested this by treating naive hPSCs in PXGL conditions with UNC1999 for eight days rather than four. We observed that after eight days of UNC1999 treatment in PXGL conditions, there were still less than 1% TROP2-positive cells in PRC2- inhibited cultures tested by flow cytometry (Extended Data Fig. 5d) and less than 0.6% GATA3-positive cells tested by immunofluorescence with or without PRC2i (Extended Data Fig. 5c). These results indicate that the discrepancy between our study and the Kumar et al. study is not caused by differences in the length of treatments. Therefore, we agree with the reviewer that it is likely that the observed differences are caused by other technical differences, such as the inhibitors used (UNC1999 versus EPZ-6438) or the naive hPSC culture conditions employed (PXGL versus t2iLGo media). These results have been added to the manuscript (page 14, lines 372-374).

Reviewer 3

The manuscript by Zijlmans et al, Integrated Multi-Omics Analyses Reveal Polycomb Repressive Complex 2 Restricts Naive Human Pluripotent Stem Cell to Trophoblast Fate Induction, uses a multi-layered approach to investigate the epigenetic landscape in naive and primed pluripotent stem cells to assign regulatory relevance to the observations. The data appear solid, lots of very interesting techniques are being utilized making it clearly a very comprehensive and informative resource for the community.

That said, some of the interpretation needs to be adjusted or further strengthened. The major claim is the specific restriction of the trophoblast lineage via PRC2 mediated H3K27me3 (title, abstract, main text and model in Figure 6g), however the data do not support this as currently presented. PRC2 clearly regulates trophoblast genes, but also many other loci in naive and primed cells as well as subsequent lineages. Hence it is not clear on what the specific claim towards trophoblast restriction is based. The specific assignment in Figure 6g of PRC2 as a result of the data shown seems not fitting. H3K27me levels decrease globally (as indicated in 6g) but the number of targeted loci notably increase, which is again something crucial that is not represented in the current simplified model. Lastly, selected genes are listed but their roles have

26not been functionally tested and hence it is difficult to position them in a model.

Response 13.

*We concur with the reviewer that the previous model needed to be revised. We have now revised the model **Fig. 7g**, including by adding the increased number of H3K27me3 peaks in primed versus naive hPSCs, which we agree with the reviewer is essential. We have removed the genes from the model. We have also added new data in the manuscript that better support the new model.*

We agree that the function of PRC2 in opposing alternative cell fate induction is not specific to the trophoblast. Please note that in our original submission, we did not claim that the effect of PRC2 inhibition is specific to the trophoblast lineage. As also referred to by the reviewer, H3K27me3 is present not only at trophoblast genes, but also at genes of other lineages in naive and primed hPSCs. One of the key advances in our current study is the demonstration that PRC2 is also restricting cell fate decisions in naive pluripotent stem cells – a state that was previously thought to be epigenetically unrestricted. Indeed, by discovering that PRC2 opposes trophoblast induction, our work lays the foundation for studying a role for PRC2 and other epigenetic pathways in controlling additional lineage fates from naive pluripotency.

To further address this comment of the reviewer in terms of the earliest lineage bifurcations in human embryogenesis, we have now included a quantification of primitive endoderm induction using the blastoid model. Interestingly, these data suggest that primitive endoderm might also be promoted by PRC2 inhibition, although the difference in the efficiency of primitive endoderm induction with or without PRC2 inhibition that we observed was non-significant.

*To address these points in the manuscript, we have now adjusted the interpretation, making it clearer that the cell fate restriction lineage function of PRC2 likely extends to other lineages in addition to the trophoblast (**page 18, lines 466-470; page 19, lines 509-510**).*

Specific comments:

1) The authors state: ‘Importantly, we observed that genes associated with embryonic and extraembryonic lineage specification were marked by H3K27me3 in naive hPSCs, similar to primed hPSCs (Figure 3G).’

As noted above this doesn't match the major claim.

Then they continue: ‘The presence of H3K27me3 at the promoters of key trophoblast regulators, therefore, raises the possibility that PRC2-mediated H3K27me3 might restrict the trophoblast specification programme in naive hPSCs.’

Figure 3g shows clearly that PRC2 mediated H3K27me3 is not specific to extraembryonic nor

27naive cells, and in fact its slightly lower in naive. It needs to be clarified and shown what supports the authors assertions of a specific role rather than the expected general repression of ALL lineages? Along those lines, the effect of PRC2 inhibition (EZHi) is only 2-fold (and only with the TSC induction conditions) and hence unclear if that supports a major role of PRC2 repression in that lineage.

Response 14.

*We have re-tuned our manuscript to make it clear that we do not consider PRC2 effects to be trophoblast-specific. Also, as referred to by the reviewer, we report H3K27me3 at trophoblast genes in both naive and primed hESCs, as well as at genes that are associated with other lineages. However, our observation that PRC2 is involved in lineage specification as early as in trophoblast is completely novel, and could only be observed due to the recent findings that naive PSCs readily make trophoblast in human (Castel et al., 2020; Cinkornpumin et al., 2020; Dong et al., 2020; Guo et al., 2021; Io et al., 2021), unlike in mouse. In our revised submission, we have extended the number of examples of non-trophoblast lineage genes that are marked by H3K27me3 to make it clear that although we find that H3K27me3 clearly opposes trophoblast induction, it also likely regulates other genes and lineages. We further discuss this issue at **page 18, lines 466-470** and **page 19, lines 509-510**.*

2) As presented the single cell analysis doesn't add much beyond presenting nice data as a resource. More should be done here or it could be left out in favor of more detailed analysis at the chromatin level including more functional investigation.

Response 15.

To address this issue, we now extensively compare our data to published human embryo transcriptional datasets (Fig. 6a-e). We observe that cells assigned as trophoblast in our data align closely with the human embryo early trophoblast and trophectoderm lineage. These new analyses suggest that the trophoblast identity induced in our experiments is similar to that of the human embryo. We have also added additional gene expression analyses in Fig. 5h and Extended Data Fig 6e, f, and improved accessibility of the data for future users. See also point 4 of Reviewer 1.

3) Dnmt3l is most upregulated (Figure 1d) and its role in the global hypomethylation context is not intuitive as it should boost de novo activity. More could be done here or at least discussed beyond: 'However, in naive hPSCs, we detected a decrease in DNMT1 and its known interactor UHRF146, and an increase in TET1 and the catalytically inactive DNMT3L47, which are differences that could potentially reinforce the hypomethylated state of naive hPSCs.' I would agree that decrease in DNMT1 and UHRF1 as well as increase in TET1 fit

the hypomethylation, but 3L is curious. Ref 47 that is cited here does not say anything relevant for this point regarding 3L and their knockdown of 3L had little effect in the transition.

Response 16.

We agree with the reviewer that the role of DNMT3L in global hypomethylation is not intuitive as DNMT3L is expected to boost de novo methyltransferase activity by stimulating DNMT3A/B. However, its knockdown during primed to naive hPSCs resetting does not affect DNA methylation levels (Patani et al., 2020), and it is possible that DNMT3L might have roles in human naive pluripotency that are methylation-independent. Of particular interest is to establish whether DNMT3L might recruit chromatin-modifying repressor proteins to silence transposable elements and other target regions, as has been recently reported in mouse PSCs and fibroblasts. We have now modified the discussion section by adding these suggested points (page 19, lines 495-503).

4) The figures in general are very busy with lots of small labels. Much of this could be condensed in the main Figures and less relevant parts shown in the supplement. It currently is very much designed and presented has a resource data summary. Data could be analyzed further and more biological insights could be presented.

Response 17.

As suggested by the reviewer, we have now made an effort to improve the readability of our manuscript. We increased label size to be between 5 and 7 pt, as per Nature Cell Biology standards, across all figures. We have also moved less relevant panels to the supplement (Extended Data Fig. 2a, f, Extended Data Fig. 4 e, f, h). For Fig. 2, we now emphasise more the interplay between histone marks and their writers/erasers and the differences between human and mouse naive pluripotent states. For Fig. 4, we now highlight that PRC2 inhibition removes H3K27me3 from gene promoters of trophoblast-associated genes. We have also reworked our analysis of H3K27me3 distribution on chromatin (Fig. 3, Extended Data Fig. 3), and we show that several genes that are associated with embryonic and extraembryonic lineage specification are, surprisingly, marked by H3K27me3 in the naive state. The presence in naive hPSCs of H3K27me3 at the promoters of key lineage regulators raises the likely possibility that PRC2-mediated H3K27me3 might oppose specification of multiple lineages in naive hPSCs. In addition, we integrated our scRNA-seq data with human embryo data (Fig. 6, Extended Data Fig. 6) and show that the trophoblast identity we induce in our differentiation experiments is similar to that of the human embryo. Please also see point 2, and point 4 of Reviewer 1. We feel that these changes now more strongly highlight the biological relevance of our work and thank the reviewer for their input.

References

Castel, G. *et al.* (2020) 'Induction of Human Trophoblast Stem Cells from Somatic Cells and Pluripotent Stem Cells', *Cell reports*, 33(8), p. 108419.

Cinkornpumin, J.K. *et al.* (2020) 'Naive Human Embryonic Stem Cells Can Give Rise to Cells with a Trophoblast-like Transcriptome and Methylome', *Stem cell reports*, 15(1), pp. 198–213.

Dong, C. *et al.* (2020) 'Derivation of trophoblast stem cells from naïve human pluripotent stem cells', *eLife*, 9. doi:10.7554/eLife.52504.

Guo, G. *et al.* (2021) 'Human naive epiblast cells possess unrestricted lineage potential', *Cell Stem Cell*, pp. 1040–1056.e6. doi:10.1016/j.stem.2021.02.025.

Io, S. *et al.* (2021) 'Capturing human trophoblast development with naive pluripotent stem cells in vitro', *Cell stem cell*, 28(6), pp. 1023–1039.e13.

Kagawa, H. *et al.* (2021) 'Human blastoids model blastocyst development and implantation', *Nature* [Preprint]. doi:10.1038/s41586-021-04267-8.

Lipinski, M. *et al.* (1981) 'Human trophoblast cell-surface antigens defined by monoclonal antibodies', *Proceedings of the National Academy of Sciences of the United States of America*, 78(8), pp. 5147–5150.

Loh, C.H. *et al.* (2021) 'Loss of PRC2 subunits primes lineage choice during exit of pluripotency', *Nature communications*, 12(1), p. 6985.

Meistermann, D. *et al.* (2021) 'Integrated pseudotime analysis of human pre-implantation embryo single-cell transcriptomes reveals the dynamics of lineage specification', *Cell stem cell*, 28(9), pp. 1625–1640.e6.

Patani, H. *et al.* (2020) 'Transition to naïve human pluripotency mirrors pan-cancer DNA hypermethylation', *Nature Communications*. doi:10.1038/s41467-020-17269-3.

Petropoulos, S. *et al.* (2016) 'Single-Cell RNA-Seq Reveals Lineage and X Chromosome Dynamics in Human Preimplantation Embryos', *Cell*, 165(4), pp. 1012–1026.

Zhou, F. *et al.* (2019) 'Reconstituting the transcriptome and DNA methylome landscapes of human implantation', *Nature*, 572(7771), pp. 660–664.

Decision Letter, first revision:

30Subject: Your manuscript, NCB-P46415B
Message: Our ref: NCB-P46415B

11th March 2022

Dear Dr. Pasque,

Thank you for submitting your revised manuscript "Integrated Multi-Omics Reveal Polycomb Repressive Complex 2 Restricts Human Trophoblast Induction" (NCB-P46415B). It has now been seen by the original referees and their comments are below. The reviewers find that the paper has improved in revision, and therefore we'll be happy in principle to publish it in Nature Cell Biology, pending minor revisions to comply with our editorial and formatting guidelines.

The current version of your manuscript is in a PDF format. Please email us a copy of the file in an editable format (Microsoft Word or LaTeX)-- we can not proceed with PDFs at this stage.

Thank you again for your interest in Nature Cell Biology. Please do not hesitate to contact me if you have any questions.

Sincerely,

Jie Wang, PhD
Senior Editor
Nature Cell Biology

Tel: +44 (0) 207 843 4924
email: jie.wang@nature.com

Reviewer #1 (Remarks to the Author):

31The authors thoroughly and carefully responded to all the issues raised by my comments. This required a considerable amount of work and the inclusion of data from quite a few new experiments. I believe that the series of studies they have presented is a valuable report on the role of PRC2 and trophoblast differentiation in naïve human pluripotent stem cells and represents a significant novelty.

Reviewer #2 (Remarks to the Author):

We would like to congratulate the authors from both teams for the thorough revision of their manuscript. We are glad that they appreciated our constructive revision and helpful suggestions for the improvement of their studies. The new data provided is of high quality and further supports the initial conclusions raised in the first version of their manuscripts. We believe that both studies offer an important source of data for the scientific community, and provide relevant functional insights into the mechanism controlling the lineage specification of human pluripotent cells. Although the discrepancy on whether PRC2 inhibition is sufficient or not to induced trophoblast fate remains unresolved between the co-submitted studies, we acknowledge that both teams have dedicated intense efforts to resolving it. As initially pointed out, the new data indicates that some technical variations are likely responsible for the observed differences. Likewise, both studies agree on the existence of an epigenetic restriction in human naïve pluripotent stem cells and, that PRC2 activity results in chromatin barrier for alternative cell fates. These major conclusions provide a relevant framework to understand cell-type specification during human early embryonic development. We are very pleased to support the publication of both studies in Nature Cell Biology.

Reviewer #3 (Remarks to the Author):

The reviewer have addressed all the prior concerns and no further issues are raised.

Decision Letter, final requests:

Subject: NCB: Your manuscript, NCB-P46415B

Message: Our ref: NCB-P46415B

23rd March 2022

Dear Dr. Pasque,

32Thank you for your patience as we've prepared the guidelines for final submission of your Nature Cell Biology manuscript, "Integrated Multi-Omics Reveal Polycomb Repressive Complex 2 Restricts Human Trophoblast Induction" (NCB-P46415B). Please carefully follow the step-by-step instructions provided in the attached file, and add a response in each row of the table to indicate the changes that you have made. Ensuring that each point is addressed will help to ensure that your revised manuscript can be swiftly handed over to our production team.

We would like to start working on your revised paper, with all of the requested files and forms, as soon as possible (preferably within one week). Please get in contact with us if you anticipate delays.

In recognition of the time and expertise our reviewers provide to Nature Cell Biology's editorial process, we would like to formally acknowledge their contribution to the external peer review of your manuscript entitled "Integrated Multi-Omics Reveal Polycomb Repressive Complex 2 Restricts Human Trophoblast Induction". For those reviewers who give their assent, we will be publishing their names alongside the published article.

Nature Cell Biology offers a Transparent Peer Review option for new original research manuscripts submitted after December 1st, 2019. As part of this initiative, we encourage our authors to support increased transparency into the peer review process by agreeing to have the reviewer comments, author rebuttal letters, and editorial decision letters published as a Supplementary item. When you submit your final files please clearly state in your cover letter whether or not you would like to participate in this initiative. Please note that failure to state your preference will result in delays in accepting your manuscript for publication.

Cover suggestions

As you prepare your final files we encourage you to consider whether you have any images or illustrations that may be appropriate for use on the cover of Nature Cell Biology.

Nature Cell Biology has now transitioned to a unified Rights Collection system which will allow our Author Services team to quickly and easily collect the rights and permissions required to publish your work. Approximately 10 days after your paper is formally accepted, you will receive an email in providing you with a link to complete the grant of rights. If your paper is eligible for Open Access, our Author Services team will also be in touch regarding any additional information that may be required to arrange payment for your article.

Please note that Nature Cell Biology is a Transformative Journal (TJ). Authors may publish their research with us through the traditional subscription access route or make their paper immediately open access through payment of an article-processing charge (APC). Authors will not be required to make a final decision about access to their article until it has been accepted. Find out more about Transformative Journals

Authors may need to take specific actions to achieve compliance with funder and institutional open access mandates. If your research is supported by a funder that requires immediate open access (e.g. according to Plan S principles) then you should select the gold OA route, and we will direct you to the compliant route where possible. For authors selecting the subscription publication route, the journal's standard licensing terms will need to be accepted, including self-archiving policies. Those licensing terms

will supersede any other terms that the author or any third party may assert apply to any version of the manuscript.

For information regarding our different publishing models please see our Transformative Journals page. If you have any questions about costs, Open Access requirements, or our legal forms, please contact ASJournals@springernature.com.

[REDACTED]

If you have any further questions, please feel free to contact us.

Best regards,

Ziqian Li
Editorial Assistant
Nature Cell Biology

On behalf of

Jie Wang, PhD
Senior Editor
Nature Cell Biology

Tel: +44 (0) 207 843 4924
email: jie.wang@nature.com

Reviewer #1:

Remarks to the Author:

The authors thoroughly and carefully responded to all the issues raised by my comments. This required a considerable amount of work and the inclusion of data from quite a few new experiments. I believe that the series of studies they have presented is a valuable report on the role of PRC2 and trophoblast differentiation in naïve human pluripotent stem cells and represents a significant novelty.

Reviewer #2:

Remarks to the Author:

We would like to congratulate the authors from both teams for the thorough revision of their manuscript. We are glad that they appreciated our constructive revision and helpful suggestions for the improvement of their studies. The new data provided is of high quality and further supports the initial conclusions raised in the first version of their manuscripts. We believe that both studies offer an important source of data for the scientific community, and provide relevant functional insights into the mechanism controlling the lineage specification of human pluripotent cells. Although the discrepancy on whether PRC2 inhibition is sufficient or not to induced trophoblast fate remains unresolved between the co-submitted studies, we acknowledge that both teams have dedicated intense efforts to resolving it. As initially pointed out, the new data indicates that some technical variations are likely responsible for the observed differences. Likewise, both studies agree on the existence of an epigenetic restriction in human naïve pluripotent stem cells and, that PRC2 activity results in chromatin barrier for alternative cell fates. These major conclusions provide a relevant framework to understand cell-type specification during human early embryonic development. We are very pleased to support the publication of both studies in Nature Cell Biology.

Reviewer #3:

Remarks to the Author:

The reviewer have addressed all the prior concerns and no further issues are raised.

Author Rebuttal, first revision:

Point-by-point Response to the Reviewers' Comments:

Manuscript ID: NCB-P46415B

We thank the reviewers for their constructive comments and helpful suggestions of our manuscript "*Integrated Multi-Omics Analyses Reveal Polycomb Repressive Complex 2 Restricts Naive Human Pluripotent Stem Cell to Trophoblast Fate Induction*". All comments from the reviewers have been addressed.

36Reviewer 1

Remarks to the Author:

The authors thoroughly and carefully responded to all the issues raised by my comments. This required a considerable amount of work and the inclusion of data from quite a few new experiments. I believe that the series of studies they have presented is a valuable report on the role of PRC2 and trophoblast differentiation in naïve human pluripotent stem cells and represents a significant novelty.

Reviewer 2

Remarks to the Author:

We would like to congratulate the authors from both teams for the thorough revision of their manuscript. We are glad that they appreciated our constructive revision and helpful suggestions for the improvement of their studies. The new data provided is of high quality and further supports the initial conclusions raised in the first version of their manuscripts. We believe that both studies offer an important source of data for the scientific community, and provide relevant functional insights into the mechanism controlling the lineage specification of human pluripotent cells. Although the discrepancy on whether PRC2 inhibition is sufficient or not to induced trophoblast fate remains unresolved between the co-submitted studies, we acknowledge that both teams have dedicated intense efforts to resolving it. As initially pointed out, the new data indicates that some technical variations are likely responsible for the observed differences. Likewise, both studies agree on the existence of an epigenetic restriction in human naïve pluripotent stem cells and, that PRC2 activity results in chromatin barrier for alternative cell fates. These major conclusions provide a relevant framework to understand cell-type specification during human early embryonic development. We are very pleased to support the publication of both studies in Nature Cell Biology.

We have added new results (Extended Data Figure 7g) where we compared the inhibitor used in this manuscript (UNC1999) versus the one used by Kumar et al. (EZP-6438) by treating naive hPSCs for 7 days in PXGL medium. We observed no spontaneous differentiation with either inhibitor, and so we believe that the difference in spontaneous differentiation between our manuscript and Kumar et al. can be attributed to the culture media used for naive hPSCs. In particular, PXGL media contains a WNT antagonist (XAV939) whereas t2iLGö media contains a WNT activator (CHIR99021), and shielding from WNT stimulation protects naive hPSCs against the induction of differentiation-associated genes¹.

Reviewer 3

Remarks to the Author:

The reviewer have addressed all the prior concerns and no further issues are raised.

References

1. Rostovskaya, M., Stirparo, G.G. & Smith, A. Capacitation of human naïve pluripotent stem cells for multi-lineage differentiation. *Development* **146** (2019).

Final Decision Letter:

Subject: Decision on Nature Cell Biology submission NCB-P46415C

Message:

Dear Dr Pasque,

I am pleased to inform you that your manuscript, "Integrated Multi-Omics Reveal Polycomb Repressive Complex 2 Restricts Human Trophoblast Induction", has now been accepted for publication in Nature Cell Biology.

38If you have any questions about our publishing options, costs, Open Access requirements, or our legal forms, please contact ASJournals@springernature.com

Please note that Nature Cell Biology is a Transformative Journal (TJ). Authors may publish their research with us through the traditional subscription access route or make their paper immediately open access through payment of an article-processing charge (APC). Authors will not be required to make a final decision about access to their article until it has been accepted. Find out more about Transformative Journals

Authors may need to take specific actions to achieve compliance with funder and institutional open access mandates. If your research is supported by a funder that requires immediate open access (e.g. according to Plan S principles) then you should select the gold OA route, and we will direct you to the compliant route where possible. For authors selecting the subscription publication route, the journal's standard licensing terms will need to be accepted, including self-archiving policies. Those licensing terms will supersede any other terms that the author or any third party may assert apply to any version of the manuscript.

If you have not already done so, we strongly recommend that you upload the step-by-step protocols used in this manuscript to the Protocol Exchange (www.nature.com/protocolexchange), an open online resource established by Nature Protocols that allows researchers to share their detailed experimental know-how. All uploaded protocols are made freely available, assigned DOIs for ease of citation and are fully searchable through nature.com. Protocols and Nature Portfolio journal papers in which they are used can be linked to one another, and this link is clearly and prominently visible in the online versions of both papers. Authors who performed the specific experiments can act as primary authors for the Protocol as they will be best placed to share the methodology details, but the Corresponding Author of the present research paper should be included as one of the authors. By uploading your Protocols to Protocol Exchange, you are enabling researchers to more readily reproduce or adapt the methodology you use, as well as increasing the visibility of your protocols and papers. You can also establish a dedicated page to collect your lab Protocols. Further information can be found at www.nature.com/protocolexchange/about

With kind regards,

Jie Wang, PhD
Senior Editor
Nature Cell Biology

Tel: +44 (0) 207 843 4924
email: jie.wang@nature.com